# Dominant *ARF3* variants disrupt Golgi integrity and cause a neurodevelopmental disorder recapitulated in zebrafish

Giulia Fasano [1,20], Valentina Muto [1,20], Francesca Clementina Radio [1,20], Martina Venditti[1], Niloufar Mosaddeghzadeh[2], Simona Coppola[3], Graziamaria Paradisi[1,4], Erika Zara[1,5], Farhad Bazgir [2], Alban Ziegler[6,7], Giovanni Chillemi [4,8], Lucia Bertuccini [9], Antonella Tinari [10], Annalisa Vetro[11], Francesca Pantaleoni [1], Simone Pizzi[1], Libenzio Adrian Conti [12], Stefania Petrini [12], Alessandro Bruselles [13], Ingrid Guarnetti Prandi[4], Cecilia Mancini[1], Balasubramanian Chandramouli[14], Magalie Barth[7], Céline Bris[6,7], Donatella Milani[15], Angelo Selicorni[16], Marina Macchiaiolo[1], Michaela V. Gonfiantini[1], Andrea Bartuli[1], Riccardo Mariani[17], Cynthia J. Curry[18], Renzo Guerrini[11], Anne Slavotinek[18], Maria Iascone[19], Bruno Dallapiccola[1], Mohammad Reza Ahmadian [2], Antonella Lauri [1]✉ & Marco Tartaglia [1]✉

Vesicle biogenesis, trafficking and signaling *via* Endoplasmic reticulum-Golgi network support essential developmental processes and their disruption lead to neurodevelopmental disorders and neurodegeneration. We report that de novo missense variants in *ARF3*, encoding a small GTPase regulating Golgi dynamics, cause a developmental disease in humans impairing nervous system and skeletal formation. Microcephaly-associated *ARF3* variants affect residues within the guanine nucleotide binding pocket and variably perturb protein stability and GTP/GDP binding. Functional analysis demonstrates variably disruptive consequences of ARF3 variants on Golgi morphology, vesicles assembly and trafficking. Disease modeling in zebrafish validates further the dominant behavior of the mutants and their differential impact on brain and body plan formation, recapitulating the variable disease expression. In-depth in vivo analyses traces back impaired neural precursors' proliferation and planar cell polarity-dependent cell movements as the earliest detectable effects. Our findings document a key role of ARF3 in Golgi function and demonstrate its pleiotropic impact on development.

The Golgi apparatus is a polarized, membrane network-built organelle organized as a series of flattened, stacked pouches (*cisternae*) held together by matrix proteins and microtubules and structured into the *cis* and *trans*-Golgi compartments. It is responsible for transporting, modifying, and packaging proteins and lipids into vesicles for their targeted delivery[1–4]. Golgi also provides signaling platforms for the regulation of a wide range of cellular processes (e.g., cell polarity, stress response, and mitosis) suggesting a role as a cell sensor and regulator similarly to other organelles, which ultimately orchestrates development[4–6]. Golgi function is highly depending upon its rapid

structural remodeling in response to different physiological stimuli, which is attained via tightly regulated processes involving ribbon disassembly, tubulovesicular conversion as well as stacks repositioning[4,7]. Of note, stimulus-dependent Golgi repositioning in the apical radial glia precursors is crucial to maintain stem cell identity, likely controlling polarized trafficking during corticogenesis[8].

In the last years, several Mendelian disorders have causally been related to the defective or aberrant function of components of the transport machinery[9]. In particular, disruption of Golgi organization underlies several diseases, most of which share altered neurodevelopment and early-onset neurodegeneration[10–12]. In these disorders, collectively named "Golgipathies", recurrent features include microcephaly, CNS defects (e.g., delayed myelination, cortical atrophy, abnormal corpus callosum, and pontocerebellar hypoplasia) and developmental delay (DD)/intellectual disability (ID)[12,13]. More generally, defective vesicle formation and unbalanced trafficking have been recognized as prominent patho-mechanisms in several neurodevelopmental disorders with CNS malformations and microcephaly[14–16], and neurodegenerative conditions[17,18].

The six members of the ADP-ribosylation factors (ARF) family of small GTPases (ARF1, ARF3-6, and ARF2, missing in primates) regulate key events of Golgi structure and function, vesicular biogenesis and cargo transport. ARF function is broadly overlapping and redundant in cells[19,20], where they participate in bidirectional membrane trafficking required for endocytosis and anterograde/retrograde transport, including protein recycling to the membranes or their degradation[20–24].

These proteins bind to guanine nucleotides with high affinity and specificity and cycle between a GTP (active) and GDP (inactive)-bound form[22,25]. Similar to other members of the RAS superfamily, release of GDP is stimulated by specific guanine nucleotide exchange factors (ARFGEFs), indirectly favoring binding to GTP[22,26,27]. As a consequence of the conformational change promoted by GTP, the N-terminal myristoylated region is exposed, allowing anchoring of the active GTPase to membranes of different organelles, including cis and trans-Golgi, plasma membrane and endosomes, where these proteins exert their function[22,25,28–30]. Moreover, the conformational rearrangement of the switch 1 [SW1] and switch 2 [SW2] regions controls the interaction with a number of effectors and regulators[30–32]. The intrinsic slow GTPase activity of ARFs is accelerated by specific GTPase-activating proteins (ARFGAPs), which result in protein inactivation and release from membrane[22,29,33,34].

By interacting with coat and adaptor proteins via this switch system[1,22], ARF proteins support various steps of the biosynthetic trafficking, such as COP-I vesicle formation and budding, which are essential for anterograde/retrograde cargo transport[35,36]. ARF proteins can also recruit non-coat Golgi-specific factors to membranes (e.g., Golgin-160 and GCC88)[37], which are fundamental for Golgi structural integrity[38,39], and thereby contribute to the control of Golgi and organelle structural organization and function[20,23,40,41].

The use of constitutively active (CA, GTP-bound) and dominant negative (DN, GDP-bound) ARF mutants as well as ARF silencing in cells has demonstrated the variable consequences of aberrant ARF function on Golgi integrity, morphology, vesicle formation, and recycling[20,36], and the redundant roles among the various ARF proteins. CA mutants (i.e., ARF1/3$^{Q71L}$) produce loss of the Golgi ribbon-like structure with an overall expansion of the Golgi and COP-I compartments due to conspicuous vesiculation[20,23]. Conversely, DN mutants (e.g., ARF3$^{N126I}$) induce a different pattern of Golgi fragmentation, with the dispersion of the coat proteins and COP-I disassembly[20]. The latter resembles the ARF poisoning effect triggered by brefeldin A (BFA), which blocks the normal activation of all ARF proteins by binding ARF1-GDP-GEF[42].

ARF-regulated Golgi dynamics during mitosis are crucial for cell division and cytokinesis[43–45]. Ultimately, by controlling Golgi structure, function, cargo sorting, and trafficking, ARFs actively participate in the fine regulation of key events during embryogenesis (i.e., cell polarity establishment and migration during gastrulation, neuronal maturation, and tissue morphogenesis)[46]. A hyperactive or reduced arf1 function in zebrafish results in altered body plan and head development[47,48]. In particular, hyperactive arf1 induces body plan alterations that are consistent with altered planar cell polarity (PCP)[47].

Notwithstanding their emerging pivotal roles in development, mutations in ARF genes have only recently been linked to human disease, with activating missense variants of ARF1 (MIM: 103180) causing a rare dominant malformation of cortical development resulting from defective neuronal migration (MIM: 618185)[49]. More recently, during the revision of this work, two pathogenic variants in ARF3 were described in three individuals with a variable neurodevelopmental phenotype, and microcephaly in the most severe case[50].

Here, we report five de novo missense ARF3 variants underlying a similar disorder affecting CNS and skeletal development. In silico and in vitro analyses provide evidence of a variable impact of mutations on protein stability, activity, Golgi integrity, vesicle formation, and cargo recycling. In-depth investigation in zebrafish corroborates the dominant nature of mutations, confirms a diverse effect on Golgi morphology during early embryogenesis, and recapitulates the variable brain and axial defects observed in patients. Experiments in live embryos further trace back the effect of aberrant ARF3 function to an altered balance of cell proliferation and death within the anterior developing brain and to impaired PCP-dependent cell axes formation.

## Results

### ARF3 mutations cause a developmental disorder affecting CNS and skeletal formation

In the frame of a research program dedicated to subjects affected by unclassified diseases, trio-based exome sequencing allowed us to identify a previously unreported de novo ARF3 variant, c.379A>G (p.Lys127Glu; NM_001659.2), as the putative disease-causing event in a girl (Subject 1) with a severe syndromic neurodevelopmental disorder characterized by growth restriction, severe microcephaly, progressive diffuse cortical atrophy, hypoplasia of corpus callosum and other brain anomalies at MRI (i.e., lateral ventricular enlargement, severe brainstem hypoplasia particularly affecting the pons, cerebellar inferior vermis hypoplasia), seizures, profound DD/ID and skeletal involvement (i.e., 11 rib pairs and severe scoliosis), inguinal hernia and congenital heart defects (CHD). (Supplementary Fig. 1, Supplementary Tables 1 and 2 and clinical reports). Whole exome sequencing (WES) data analysis excluded the presence of other relevant variants compatible with known Mendelian disorders based on their expected inheritance model and associated clinical presentation, and high-resolution SNP array analysis excluded the occurrence of genomic rearrangements. The missense change, which had not previously been reported in population databases, affected an invariantly conserved residue among orthologs, paralogs, and other structurally related GTPases of the RAS family (Supplementary Fig. 2a). Through networking and GeneMatcher[51], we identified four additional subjects with de novo ARF3 missense variants, which had not been reported in ExAC/gnomAD and involved amino acid residues located in regions highly constrained for variation (Supplementary Table 2, Supplementary Fig. 2a, b). No additional candidate variants in clinically associated genes were identified in any patients (WES statistics and data output, Supplementary Tables 3–7). Affected residues but Leu$^{12}$ were conserved among ARF3 orthologs and paralogs, and three of them were also conserved among other RAS GTPases (Supplementary Fig. 2a). The identified missense variants affected residues whose corresponding positions in other GTPases of the RAS superfamily had previously been associated with human disease (Supplementary Table 8). Among these, the same Lys-to-Glu substitution at codon 127 in Subject 1 was recently reported to affect the corresponding residue in ARF1 in a patient with DD, microcephaly, periventricular heterotopia, progressive cerebral atrophy, and epilepsy[49].

Affected subjects showed variable degrees of DD/ID associated with brain and skeletal anomalies (Supplementary Fig. 1, Supplementary Table 1 and clinical reports). No characteristic craniofacial gestalt was noted, with only minor craniofacial features reported in single patients, mainly related to microcephaly (Supplementary Fig. 1a, b). Similar to Subject 1, Subject 2 (p.Leu12Val; p.Asp67Val) showed microcephaly, profound DD/ID, absence of speech and language development, progressive diffuse cortical atrophy with diminished hemispheric white matter, thin corpus callosum, progressive pontocerebellar hypoplasia without the involvement of the cerebellar vermis, hypotonia, microsomia, and consistent skeletal defects (Supplementary Fig. 1b, c; Supplementary Table 1 and clinical reports). A comparable but less severe condition was also observed in Subject 4 (p.Asp93Asn), who manifested hypotonia, severe DD/ID, delayed speech and language development, post-natal microcephaly, thinning of the corpus callosum as well as milder skeletal defects (Supplementary Fig. 1, Supplementary Table 1 and clinical reports). Subject 3 (p.Pro47Ser) and Subject 5 (p.Thr32Asn) showed the mildest phenotype with DD/ID and delayed (Subject 3) or severely delayed (Subject 5) speech and language development. Subject 3 also shows early-onset seizures and severe hypoplasia of the anterior part of the temporal lobe associated with hypomyelination and thin corpus callosum, while Subject 5 showed hypoplasia of the corpus callosum, mild white matter involvement in periventricular and supraventricular areas, and a large cisterna magna with a milder skeletal involvement (Supplementary Fig. 1b, c, Supplementary Table 1 and clinical reports).

## Disease-associated *ARF3* variants variably affect protein stability and function

The identified disease-associated variants affected residues spotted throughout the coding sequence except for the C-terminal region (Fig. 1a). First, we examined the possible functional consequences of each amino acid substitution using a three-dimensional structure of the GTPase recently solved by X-ray diffraction[52] as reference. All residues except for Leu[12] cluster within or close to the GTP/GDP binding pocket (Fig. 1b). Lys[127] is one of the four residues of the NKXD motif directly mediating binding to the ribose ring of GTP/GDP[32], and substitution of the positively charged residue with a negatively charged glutamate was predicted to affect nucleotide binding (Fig. 1c). Similarly, Thr[32] contributes to stabilizing the GTP/GDP binding via direct hydrogen bonding with one oxygen atom of the α phosphate (Fig. 1c). While the conservative Thr to Asn substitution was predicted to result in a steric hindrance. Asp[93] does not directly contact GTP, even though it participates in the overall structure of the nucleotide-binding pocket by a direct hydrogen bond with the lateral chain of Lys[127] (Fig. 1c). The Asp-to-Asn change was anticipated to disrupt the interaction between the two residues, destabilizing GTP/GDP binding (Fig. 1d). Pro[47] and Asp[67] were predicted to affect ARF3 GTPase activity. Pro[47] is located within the SW1 region, which plays a key role in the catalytic activity of the GTPase and the conformational rearrangement mediating binding to effectors[22,32]. Substitution of this non-polar residue with a polar serine was expected to strongly perturb the functional behavior of the protein. Similarly, Asp[67] participates in the coordination of the Mg$^{2+}$ ion through direct hydrogen bonds with a water molecule[31] (Fig. 1c), and contributes to the regulation of GDP/GTP binding upon the "inter-switch toggle" mechanism[53]; its substitution with valine was predicted to considerably perturb GTP/GDP binding[30] and the overall organization of the nucleotide-binding pocket. Similar pathogenic variants in RAS proteins were predicted to destabilize the binding to GTP/GDP[54,55], and ARF1 substitutions in Lys[127], Asp[67], and Asp[93] were documented to have a deleterious effect in yeast[56]. Leu[12] (in *cis* with p.Asp67Val in Subject 2) is located within the flexible N-terminal myristoylated alpha helix implicated in membrane-cytoplasm shuttling[22,30], a region that has not been resolved structurally. No obvious consequence could be hypothesized for p.Leu12Val. However,

a possible impact on nucleotide binding and GTPase activity cannot be excluded[57]. Of note, while Thr[32], Asp[93], and Lys[127] map regions of the GTPase not directly involved in intermolecular contacts, Pro[47] and Asp[67] lie in regions close to the surface of the GTPase interacting with effectors/regulators[58], which does not rule out the possibility of a more complex functional behavior of the p.Pro47Ser and p.Asp67Val changes. To explore the structural and functional consequences of these two substitutions, we built a model of ARF3 interacting with the cytosolic coat protein complex (COP) formed by γ-COP (COPG1) and ζ-COP (COPZ1) starting from an available GTP-bound ARF1:COPG1-COPZ1 complex (PDB: 3TJZ) as template[59] (Fig. 1e–h). The model for the wild-type (WT) ARF3 protein was validated by a 500-ns molecular dynamics (MD) simulation, documenting the conservation of all known interactions with GTP and Mg$^{2+}$ (Fig. 1e, f; Supplementary Table 9). The ARF3:COPG1 interface is stabilized by an intermolecular hydrogen bonding network involving Arg[19], Thr[48], and Asn[84] ARF3 residues (Supplementary Table 10). We assessed the structural perturbations due to the introduced p.Pro47Ser and p.Asp67Val changes using the same time frame. A minor impact on the ARF3 surface interacting with COPG1 was evident in the simulation when introducing the p.Asp67Val substitution (Fig. 1g; Supplementary Table 10). As predicted by the structural inspection, this change resulted instead in a significant rearrangement of the nucleotide-binding pocket with a reduction of the interactions of Lys[127] and Thr[45] with GTP (Supplementary Table 9). The Pro-to-Ser substitution at codon 47 did not significantly affect ARF3 binding to GTP (Supplementary Table 9), while a dramatic perturbation of the intermolecular binding network with COPG1 due to a substantial rearrangement of the SW1 region was observed (Fig. 1h; Supplementary Table 10). Consistently, essential dynamics analysis documented a major effect of p.Pro47Ser in terms of global fluctuations and long-range-correlated movements, compared to the other simulations (Supplementary Fig. 3). These structural analyses predicted that all variants but p.Leu12Val affect ARF3 GTP/GDP binding and/or the GTPase activity. A more articulated impact on conformational rearrangements mediating binding to effectors was suggested for p.Pro47Ser.

To experimentally validate the predicted consequences on ARF3 function, we examined the protein levels of each mutant in transiently transfected COS-1 cells, basally and after 3 and 6 hour-treatment with the protein synthesis inhibitor, CHX. The immunoblotting analysis documented levels of ARF3$^{D93N}$ and ARF3$^{T32N}$ comparable to the WT protein while showing a slightly reduced level for ARF3$^{P47S}$ and a marked reduction for ARF3$^{K127E}$ and particularly ARF3$^{L12V/D67V}$ (Fig. 2a; Supplementary Fig. 4a), the latter also confirmed in zebrafish embryos (Supplementary Fig. 4b), which was not related to a significant reduction in the mRNA levels (Supplementary Fig. 5a). A similar reduction in expression was confirmed by quantitative imaging analysis in COS-1 cells expressing mCherry-tagged ARF3$^{K127E}$ and ARF3$^{L12V/D67V}$ (Supplementary Fig. 5b, c'). Treatment with MG132 and bafilomycin A1, partially rescued the reduced levels of ARF3$^{K127E}$ and ARF3$^{L12V/D67V}$, indicating an involvement of both the proteasomal pathway and autophagy in degradation (Fig. 2a'). In its active GTP-bound state, ARF3 is able to bind to the Golgi-associated gamma-adaptin ear-containing ARF-binding protein 3 (GGA3) to regulate downstream events controlling *trans*-Golgi function and intracellular trafficking[60]. Thereby, we performed pull-down experiments using the GGA3 protein-binding domain (PBD) on cell lysates from transfected COS-1 cells to compare the relative amounts of GTP-bound fraction of WT and mutant ARF3 proteins. In the same assay, we parallelly assessed ARF3$^{Q71L}$ and ARF3$^{T31N}$ as CA and the DN mutants, respectively[23,41]. Compared to cells expressing ARF3$^{WT}$, those expressing the ARF3$^{K127E}$, ARF3$^{L12V/D67V}$, and ARF3$^{T32N}$ mutants showed a statistically significant reduction of the absolute ARF3 GTP-bound fraction, while a significant increase and a trend in the same direction were documented for ARF3$^{D93N}$ and ARF3$^{P47S}$, respectively (Fig. 2b, c).

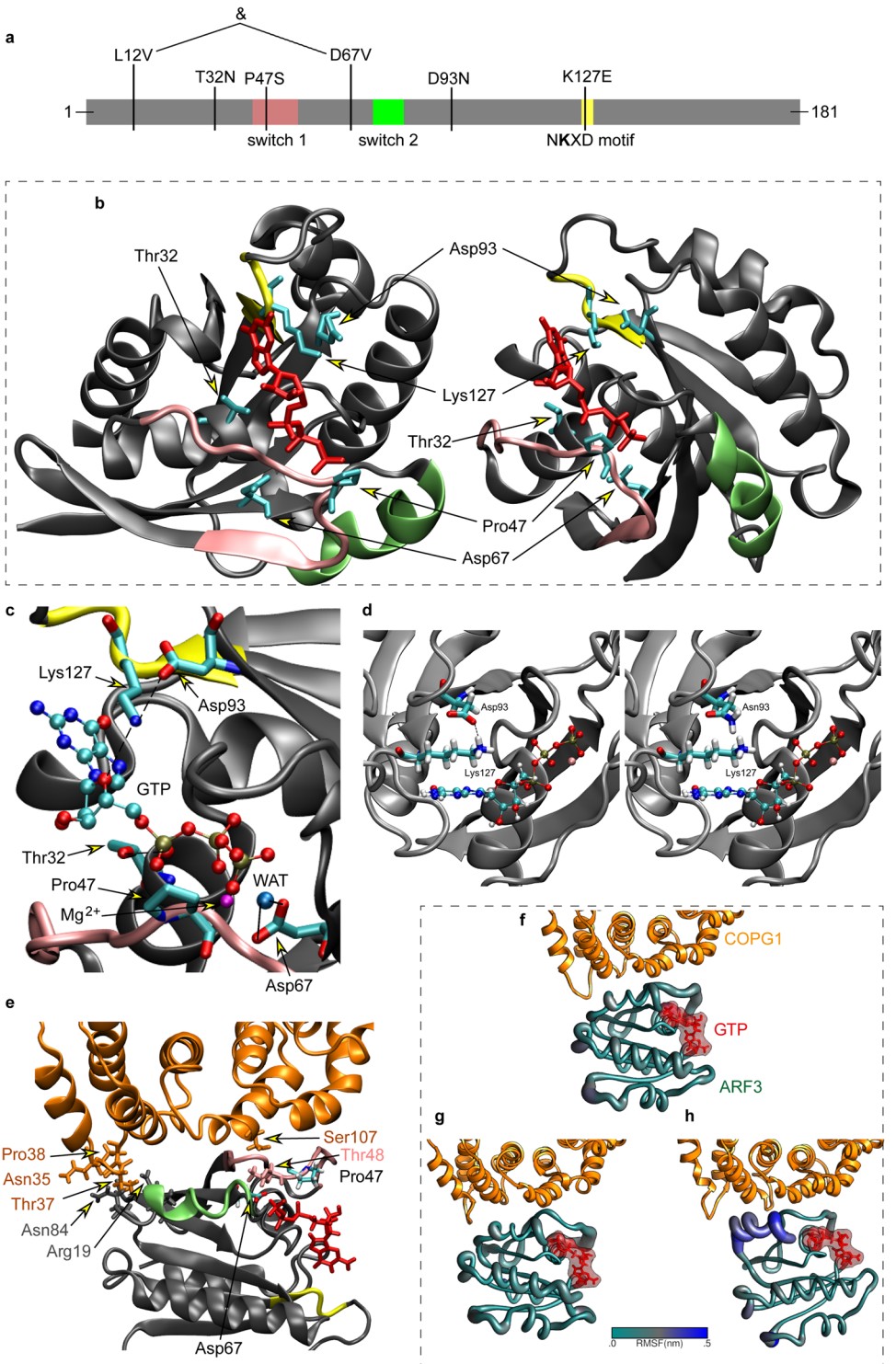

**Fig. 1 | Structural organization of ARF3, location of mutated residues, and molecular dynamics analyses. a** Domain organization of ARF3 excluding the unstructured *C*-terminal tail. Switch 1, switch 2 and the NKXD fingerpoint motif are highlighted in pink, green and yellow, respectively. The variants identified in affected subjects are also reported. **b** 3D structure in two different orientations of GTP-bound ARF3 interacting with the MARTX toxin (PDB 6ii6). Side chains of the ARF3 residues mutated in the affected subjects and GTP are in cyan and red, respectively. Main chain of residues belonging to switch 1, switch 2 and NKXD fingerpoint motif are colored as above. **c** Enlargement of the ARF3 GTP binding pocket with the five mutated residues. The direct hydrogen bond between the N atom in the Lys127 lateral chain and the oxygen atom of the GTP ribose ring is highlighted in dashed line. The Mg$^{2+}$ ion is colored in magenta, while the oxygen atom of the water molecule, mediating the interaction between Asp$^{67}$ and the

manganese ion, is shown in light blue color. The two hydrogen bonds between Asp$^{67}$ and the water molecule are highlighted with dotted lines. **d** Zoom showing the structural organization around residue 93. Left: view of the WT Asp$^{93}$ forming a hydrogen bond with Lys$^{127}$. Right: structure with the p.Asp93Asn mutation and hydrogen bond breaking. The Mg$^{2+}$ ion is colored in magenta. **e** Homology model of GTP-bound ARF3 interacting with the cytosolic coat protein complex COPG1-COPZ1 (PDB: 3TJZ) validated by a 500-ns molecular dynamics (MD) simulation. The region of contact between ARF3 and COPG1 (orange color) is shown in (**e**). **f–h** MD simulations of wild-type (**f**), p.Asp67Val (**g**), and p.Pro47Ser (**h**) ARF3 complexed with COPG1-COPZ1. Residues involved in the contact are shown with their side chain and colored as the respective protein/region. ARF3 backbone is represented with a diameter proportional to its per-residue fluctuations (RMSF).

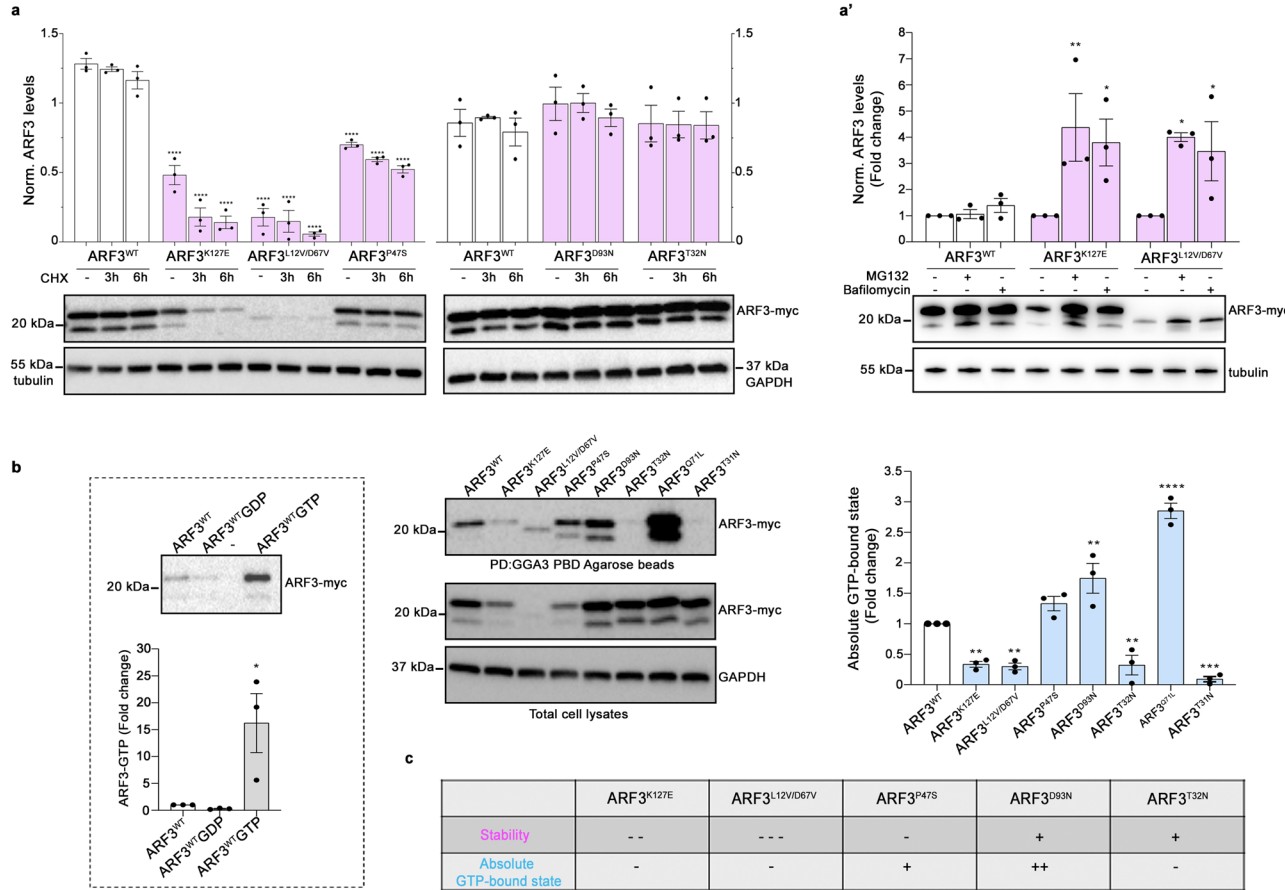

**Fig. 2 | Expression, stability, and GTPase activity of WT and mutant ARF3 proteins in COS-1 cells. a, a′** Western blot analysis showing the protein levels of myc-tagged ARF3[WT] and all the identified mutants in transfected COS-1 cells, basally and after treatment with cycloheximide (CHX) (10 μg/ml) for the indicated time points (**a**), and with MG132 (100 μM), or bafilomycin A1 (200 nM) for six hours (**a′**). **b** Pull-down assay using GGA3-conjugated beads shows ARF3 activation in COS-1 cells transiently transfected with WT or mutant myc-tagged ARF3 expression constructs. Active and total ARF3 levels are monitored using anti-myc antibodies. GAPDH and beta-tubulin are used as loading controls. Pull-down assays of ARF3[WT] transfected cells performed in the presence of an excess of GDP and γGTP are used as negative and positive controls, respectively (b, left panel). Pulldown samples in **b** (left and right), are loaded on different blots and processed parallelly.

Representative blots are shown and data are expressed as mean ± SEM of three independent experiments. Two-way ANOVA followed by Tukey's *post hoc* test (**a**, WT vs. all mutants, ****$p < 0.0001$; **a′**, K127E vs. K127E + MG132 **$p = 0.0052$; K127E vs. K127E + Bafilomycin *$p = 0.0195$; L12V/D67V vs. L12V/D67V + MG132 *$p = 0.0123$; L12V/D67V vs. L12V/D67V + Bafilomycin *$p = 0.0411$), One-way ANOVA followed by Sidak's *post hoc* test (**b** left panel, WT vs. WT + GTP *$p = 0.0197$), One-way ANOVA followed by Dunnett's *post hoc* test (**b**, WT vs. K127E **$p = 0.0088$; WT vs. L12V/D67V **$p = 0.0058$; WT vs. D93N **$p = 0.0035$; WT vs. T32N **$p = 0.0075$; WT vs. Q71L ****$p < 0.0001$; WT vs. T31N ***$p = 0.0006$) are used to assess statistical significance. **c** Summary table of the data obtained relative to the stability and activity of the different ARF3 mutants. Source data are provided as a Source Data file.

Next, by employing a cell-free system and fluorescence polarization we examined the biochemical behavior of a subset of mutants for which we obtained purified proteins. Compared to the WT protein, we observed an increased intrinsic (i.e., GEF-independent) nucleotide exchange for the ARF3[K127E] and ARF3[D93N] mutants. A reduced exchange rate was instead registered for ARF3[L12V/D67V], while ARF3[T32N] did not show significant alterations (Supplementary Fig. 6a, a′). By inspecting the GTP hydrolysis of the purified proteins, we failed to note major changes compared to ARF3[WT] (Supplementary Fig. 6b, b′). Altogether, these data suggest a stabilized GTP-bound conformation and an overall hyperactive behavior for ARF3[D93N] and, to a minor extent, ARF3[P47S], while a DN behavior could be established for the ARF3[T32N] variant. These findings could not unambiguously functionally classify the ARF3[K127E], and ARF3[L12V/D67V] behavior.

### Disease-associated ARF3 mutants differentially impact on Golgi morphology
Next, we assessed the specific Golgi phenotype resulting from the overexpression of the individual mutants in cells. Given the role of ARF proteins in maintaining proper Golgi integrity, organization, and

function[4,20,36,41,61], we performed confocal microscopy analysis of COS-1 cells overexpressing mCherry-tagged WT and ARF3 mutants and labeled for the resident *trans*-Golgi protein Golgin-97[62]. To specifically ascribe the observed phenotype to known dysregulated ARF function and derive possible insights into the mechanism, we directly compared Golgin-97 patterns to that obtained by known CA (p.Q71L) and DN (p.T31N) ARF3 proteins. Four major Golgi morphotypes were identified (Fig. 3a, b). As expected, ARF3[WT]-expressing cells showed diffuse cytoplasmic ARF3 localization. Perinuclear (PN) localization of the protein was also observed, partially co-localizing with Golgin-97 (i.e., GTP-bound ARF3), which showed a canonical, compact ribbon-like morphology (Fig. 3a, upper row, Supplementary Fig. 7). Conversely, only in a minority of cells expressing the DN ARF3[T31N], the *trans*-Golgi was recognizable as a discrete compact entity, while most cells showed partial or total dispersion of Golgin-97 within the cytosol (Fig. 3a, 2nd row, Supplementary Fig. 7), in line with previous reports[41,50], indicating occurrence of massive Golgi disassembly. In striking contrast, cells expressing the CA ARF3[Q71L] protein showed a compact and expanded Golgin-97 staining, likely reflecting an expansion in the size of the *trans*-Golgi (Fig. 3a, 3rd row, Supplementary Fig. 7), as previously

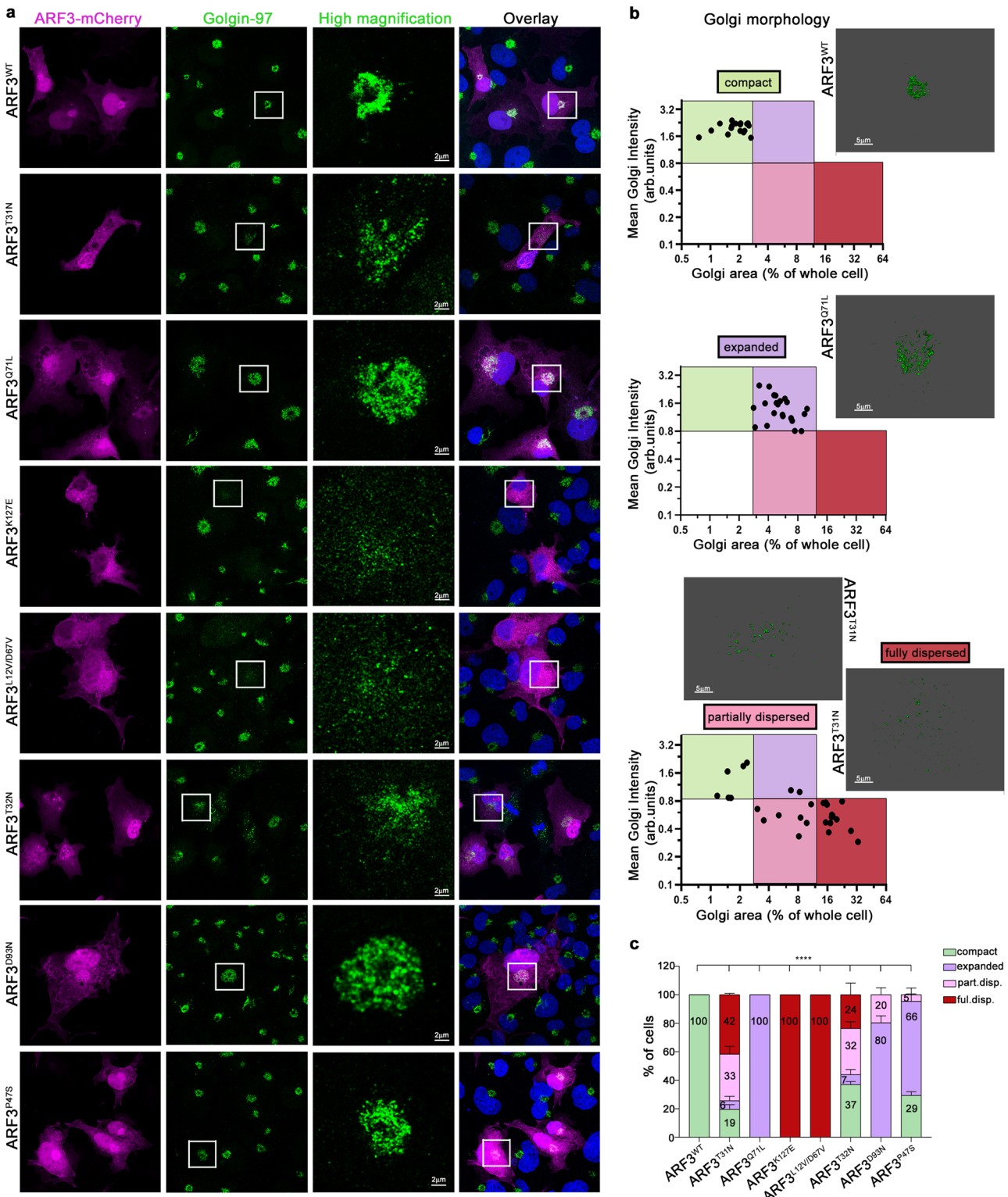

reported[23]. Dot plot representation of the Golgin-97 mean intensity (I) and area (A) in cells expressing ARF3[WT], ARF3[Q71L] and ARF3[T31N] allowed classifying these distinct Golgi structural rearrangements into discrete classes: compact, expanded, partially or totally dispersed (Fig. 3b, Supplementary Fig. 7), providing an unbiased tool for the Golgi morphology assessment. Notably, altered Golgi morphology was not only characterized by an increase in area, but also by a significant decrease in Golgin-97 signal intensity with respect to compact Golgi. The three altered Golgi morphotypes were variably observed in cells expressing the identified *ARF3* variants, and variable severity of the phenotype was

documented. Cells expressing the ARF3[K127E], ARF3[L12V/D67V], and ARF3[D93N] showed Golgi structural alterations in all analyzed cells, while compact Golgi organization was observed in a fraction of cells expressing the ARF3[T32N] and ARF3[P47S] (Fig. 3a, c), which is suggestive of a milder impact of these mutants, in line with the mild clinical features of patients (Supplementary Table 1 and clinical reports). Notably, similar to what was observed for the DN mutant, ARF3[K127E], ARF3[L12V/D67V], and ARF3[T32N] mainly induced variable Golgi dispersal, which was observed in all cells expressing either ARF3[K127E] or ARF3[L12V/D67V] (Fig. 3a, 4th and 5th row, c). Cells expressing ARF3[T32N] were characterized by a milder

**Fig. 3 | ARF3 mutants induce variable Golgi morphological alterations in COS-1 cells. a** Maximum intensity confocal z-projections showing immunostaining against Golgin-97 (*trans*-Golgi marker) (green) performed in fixed COS-1 cells transiently transfected with mCherry-tagged ARF3[WT], ARF3[T31N], and ARF3[Q71L] (DN and CA variants, respectively) or mutants identified (magenta) for 48 h. Composite colocalization images are shown in the right panels with nuclei (DAPI staining) in blue. The images are representative of three independent experiments. Scale bars = 2 μm (high magnification) and 10 μm (all the other images). **b** Golgi means intensity and area define distinct Golgi morphotypes. Dot plots of mean intensity (MI) and area of Golgi in cells transiently transfected with mCherry-tagged ARF3[WT], or ARF3[Q71L] and ARF3[T31N] mutants (up, middle, bottom panels) are shown. Golgi MI and area (% of the whole cell) of cells were measured based on Golgin-97 staining. Whole-cell area was determined using the area covered by mCherry fluorescence as a mask. Representative 3D rendering images of the observed Golgi staining are shown. Cell populations located in different gates are characterized by distinct Golgi morphologies: Compact: A < 2.6 and MI > 1.5 (green gate); expanded Golgi: 2.7 < A < 12 and MI > 0.8, (purple gate); partially dispersed Golgi: 2.7 < A < 12 and MI < 0.8 (pink gate); totally dispersed Golgi: A > 12 and MI < 0.8 (bordeaux gate). **c** Incidence of *trans*-Golgi morphotypes. The bar graph represents the percentage of cells showing compact, expanded, partially or fully dispersed distribution (part.disp and ful.disp.) of Golgi in mCherry-tagged ARF3 transfected cells, based on the classification described above in (**b**). No. of cells = 26 (WT); 22 (Q71L); 29 (T31N); 20 (K127E); 20 (L12V/D67V); 21 (P47S); 28 (D93N) and 27 (T32N). Data are expressed as mean ± SEM of three independent experiments. Two-sided Chi-square's test in a 2 × 2 contingency table (WT vs. all mutants, compact vs. all phenotypes ****p < 0.0001) is used to assess statistical significance. Arb.units = arbitrary units. Source data are provided as a Source Data file.

reorganization of the Golgi with only a minority of cells showing total dispersion (23%, Fig. 3a, 6th row, c). On the other hand, resembling the effect of the CA mutant, ARF3[D93N] and ARF3[P47S] showed a significant increase of cells with expanded Golgi (79% and 67%) (Fig. 3a, 7th and 8th row, c). Of note, a more severe effect was documented for ARF3[D93N], with a fraction of cells also exhibiting partial dispersion, while a compact Golgi was observed in approximately 30% of cells expressing ARF3[P47S] (Fig. 3c). These findings provide evidence of a differential functional impact of the identified *ARF3* variants on Golgi structural morphology (Fig. 3c).

Next, we further investigated Golgi morphology via ultra-structure inspection by performing transmission electron microscopy (TEM) on cells expressing ARF3[K127E] and ARF3[D93N], which showed the "fully dispersed" and "expanded" *trans*-Golgi morphotypes, respectively. While perinuclear Golgi mini-stacks well organized in ribbons were recognizable in cells expressing ARF3[WT] (Supplementary Fig. 8a, a'), those expressing ARF3[K127E] exhibited Golgi fragmentation characterized by an integrity loss of the mini-stacks with numerous diffused vesicles and small cisternae scattered in a wide area (Supplementary Fig. 8b, b'). Cells expressing ARF3[D93N] displayed a different pattern with loss of the typical ultrastructure of Golgi elements and a marked increase in swollen cisternae and diffuse vesiculation, which were confined within the defined area normally occupied by Golgi (Supplementary Fig. 8c, c'). A similar fragmentation pattern had previously been described[4,23,63,64], also for CA ARF and ARF-like mutants[4,23,63,64]. We cannot exclude the occurrence of more complex morphological alterations, whose assessment would require electron tomography analysis or 3D super-resolution microscopy.

To further validate these findings within an organismal context, we next set out to examine the *trans*-Golgi in zebrafish embryos expressing ARF3[K127E] and ARF3[D93N], for which opposite effects were observed in cells. Zebrafish harbors two paralogs, *arf3a* and *arf3b*, which share common ancestry with mammalian *ARF3* and conservation of the amino acids involved in the identified mutations (Supplementary Fig. 2a). *arf3a* and *arf3b* are both expressed during early embryonic development, and *arf3b* shows a higher level of expression after maternal-to-zygotic transition (MZT) throughout gastrulation and somitogenesis (Supplementary Fig. 9), indicating its predominant role during these developmental stages. Next, we overexpressed mRNAs encoding ARF3[WT], ARF3[K127E], and ARF3[D93N], and used specific cellular and subcellular makers to assess *trans*-Golgi morphology in precursor cells of the envelope layer (ELV) in early gastrula (Fig. 4a). We reasoned that the complexity of the physiological Golgi dynamics in vivo and the expected occurrence of fragmented Golgi in proliferating cells[43] might limit our ability to distinguish the specific pathogenic effect of the mutants on Golgi in fish. Thereby, we first verified whether the dispersion of Golgi elements due to the expression of ARF3[K127E] could be observed in early zebrafish embryos. To this aim, we injected mRNAs encoding the mCherry-tagged ARF3[WT] and

ARF3[K127E] in the first batch of siblings, together with a fluorescent membrane marker and EGFP-tagged GalT (galactosidase T[65]), a marker of *trans*-Golgi. By using this marker, we parallelly confirmed ARF3[K127E]-mediated Golgi elements dispersal in a live time-lapse of COS-1 cells (Fig. 4a–c; Supplementary Movie 1) and in alive zebrafish embryos (Fig. 4d, e). In embryos, we observed a diffused distribution of ARF3[WT] partially overlapping EGFP-GalT staining. The latter was structured in ribbon-like elements. Conversely, in ARF3[K127E] expressing fish, EGFP-GalT signal distribution appeared less intense and organized in small and large puncta, some of which also co-localized with ARF3 (indicating Golgi-localization) (Fig. 4d, e, Supplementary Fig. 10a). Next, we compared the EGFP-GalT staining associated with ARF3[K127E] and ARF3[D93N], using the patterns resulting from CA and DN ARF3 mutants as reference. Again, we observed a reduced number of ribbon-like Golgi in cells expressing ARF3[K127E] (<20%) as compared to WT, similar to the DN ARF3-expressing embryos (Supplementary Fig. 10b, c). Despite the changes in Golgi morphology being subtler in fish expressing with ARF3[D93N], the loss of typical Golgi ribbon-like structures was evident. This pattern was accompanied by instances of large EGFP-GalT+ structures (39%), also documented in fish expressing the known CA mutant (Supplementary Fig. 10b, c).

Overall, the collected in vitro and in vivo findings suggest that the identified pathogenic variants in *ARF3* have a variable dominant impact on protein stability, activity, and Golgi morphology. The Golgi morphotype analysis established the presence of different functional classes of disease-causing ARF3 mutants, broadly ascribing to a DN or CA mechanism. Notably, their variable strength appeared to correlate with the severity of clinical features observed in patients.

## ARF3 mutants impair COP-I vesicle formation and cargo recycling in COS-1 cells

Given the known involvement of ARF GTPases in vesicles budding, endosomal transport, and recycling, and considering the observed Golgi phenotypes, we then asked whether and how ARF3 mutants impact the formation and activity of the endolysosomal compartments along the endocytic-recycling pathway. First, we examined the integrity of COP-I vesicles by immunostaining the β-COP subunit of COP-I. In line with the Golgi phenotypes and previous reports on DN and CA ARF3[20,65], cells expressing ARF3[K127E] and ARF3[L12V/D67V] were characterized by a sparse distribution of the signal throughout the cytoplasm, indicating a disassembly of COP-I vesicles. Differently, a large number of cells expressing ARF3[P47S] and ARF3[D93N] showed an expanded β-COP signal, which is indicative of an enlarged COP-I compartment. Cells expressing ARF3[T32N] did not show a clear-cut phenotype, with a minor incidence of cells showing partial/complete disassembly, indicative of a mild effect of the mutant (Fig. 5a, b).

Next, to follow cargos destiny within the endocytic-recycling pathway, we continuously incubated COS-1 cells with fluorescently labeled transferrin (Tfn) at 37 °C for 5 or 30 min to trigger

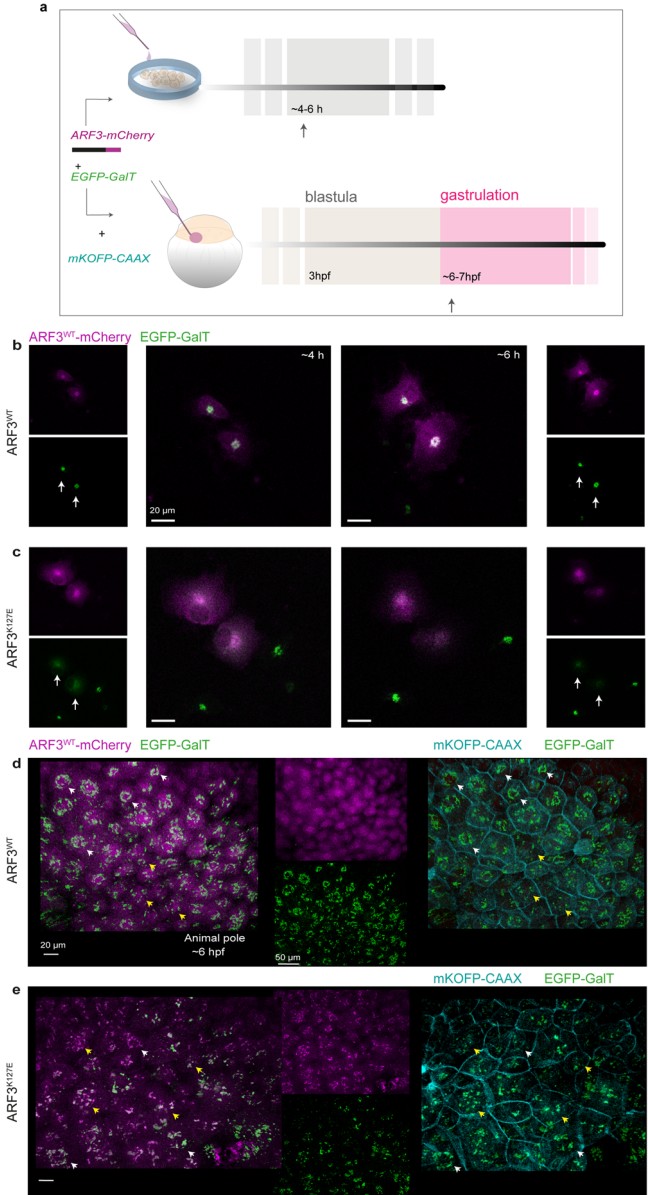

**Fig. 4 | *Trans*-Golgi fragmentation visualized by EGFP-GalT in cells and zebra-fish embryos expressing the mutant mCherry-tagged ARF3^K127E. a** Schematic representation of the experimental setup in both in vitro and in vivo systems. COS-1 cells are transfected with DNA constructs expressing WT and mutant ARF3-mCherry (magenta) and EGFP-GalT (*trans*-Golgi marker, green) and analyzed by live confocal microscopy between 4 and 6 h post-transfection. Zebrafish embryos are injected at 1 cell stage with WT and mutant ARF3-mCherry and EGFP-GalT mRNA. mKOFP-CAAX mRNA is used as a membrane marker (cyan). Animals are analyzed by live confocal microscopy during gastrulation (-6–7 hpf). **b**, **c** Maximum intensity projections of confocal images of a single time-lapse experiment (Supplementary Movie 1) performed in transfected COS-1 cells at 15 min (-4 h post-transfection) and 120 min later (-6 h post-transfection) from the start of the time-lapse experiment. The images show diffused EGFP-GalT signal (*trans*-Golgi fragmentation) in ARF3^K127E over time (white arrows). Scale bar = 20 μm. **d**, **e** 3D image reconstructions from live confocal acquisitions of the animal pole in developing zebrafish embryos expressing ARF3^WT and ARF3^K127E at the mid-gastrulation stage (-6 hpf). White arrowheads indicate a compact *trans*-Golgi morphology surrounding the nucleus ("ribbon") in the EVL cells. Yellow arrow-heads indicate cells showing "*punta*" morphology of the *trans*-Golgi dispersed throughout the cytosol. Scale bars = 20 and 50 μm. The images are representative of embryos from two independent batches. Quantification is shown in Supple-mentary Fig. 10a. Source data are provided as a Source Data file.

internalization and trafficking of the endocytic vesicles containing labeled Tfn/TfnR complex to the endolysosomal compartment[66]. The subcellular distribution of vesicles in these two-time points was assessed by confocal microscopy. In a normal scenario, upon 5 min incubation, Tfn is internalized and found along the endocytic pathway; following longer incubation time, the majority of the Tfn⁺ cargo is expected to have recycled back to the cell surface, such that limited Tfn levels are observed in the PN compartment[67,68]. Upon 5 min incu-bation, the distribution of Tfn appeared nonuniform in ARF3^WT expressing cells, with sparse Tfn⁺ vesicles clusters observed through-out the cell, mostly within the PN space ("semi-clustered"), similar to non-transfected cells (Supplementary Fig. 11a, b). In contrast, a higher fraction of the cells expressing ARF3^K127E and ARF3^L12V/D67V showed Tfn⁺ vesicles enriched within the PN region ("clustered") (Supplementary Fig. 11a, b'). In cells incubated for a longer time with Tfn, both mutants showed an even stronger cargo accumulation (Fig. 6a, b'; Supple-mentary Fig. 11c).

To further assess possible defects in recycling, cells incubated for 30 min with Tfn were stained for Rab5 and Rab11, early (EE), and recycling (RE) endosome markers, respectively[68–71]. The fraction of internalized Tfn co-localizing with Rab5⁺ vesicles was significantly higher in ARF3^K127E and ARF3^L12V/D67V expressing cells compared to the control cells (Fig. 6c, d). A similar trend was observed with respect to Rab11, which was statistically significant for ARF3^L12V/D67V (Supplemen-tary Fig. 12). These findings indicate a dominant behavior of ARF3^K127E and ARF3^L12V/D67V in causing cargo transport delay. None of the other mutants showed altered behavior.

Not recycled Tfn is normally eliminated via the lysosomal pathway[72–75]. To evaluate mis-targeting of the Tfn to lysosomes or overload of the degradative pathway, cells incubated with Tfn for 30 min were stained with the lysosomal marker Lamp2. Compared to cells expressing ARF3^WT, only cells expressing ARF3^D93N showed a sig-nificant increase in the fraction of Tfn colocalized with Lamp2 at the PN. Nevertheless, expression of all mutants except ARF3^P47S was asso-ciated with a significantly increased fraction of Lamp2⁺ vesicles colo-calized with Tfn (Supplementary Fig. 13). Hence, despite the mutation-specific patterns, lysosomes seem to generally increase their Tfn loading in the majority of the mutants.

## Functional validation in zebrafish confirms the pathogenicity and dominant mechanism of action of ARF3 variants

We expanded our in vivo validation by investigating a possible variable impact of all the identified ARF3 mutants on embryogenesis. To this aim, myc-tagged WT and mutant *ARF3* mRNAs were microinjected at one-cell stage zebrafish embryos (Fig. 7a). As anticipated, endogenous *arf3* is detected at early stages of embry-ogenesis (i.e., before MZT) and it accumulates only later starting at late blastula/early gastrula period (Supplementary Fig. 9). In the injected embryos, we profiled the expression timing of myc-tagged protein and determined a subtle expression of both WT and mutant ARF3 before MZT, with a clear increase only later during develop-ment. This pattern mimicked the endogenous *arf3* expression (Sup-plementary Fig. 14).

Injected embryos were sorted based on the expression of GFP-CAAX (used as injection marker), and developmental progression was followed from early time points of gastrulation till 48 hours post fer-tilization (hpf) (long-pec stage) (Fig. 7a, b'), when morphogenesis is nearly completed and sub-compartmentalization of different neural structures can be appreciated[76]. Embryos expressing each of the tested ARF3 mutants showed significant phenotypic alterations compared to siblings expressing ARF3^WT and not injected controls. Compared to normal development (class I), mutant embryos showed variable degrees of survival rate and developmental delay (class II) (Fig. 7c–c", Supplementary Fig. 15a). A statistically significant decrease in the

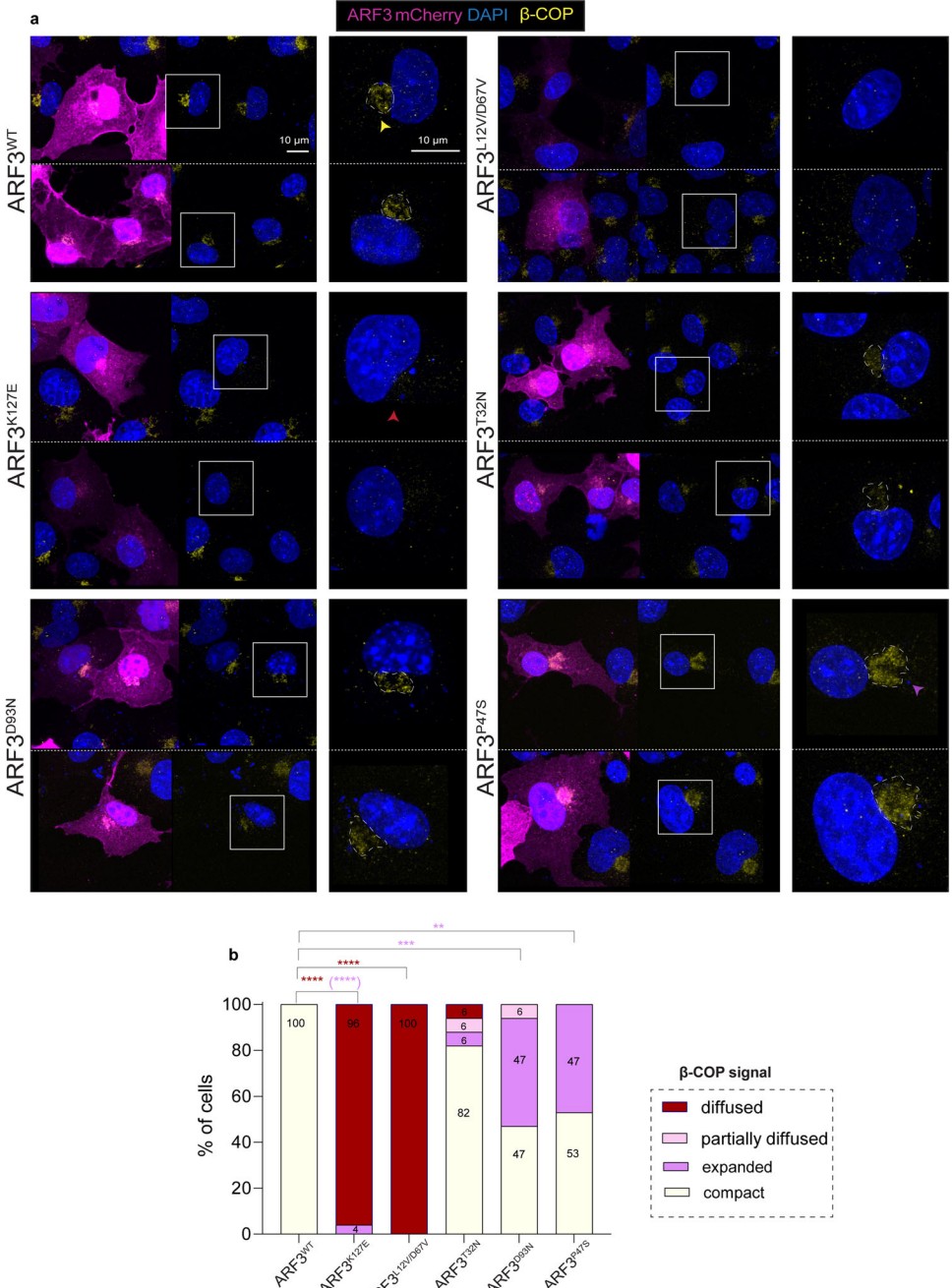

**Fig. 5 | Variable impact of *ARF3* mutations on COP-I vesicle assembly.**
**a** Maximum intensity confocal z-projections of COS-1 cells expressing mCherry-tagged ARF3^WT and all identified mutants and stained for the β-subunit of COP-I. The right panel shows a magnification of β-COP signal. Nuclei are labeled with DAPI (blue). Yellow and red and purple arrowheads: normal, diffused, and expanded β-COP signal, respectively. Scale bar is 10 μm. The images are representative of cells from a single experiment. **b** Quantification of the percentage of WT and ARF3 mutant cells showing different classes of phenotypes as indicated in the legend

(compact and expanded: clustered signal visible at the PN region and with a ratio between COP-I area/nucleus area ≤0.25 or >0.25, respectively). No. of cells = 18 (WT); 26 (K127E, compact vs. diffused ****$p < 0.0001$, compact vs. expanded ****$p < 0.0001$); 16 (L12V/D67V, compact vs. diffused ****$p < 0.0001$); 18 (T32N); 15 (D93N, compact vs. expanded ***$p = 0.0007$) and 15 (P47S, compact vs. expanded **$p = 0.011$). Two-sided Chi-square test in 2 × 2 contingency table is used to assess statistical significance. Source data are provided as a Source Data file.

survival rate of embryos expressing ARF3^K127E was documented (Fig. 7c′), and morphogenesis appeared particularly perturbed both at the level of the head and trunk for a significant fraction of embryos. For the majority of the analyzed mutants, a substantial fraction of embryos (≥25%) showed mild or severe phenotypes (class III and IV, respectively) that were characterized by reduced head size, with/without microphthalmia, and/or mild shortening and lateral bending of body axis (class III), or considerably reduced head (microcephaly or

anencephaly) and eye size, with marked reduction of the trunk, defective body elongation and severe lateral bending (kinked notochord, class IV) (Fig. 7c–c″, Supplementary Fig. 15a).

To validate the mechanism of action and further test the dominant behavior of the *ARF3* variants in vivo, we directly compared the observed ARF3 overexpression phenotype with that obtained by downregulating endogenous *arf3* via translation blocking morpholino (MO) approach, targeting both *arf3a* and *arf3b* maternal and zygotic

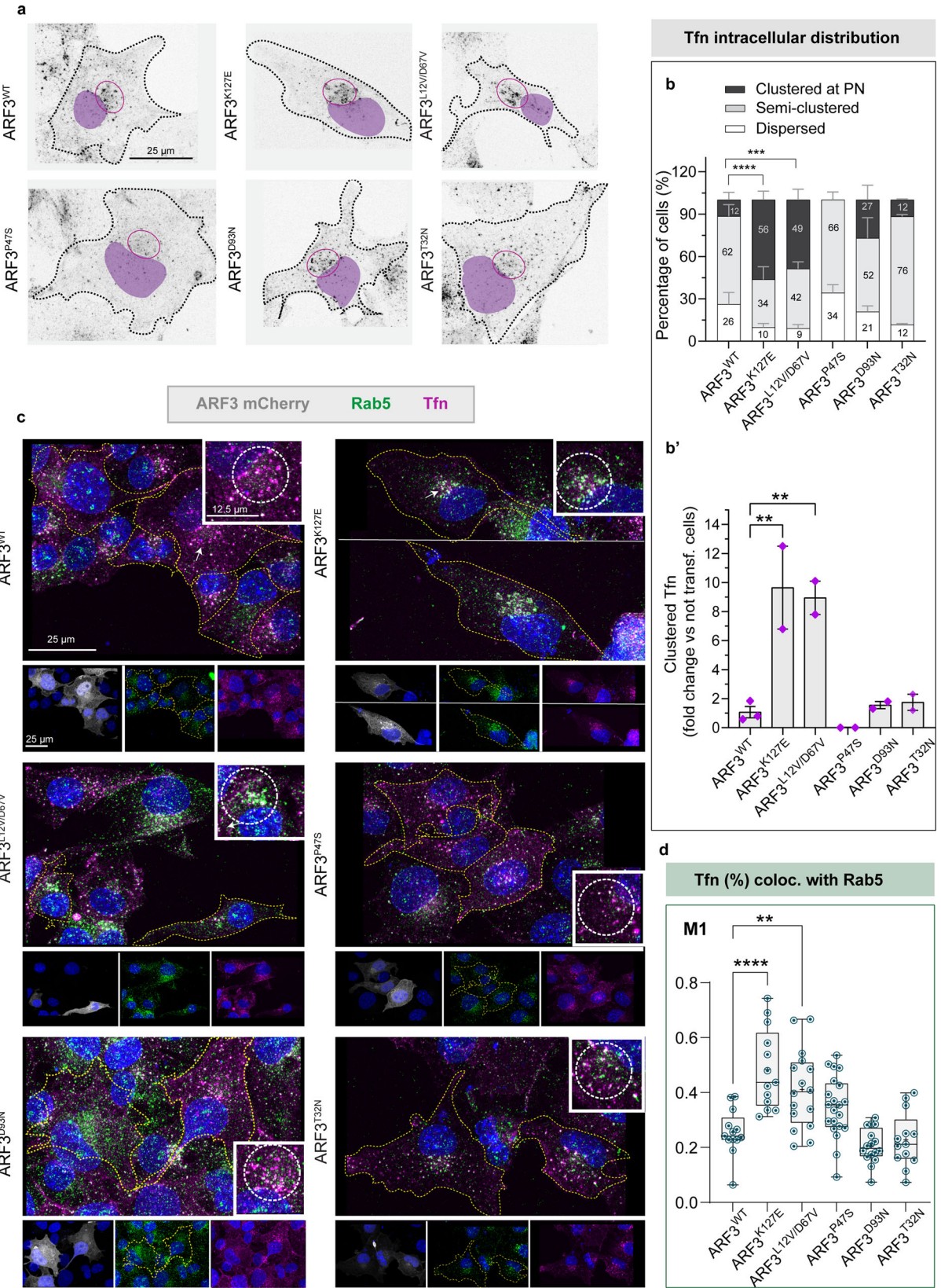

translation. At 24 hpf, fish injected with MO against *arf3a* and *arf3b* showed a subtler phenotype, with a prevalence of class II phenotype (developmental delay), and only a small percentage (<8%) of animals showing the characteristic body curvatures observed in fish expressing ARF3 mutants (Fig. 7d, d'). These defects increased only later in development but remained <20% on average (Fig. 7d"). Moreover,

contrary to fish expressing ARF3 mutants, we did not observe severely affected embryos among the *arf3* MO-injected embryos (Fig. 7d–d") nor significant death (Supplementary Fig. 15b). Notably, the incidence of the observed phenotype increased with increasing MO doses (Fig. 7d–d"), and by 48 hpf a significant rescue of the phenotype could be observed when co-injecting mRNA encoding ARF3^{WT} (Fig. 7e–e"),

**Fig. 6 | Internalized Tfn accumulates in the perinuclear region and in Rab5 + endosomes of COS-1 cells expressing ARF3$^{K127E}$ and ARF3$^{L12V/D67V}$. a** Maximum intensity confocal z-projections showing the distribution of Tfn-488 (black dots) upon 30 min of incubation in COS-1 cells expressing mCherry-tagged ARF3$^{WT}$ and all identified mutants. Red circle indicates Tfn signal at the perinuclear region (PN). Outlines (black in a and yellow in c) depict the boundaries of representative transfected cells. The black and white images are rendered by inverting the original LUT in Fiji and nuclei are pseudo-colored (purple) in the images. The images are representative of two independent experiments. **b–b'** Incidence of cells showing "clustered", "semi-clustered" or "dispersed" Tfn staining (**b**) and the ratio of the cells (%) showing "clustered" Tfn phenotype normalized by not-transfected cells (NT) with the same phenotype (**b'**, internal control). No. of cells = 42 (WT); 22 (K127E, ****$p < 0.0001$); 33 (L12V/D67V, ***$p = 0.0002$); 31 (P47S); 26 (D93N); and 25 (T32N). Data are expressed as mean ± SEM (**b**, **b'**) of three (WT) and two (all the other mutants) independent experiments. **c** Maximum intensity confocal z-projections showing COS-1 cells expressing ARF3$^{WT}$ and all identified mutants,

incubated with Tfn-488 for 30 min followed by immunostaining against Rab5 (marker of early endosomes). For all the panels single channels (ARF3mCherry: gray, Tfn-488: magenta, Rab5: green), the merge showing the co-localization between Tfn and Rab5 are shown. The insets in the white square show a zoom on the PN co-localization signal. Nuclei are stained with DAPI. The images are representative of cells from a single experiment. **d** Colocalization analysis showing the spatial co-occurrence of Tfn and Rab5+ signals at the PN region in the z-stacks analyzed, no. of cells = 13 (WT; K127E ****$p < 0.0001$; T32N), 16 (L12V/D67V, **$p = 0.0016$; D93N), 21 (P47S). The fraction (%) of Tfn$^+$ signal co-localized with Rab5$^+$ vesicles at the PN (thresholded Mander's coefficient M1) is reported as box-and-whisker with median (middle line), 25th–75th percentiles (box), and min–max values (whiskers) of a single experiment. All the data points and the mean ("+") are also shown. Two-sided Chi-square's test in a 2 × 2 contingency table (semi-clustered and dispersed vs. clustered, **b**), One-way ANOVA followed by Dunnett's multiple comparison *post hoc* test (**b'**, **d**) are used to assess the statistical significance. Source data are provided as a Source Data file.

demonstrating the specificity of the phenotype in relation to *arf3* downregulation.

The MO approach had previously been used as a tool to test the genetic mechanism of action in vivo, assuming that downregulation of endogenous protein expression alleviates the phenotypes associated with CA mutants but exacerbates the phenotype of DN mutants[77]. Therefore, we performed a set of experiments in which each of the pathogenic *ARF3* alleles was co-injected with *arf3a/b* MO (+MO). When we statistically assessed the incidence of phenotypes in "+MO" conditions against those observed by injecting solely mutant *ARF3* mRNA (−MO), a significant worsening of the most severe traits was documented for ARF3$^{K127E}$ (class IV), ARF3$^{L12V/D67V}$ and ARF3$^{T32N}$ (both for class III). On the other hand, we observed a significant alleviation of the phenotype (class III) in embryos expressing ARF3$^{D93N}$ and co-injected with *arf3a/b* MO. We did not observe any substantial change in the phenotype severity for ARF3$^{P47S}$ (Fig. 7f). A ratio between the percentage of embryos showing the most severe traits, including class IV and V (deceased fish) with or without MO confirmed the trend for most of the mutants (Fig. 7f). Altogether, these data provided in vivo evidence of a dominant mechanism of the identified disease-causing variants, clearly distinguishable from the *arf3* loss-of-function effect. Moreover, corroborating the in vitro results, these findings support a DN mechanism for p.K127E, p.L12V/p.D67V, and p.T32N, and a CA behavior for p.D93N.

## Zebrafish embryos expressing ARF3 mutants recapitulate the variable disease severity

To explore further the consequences of *ARF3* mutations on neurodevelopment, we more accurately characterized zebrafish head and brain phenotype. At 24 and 48 hpf, compared to not injected controls and siblings expressing ARF3$^{WT}$, we registered a significant reduction of the head area for p.K127E and p.L12V/D67V, with the most severe cases lacking the frontal part of the brain and eyes (Figs. 7b'; 8a, b). Phenotypic assessment at later stages (4.5 days post fertilization, dpf) documented the appearance of microcephaly also in embryos expressing ARF3$^{D93N}$, while none of the other mutants showed significant changes (Fig. 8b').

These in vivo measurements resembled the variable clinical traits reported in patients, with only Subjects 1 (p.K127E) and Subject 2 (p.L12V/p.D67V) showing severe microcephaly at birth, and Subject 4 (p.D93N) displaying post-natal microcephaly (Fig. 8b", Supplementary Fig. 1, Supplementary Table 1 and clinical reports). Next, taking advantage of our live whole-brain/embryos samples, we examined the anterior brain volume in fish exhibiting early- and late-onset microcephaly as well as in embryos expressing ARF3$^{P47S}$, which was associated with a mild reduction within the developing forebrain in Subject 3 (Supplementary Fig. 1, Supplementary Table 1 and clinical reports). To this aim, we employed the *NBT:dsRed*

transgenic line, labeling differentiated neurons. Volumetric reconstructions from live confocal z-stack acquisitions confirmed a significant reduction of the brain volume for ARF3$^{K127E}$ and ARF3$^{L12V/D67V}$ (Fig. 8c, c').

Additional volumetric measurements obtained from fixed specimens at 48 hpf by labeling mature axonal and neuronal structures confirmed the observed brain volume reduction for ARF3$^{K127E}$ (Supplementary Fig. 16). Of note, despite head measurements documented only a delayed effect of ARF3$^{D93N}$ and did not show significant changes for ARF3$^{P47S}$, the volumetric analysis of the anterior brain at 48 hpf was able to capture a significant reduction of brain mass for both mutants (Fig. 8c, c').

Defective formation of the forebrain commissural fibers of the corpus callosum (CC) is a common feature of all patients. No evolutionary-related structure has been described in teleost fish; nevertheless, the anterior commissure (AC) is the major white matter structure within the developing zebrafish telencephalon. Similar to the CC, the AC in zebrafish consists of thick axonal bundles connecting the two hemispheres of the telencephalic forebrain[78]. To expand our brain phenotyping, we therefore assessed AC formation in 48 hpf fish injected with WT and mutant *ARF3* mRNA by using staining against anti-acetylated tubulin to visualize the axonal bundles. A significant reduction in the width of the AC lateral bundles was observed for all the ARF3 mutants. A stronger effect was recorded for ARF3$^{K127E}$, ARF3$^{L12V/D67V}$, and ARF3$^{D93N}$ when the width of the entire AC was considered (Supplementary Fig. 17).

Altogether, the morphometric parameters measured in vivo are consistent with the variable degree of impaired brain development as a distinctive feature of the disease and support the occurrence of telencephalic white matter defects as a common trait of this new Golgipathy. Our findings further document a severe effect on brain development for the p.K127E and p.L12V/D67V ARF3 substitutions in zebrafish, which captures the severity of phenotype observed in Subjects 1 and 2.

## Aberrant ARF3 function induces proliferation and cell cycle defects within the anterior brain

Cortical malformations resulting in microcephaly are often caused by aberrant neurogenesis underlying altered proliferation and cell cycle progression, which ultimately lead to premature stem cell death[79–81]. To test this hypothesis and probe into the mechanism causing reduced brain volume in mutant embryos, we examined the proliferative status and quantified cell death. By performing whole-brain immunohistochemistry using anti-proliferating cell nuclear antigen (PCNA) and anti-phospho-histone 3 (pH3) antibodies, we queried the proliferative and mitotic ability of precursor cells at 48 hpf within the forebrain proliferative zone (pz), which is clearly discernible from ventral confocal images (Fig. 9a, b). The number of pH3$^+$ cells within this region

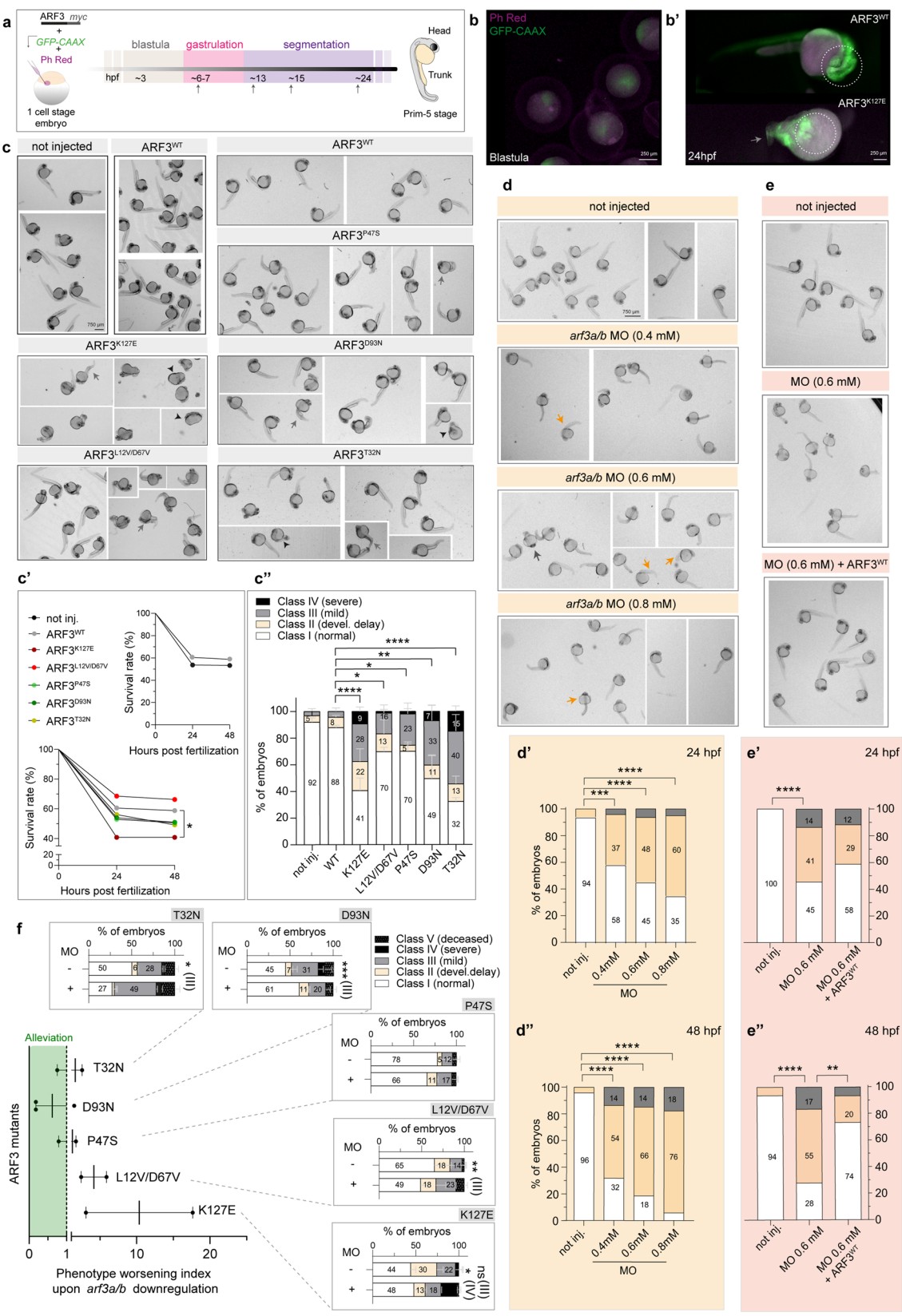

was significantly reduced in embryos expressing ARF3^K127E; a similar trend was observed for ARF3^L12V/D67V and ARF3^D93N expressing fish (Fig. 9b, c). In addition, we detected changes in the overall distribution of proliferative cells within the dorsal brain domain, which ectopically invaded the midbrain territory normally populated by differentiated neurons and nerve bundles (tectal neuropil)[82,83], with a stronger effect

observed for ARF3^K127E and ARF3^D93N (Supplementary Fig. 18a–h). This pattern indicates the occurrence of a complex impairment of the developmental processes within the anterior brain. No major changes in the total number of pH3+ cells were observed except for an increase in the cerebellum in ARF3^D93N expressing fish (Supplementary Fig. 18i–k).

**Fig. 7 | Expression of mutant ARF3 and downregulation of endogenous *arf3* induce distinct phenotypes in zebrafish. a** Experimental strategy in zebrafish models. Injected with WT and mutant ARF3-encoding mRNAs at 1 cell stage and phenotyped at different stages. **b–b′** Images and close-ups of ARF3^WT and ARF3^K127E expressing embryos at 24 hpf, co-injected with GFP-CAAX-encoding mRNA and Phenol-Red (dashed circle depicts cephalic region). **c** Bright-field images of embryos expressing WT and mutant ARF3. The images are representative of embryos from two (**b**, **b′**) and five (**c**) independent batches. **c′** Embryo survival, no. of embryos = 246, 114, 53, 86, 85, 161, 114 (not injected, WT, K127E, *p = 0.03, L12V/D67V, P47S, D93N, T32N) from pooled batches. **c″** Incidence of gross phenotypes at 24 hpf (classes: I, II = yellow arrows, III, IV = gray and black arrows, respectively), no. of embryos = 132 (not injected); 69 (WT); 21 (K127E, ****p < 0.0001); 58 (L12V/D67V, *p = 0.02); 45 (P47S, *p = 0.03); 86 (D93N, **p = 0.0018); 64 (T32N, ****p < 0.0001). Data are expressed as mean ± SEM of four (not injected, WT), three (D93N), and two (K127E, L12V/D67V, P47S, T32N) independent batches. **d–d″** Bright-field images (**d**) and phenotype incidence at 24 and 48 hpf of *arf3a/arf3b* MO-injected embryos (**d′**, **d″**). Respectively, in **d′** and **d″** no. of embryos = 50, 48 (not injected); 25, 22 (MO 0.4 mM, ***p = 0.0002, ****p < 0.0001); 31, 27 (MO 0.6 mM **** < 0.0001); 21, 17 (MO 0.8 mM, ****p < 0.0001) of one batch. **e–e″** Bright-field images (**e**) and phenotype incidence at 24 and 48 hpf (**e′**, **e″**) of *arf3a/arf3b* MO-injected embryos (0.6 mM)

−/+ARF3^WT-encoding mRNA. The images in **d** and **e** are representative of embryos of one batch. Respectively in **e′** and **e″**, no. of embryos = 47 (not injected); 22, 18 (MO 0.6 mM, ****p < 0.0001); 17,15 (MO 0.6 mM + ARF3, **p = 0.0091 in **e″**) of one batch. **f** Phenotype worsening index at 48 hpf (fold-change) for ARF3 mutants (severe + deceased) compared to controls (co-injected with *arf3* MO). In the scatter plot the values < =0 (green) are found in the "alleviation window" depicted with green shading. Dots represent the "worsening index" for each experiment, calculated by dividing the percentage of severely diseased fish (class IV–V) in "MO+" condition by the same percentage obtained in "MO−"condition of two (K127E, L12V/D67V, P47S, T32N) or three (D93N) independent batches. The mean effect of MO for each mutation is also shown as bar graph. No. of embryos = 21 and 36 (K127E − and + MO *p = 0.0307); 58 and 47 (L12V/D67V − and + MO, **p = 0.0068); 45 and 54 (P47S − and +MO); 86 (D93N+ and − MO, ***p = 0.0004); 64 and 35 (T32N − and + MO, *p = 0.0370). Data in the bar graphs are expressed as a mean ± SEM of two independent batches. Survival is assessed by Log-Rank (Mantel−Cox) test (**c′**), Two-sided Chi-square's test in a 2 × 2 contingency table (class II, III and IV vs. I in **c″**, **d′**, **d″**, **e′**, **e″**) or Two-sided One sample *t*-test (class III/ IV/V vs. I in **f**) testing null hypothesis H₀, represented by the expected mean value of the control population, are used to assess statistical significance. Source data are provided as a Source Data file.

Next, by assessing the known chromatin morphology through the inspection of pH3 staining appearance, we profiled cells with respect to the cell cycle stage[84]. Compared to controls, significant alterations in the relative proportion of mitotic cells between early phases (prophase/prometaphase), metaphase, or late phases (anaphase/telophase) were observed in embryos expressing the ARF3 mutants with the exception of ARF3^P47S. Specifically, precursor cells scored a higher percentage of pH3^+ cells in prophase/prometaphase at the expense of later cell cycle stages, suggesting a delay or arrest in early mitosis (Fig. 9b, d, e).

Precursor cells failing to progress through the cell cycle are normally targeted to apoptosis via mitotic surveillance systems[85]. Similar mechanisms activated during aberrant neurogenesis deplete the pool of stem cells available for neurogenesis and brain growth and result in microcephaly[86,87]. To test this possibility, we next assessed the cell death rate within the forebrain of our fish mutants by live embryo staining with acridine orange (AO). The analysis showed a significant increase of AO^+ spots (i.e., dying cells and/or apoptotic bodies) in ARF3^K127E and ARF3^L12V/D67V expressing fish (Fig. 9f, g, upper graph). This finding is in line with the clinical and functional in vivo data reporting p.K127E and p.L12V/D67V as the ARF3 amino acid substitutions associated with the most severe phenotype characterized by early-onset microcephaly in patients and severe head area reduction in fish, respectively. Increased cell death was also recorded for ARF3^P47S and ARF3^D93N expressing fish when a larger area of the forebrain including the eyes was examined (Fig. 9g, lower graph).

Last, given the importance of Golgi for the establishment and the dynamics of mitotic spindles in dividing precursors[88–90], we asked whether spindle aberrations could at least partially explain the cell cycle alterations observed in fish, as previously reported in a number of cortical malformations with microcephaly[91–93]. We took advantage of the transgenic line *Tg(XlEef1a1:dclk2DeltaK-GFP)* marking microtubules in early embryos and investigated metaphase spindles morphology within the anterior ventral brain in live embryos expressing ARF3^WT, ARF3^K127E and ARF3^D93N, causing early- vs. late-onset microcephaly. Compared to controls, aberrantly elongated spindle morphology was recorded for both mutants (Supplementary Fig. 19), indicating a common effect on spindle microtubule organization, likely explaining the similar impact on the cell cycle.

Collectively these data suggest a complex impact of different ARF3 mutants on neurogenesis and point to an impaired balance between precursors' cell mitosis and cell death as a mechanism contributing to the observed neurodevelopmental phenotypes.

## ARF3 mutants variably impact PCP-dependent axes formation in early zebrafish development

We next detailed the morphological defects and developmental processes implicated in the observed body curvature. We focused on the notochord, which supports the body elongation along the anterior to the posterior axis (AP) and spine formation[94], and whose altered development has been associated with CA ARF1 in fish[47]. We documented the occurrence of multiple notochord curvatures of variable degrees in animals expressing each of the ARF3 mutants except ARF3^P47S (Fig. 10a–c). Quantification of the degree of bending (180°: normal; 179° ≥ angle ≥ 110°: mild; angle ≤ 109°: severe) showed a similar incidence of mild and severe bending (in >90% of embryos) in fish expressing ARF3^K127E and ARF3^L12V/D67V (Fig. 10b, c), in line with the overall severity of the skeletal phenotype characterizing subjects 1 and 2 (Supplementary Fig. 1, Supplementary Table 1 and clinical reports). Overall, all mutants except ARF3^P47S showed a significantly higher number of notochord curvatures (Fig. 10c′).

We further examined the underlying causes of the perturbed body trunk and notochord morphogenesis by tracing back axes establishment in development. First, we examined patterning and morphogenesis in animals in their segmentation stage (15 hpf). During this period, the embryo AP and mediolateral (ML) axes are already established and somitogenesis occurs. mRNA levels of *Krox20* and *MyoD* (markers of the hindbrain rhombomeres in the anterior cephalic domain and of developing somites from the paraxial mesoderm, respectively) were assessed in whole-mount embryos by in situ hybridization (ISH). While proper patterning of the cephalic region and paraxial mesoderm was in place, we observed a variable perturbation of the AP and ML axes in the ARF3 mutants (Fig. 10d, e). Embryos expressing ARF3 mutants showed a clear shortening of the AP length compared to WT (Fig. 10e, f). Consistently, the number of somites in mutant embryos was also reduced (Supplementary Fig. 20). Expansion of the paraxial tissue in the ML axis was also apparent for some mutants (Fig. 10e). The data pointed to a defective convergence-extension (CE) process, which was evident for all mutants except ARF3^P47S when the CE index (i.e., the ratio between the AP extend and ML extend of the anterior somites) was assessed (Fig. 10e, g). For both AP and ML axes defects, severely affected embryos were more prevalent among those expressing ARF3^K127E.

Lastly, benefiting from the transparent and fast zebrafish development, we assessed the time occurrence of axes defects linked to gastrulation (and thereby CE) perturbation by investigating earlier stages (Supplementary Fig. 21a). Already between 10 and 13 hpf, when segmentation has just started, brain thickenings and tail bud are visible at the very anterior and posterior end of the embryo, respectively, as a

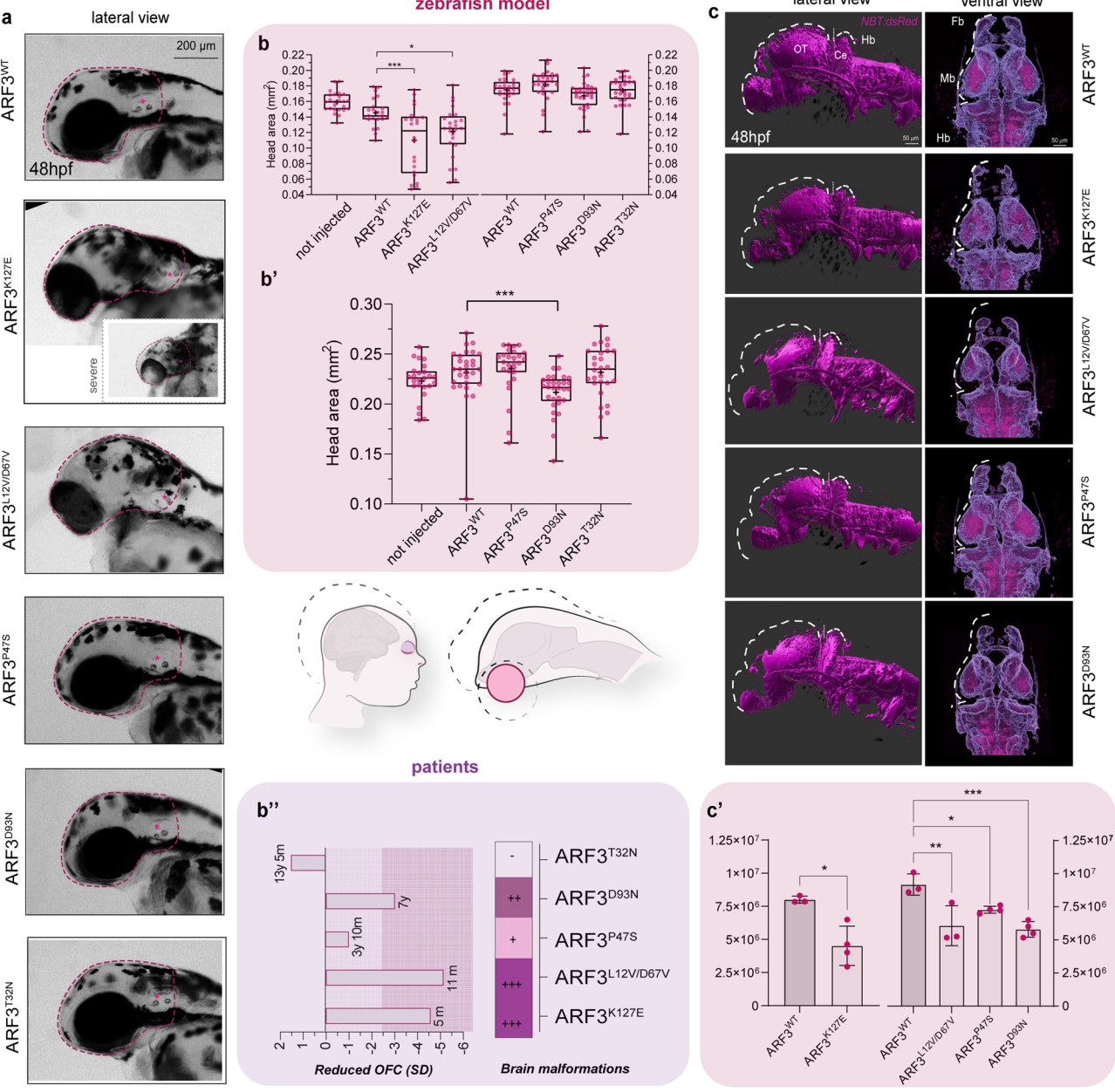

**Fig. 8 | Occurrence of microcephaly and reduced brain volume in zebrafish expressing a subset of ARF3 mutants. a** Bright-field images of the head (purple dashed line) in fish expressing WT and mutant ARF3 at 48 hpf (the inset for ARF3^K127E documents a severe case). The images are representative of embryos from two independent batches. **b–b'** Head area quantification at 48 hpf (**b**) and 4.5 dpf (**b'**). In **b**, set 1: no. of embryos = 25 (not injected), 23 (WT); 22 (K127E, ***p = 0.0002) and 25 (L12V/D67V, *p = 0.0106) of one batch; set 2: no. of embryos = 29 (WT); 28 (P47S); 30 (D93N); 29 (T32N). in **b'**, no. of embryos = 30 (not injected); 30 (WT); 27 (P47S); 30 (D93N, ***p = 0.0006) and 28 (T32N) of one batch. Data are expressed as box-and-whisker with median (middle line), 25th–75th percentiles (box), and min–max values (whiskers). All the data points and the mean ("+") are also shown. **b''** Schematics of the brain volume reduction in human patients harboring a subset of ARF3 mutants and in zebrafish models generated in this study. The human brain in the illustration was created with BioRender.com and modified using Illustrator (Adobe). A summary of OFC and brain malformations data from patients in this study are depicted below (no sign of brain malformation (–), mild (+), moderate (+

+), and severe (+++) malformations). (**c**) Volumetric reconstructions (c) and anterior brain volume (white dashed line) from live confocal acquisitions from whole brains of 48hpf *Tg(NBT:dsRed)* fish injected with mRNA encoding WT and ARF3 mutants. The images are representative of embryos from two independent batches for WT, K127E, and L12V/D67V and from one batch for the other mutants. OT: optic tectum, Ce: cerebellum, Fb: forebrain, Mb: midbrain, Hb: hindbrain. **c'** Quantification of the brain volume. Set 1: no. of embryos = 4 (WT); 4 (K127E, *p = 0.0163) from one batch; set 2: no. of embryos = 3 (WT); 3 (L12V/D67V, **p = 0.0029); 4 (P47S, *p = 0.0350 and D93N, ***p = 0.0010) of one batch. Data are expressed as mean ± SEM. Different datasets for the same measurement are shown in adjacent plots with the internal WT control for each set, not injected controls between batches are not significantly different. One-way ANOVA followed by Dunnett's multiple comparison *post hoc* test (**b**, left panel; **c'**, right panel), Krustal–Wallis followed by Dunn's multiple comparison *post hoc* test (b, right panel; **b'**), unpaired *t*-test with Welch's correction (**c'**, left panel) are used to assess statistical significance. Source data are provided as a Source data file.

result of correctly orchestrated gastrulation movements[76]. By measuring the angle between the developed cephalic and caudal structures at this stage, we documented a reduced embryo elongation (likely due to aberrant/delay gastrulation) for ARF3^K127E and ARF3^D93N

(Supplementary Fig. 21b, c), indicating an early impact of these mutants on axes formation.

Perturbed cell movements were further confirmed in live embryos expressing ARF3^K127E, which exhibited the strongest effect in terms of

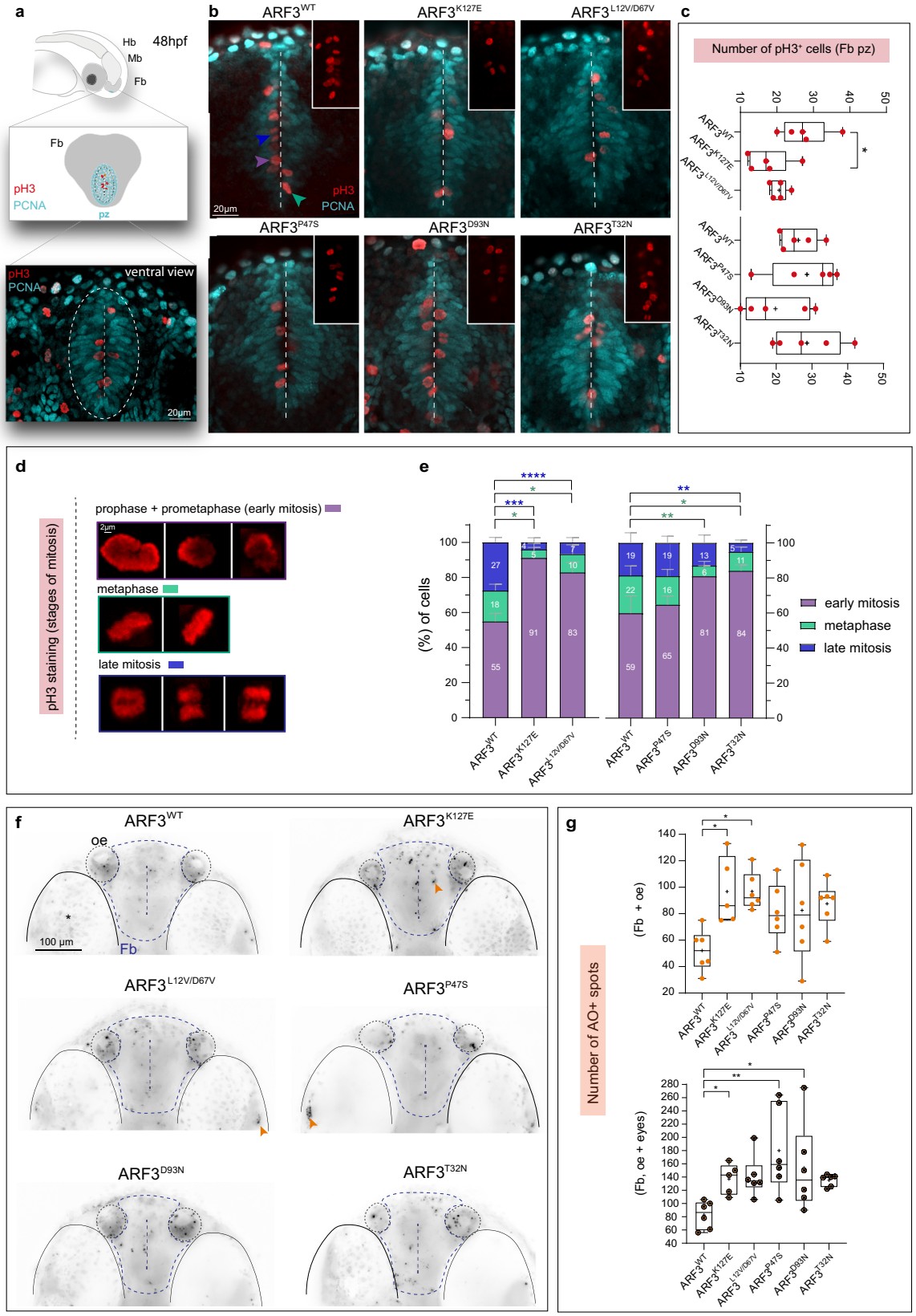

axis elongation (AP, ML). Early embryos (6–7 hpf) showed reduced epiboly and impaired gastrulation, which ultimately resulted in defective head and tail bud formation (Supplementary Fig. 21d, e). Of note, cells expressing the mutant appeared mostly round, with a reduced number of protrusions, with respect to cells expressing the WT protein (Supplementary Fig. 21d–f), suggesting the occurrence of

altered polarity establishment and cytoskeletal organization as an early molecular event, in line with the emerging roles of Golgi in instructing cell polarity[95].

These in vivo findings demonstrate impairment of axes formation of variable degree as a common trait of the mutants causing skeletal deformities in patients, broadly recapitulating the severity of the

**Fig. 9 | Increased number of cells in early mitosis and cell death within the developing forebrain of zebrafish expressing a subset of ARF3 mutants.**
**a** Schematics of the forebrain (Fb, gray) proliferative zone (pz, cyan) and a confocal scan of the ventral Fb in zebrafish expressing ARF3^WT (dashed white circle) showing proliferative and mitotic cells (PCNA and pH3 staining in red and cyan, respectively). Mb:midbrain, Hb:hindbrain. **b** Maximum intensity z-projections from a subset of confocal sections showing pH3+ mitotic cells within the ventral Fb (vFb) in zebrafish expressing WT and mutant ARF3. The images are representative of embryos from one batch. Dashed white line indicates the Fb ventricle, insets show zooms on pH3+ cells. Arrowheads indicate examples of pH3+ cells in different stages of mitosis: early mitosis (purple); metaphase (green) and late mitosis (blue). **c** Quantification of the total number of pH3⁺ cells in pz, no. of embryos = 5 (WT and K127E, *p = 0.0217) of one batch. Data are expressed as box-and-whisker with median (middle line), 25th–75th percentiles (box), and min-max values (whiskers). All the data points and the mean ("+") are also shown. **d, e** Incidence of pH3⁺ cells in the different mitosis stages. No. of cells = 111 and 110 (WT); 61 (K127E, early mitosis vs. metaphase *p = 0.0168, early vs. late mitosis ***p = 0.0003); 88 (L12V/D67V, early mitosis vs. metaphase *p = 0.0115, early vs. late mitosis ****p < 0.0001); 75 (D93N,

early mitosis vs. metaphase **p = 0.0028); 110 (P47S); 109 cells (T32N, early mitosis vs. metaphase *p = 0.0267, early vs. late mitosis **p = 0.0069) from five embryos of one batch. Data are expressed as mean ± SEM. In **c** and **e** different datasets for the same measurement are shown in adjacent plots with the internal WT control for each set. **f** Maximum intensity z-projections of the ventral brain stained with the acridine orange (AO). The black and white images are rendered by inverting the original LUT in Fiji. The images are representative of embryos from two independent batches. Orange arrowheads indicate specific staining. vFb pz, ventricle, eyes, and olfactory epithelium (OE) are outlined for morphological guidance, *indicates eyes with pigmentation background. **g** Quantification of the number of AO + spots. No. of embryos = 6 (WT); 5 (K127E, *p = 0.0163, *p = 0.0461 for upper and lower graphs, respectively); 6 (L12V/D67V, *p = 0.0109); 6 (P47S, **p = 0.0017 and D93N, *p = 0.0387) of one batch. Data are expressed as box-and-whisker with median (middle line), 25th–75th percentiles (box), and min–max values (whiskers). All the data points and the mean ("+") are also shown. Two-sided Chi-square's test in a 2 × 2 contingency table (**e**), One-way ANOVA followed by Dunnett's (**c** and **g**, upper graph) or Kruskal–Wallis followed by Dunn's (**g**, lower graph) post hoc tests are used to assess statistical significance. Source dData are provided as a source data file.

clinical phenotype, and tracing back the mechanism to a compromised PCP-dependent CE cell movement for the severe cases.

## Discussion

Controlling organelle stability, targeted trafficking of proteins and lipids, and signaling, the highly conserved ARF GTPases contribute to cell polarity, division, and migration ultimately instructing development[1,46,47]. Here we identify de novo missense ARF3 variants as the molecular event underlying a clinically variable neurodevelopmental disorder characterized by DD/ID and variable CNS defects as common features. Microcephaly and progressive cerebral atrophy occurred in most affected individuals, while epilepsy and skeletal abnormalities were variably documented as associated traits. The clinical phenotype of this disorder is reminiscent of the condition caused by activating mutations in ARF1[49], characterized by DD/ID, microcephaly, delayed myelination, cortical and cerebellar atrophy, periventricular heterotopia and seizures as major features, but also showing periventricular heterotopia. A related neurodevelopmental disorder characterized by DD/ID, progressive microcephaly, failure to thrive, and periventricular heterotopia has been linked to biallelic inactivating variants of ARFGEF2 (ARPHM, MIM: 608097)[16], encoding ARF-specific GEF stimulating the GTPase activation. Consistent with the observed clinical variability, our in vitro data demonstrate variable consequences of the identified disease-causing ARF3 variants on protein stability, nucleotide binding activity, and exchange, as well as on trans-Golgi and vesicle integrity and function. The differential impact of DN and CA on Golgi integrity is supported by in vivo validation. Zebrafish models, which recapitulate the pleiotropic effect and the variable strength of each ARF3 variant on developing brain and body axes, offer further insights into the underlying sub-cellular and cellular pathogenic mechanisms.

The activity of ARF3 at the trans-Golgi is tightly regulated via a conformational switch controlled by reversible GDP-to-GTP binding, which determines Golgi stability and trafficking[22,32]. Our structural inspection indicates that the majority of the disease-causing ARF3 mutations affect conserved residues involved in GDP/GTP binding/ exchange, previously reported to be mutated in other GTPases of the RAS superfamily (e.g., ARF1, HRAS, KRAS, NRAS, MRAS, RRAS, RRAS2), which cause neurodevelopmental syndromes or contribute to oncogenesis[49,96–101] (COSMIC database, Supplementary Fig. 2a and Supplementary Table 8). Lys127 in ARF3 (mutated in Subject 1) is homologous to Lys117 in HRAS (MIM: 190020), which if mutated causes Costello syndrome (CS [MIM: 218040])[102]. Pro47 (mutated in Subject 3) is homologous to Pro34 in HRAS, KRAS, and NRAS. The same Pro-to-Ser change has previously been reported as a somatic event in HRAS underlying vascular tumors[103], and other changes affecting this residue

in KRAS, HRAS, and NRAS have been described in RASopathies[104–106] (ClinVar IDs: VCV000040454, VCV001052630, VCV000039647). Furthermore, a missense change affecting the adjacent residue in ARF1 (p.Thr48Ile) was observed in a patient with clinical features overlapping with the present series[106]. In HRAS and KRAS, mutations affecting Thr58, which is adjacent to the aspartic acid residue homologous to Asp67 in ARF3 (mutated in Subject 2), have causally been linked to RASopathies[104,107,108]. Consistent with our findings, a recent report identified two missense changes affecting Asp67 and Arg99 of ARF3 in patients showing severe microcephaly at birth and progressive cortical and brainstem atrophy and epileptic seizures, and neurodevelopmental delay, cerebellar hypoplasia, and epilepsy, respectively[50]. Finally, both p.Pro47Ser and p.Asp67Val affect a conserved hydrophobic region of ARF3 involved in effector binding[24,30,109], with molecular dynamics simulations suggesting a major perturbation exerted by p.Pro47Ser on ARF3 binding to effectors. These considerations stimulate future studies aimed to demonstrate whether effector binding in these mutants is qualitatively and/or quantitatively altered.

Our in vitro data show an altered behavior of all ARF3 mutants in terms of stability and GTP binding. Among these, two amino acid changes, p.D93N and p.P47S were classified as CA, with the former exhibiting the strongest activation, in line with the severe and milder phenotypes observed in patients harboring these variants, respectively. The GTP binding behavior of the p.T32N substitution could be classified as DN, while a more complex behavior emerged for p.K127E and p.L12V/ D67V variants. The dramatically accelerated degradation of the two ARF3 mutants and reduced absolute levels of their GTP-bound forms cannot rule out the possibility of a loss-of-function behavior, which is in contrast with the activating role of the p.Lys117Arg substitution in HRAS causing upregulation of MAPK signaling in Costello syndrome[102,110]. Similar to this variant, however, a biochemical investigation performed on purified proteins in a cell-free system demonstrates an increased nucleotide exchange rate and excludes any significant impact on the GTPase catalytic activity. The reduced activity of the purified ARF3^{L12V/D67V} is instead in line with the structural prediction, anticipating a destabilization of both GTP and GDP binding via impaired coordination with Mg²⁺. It should be noted that myristylation is not achieved in the bacterial expression system employed to purify the proteins and the used cell-free assay does not account for the relevance of the lipid bilayer on the structural rearrangement of the GTPase and its and function[20,111]. Future dedicated experiments are required to more accurately examine the biochemistry of these mutants.

To functionally characterize the behavior of ARF3 mutants we profiled their impact on Golgi morphology. While depletion of ARF1 or ARF3 was not reported to affect Golgi structure[20], a differential impact of DN and CA forms are known. Golgi fragmentation with the

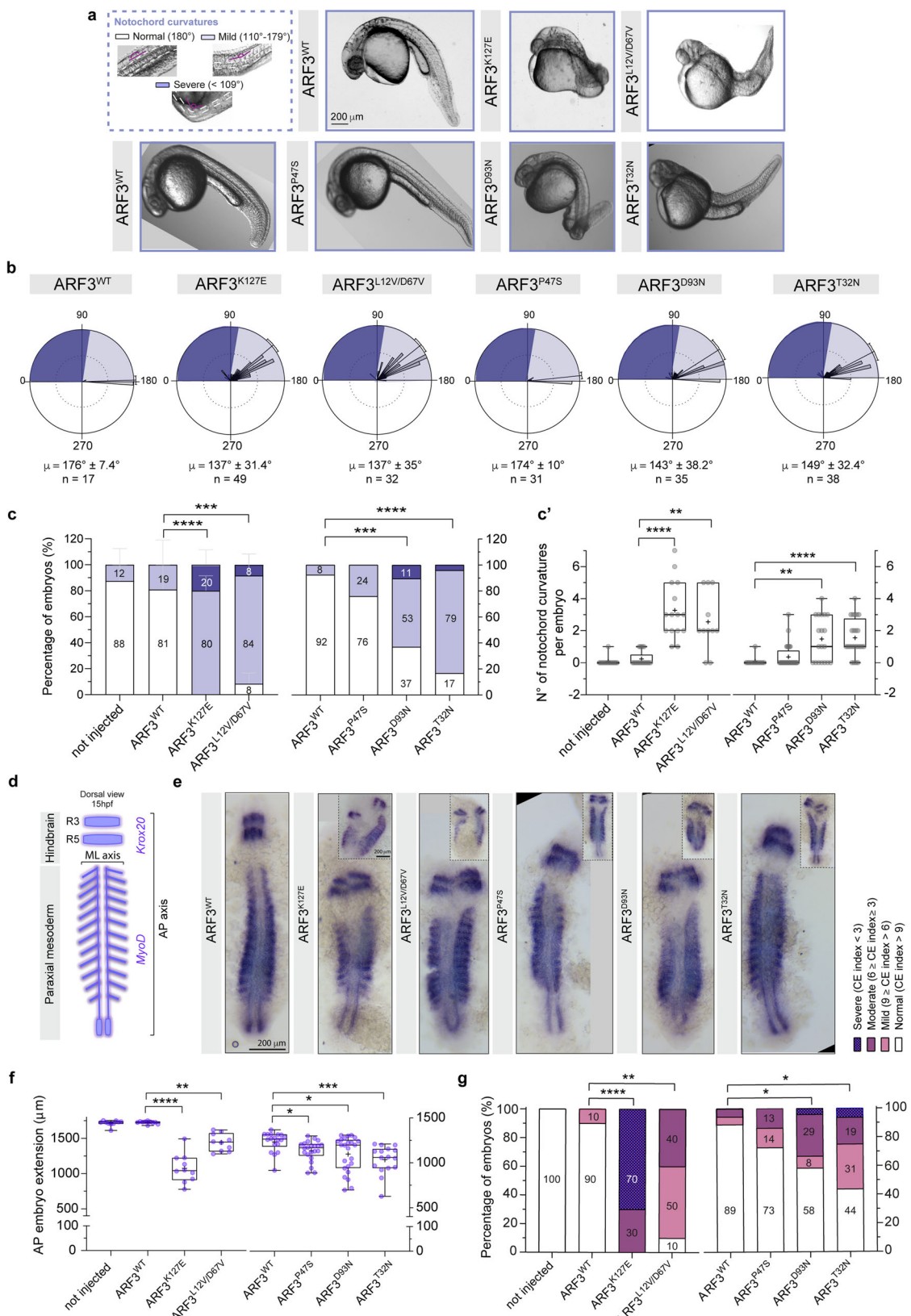

dispersion of specific Golgi associated and coat proteins (β-COP) is reported for DN ARF mutants[20,50,61], while an extension of the Golgi compartment, with swelling of Golgi and COP-I vesicles resulting in sustained vesiculation, has been associated with expression of constitutively active mutants[23,61]. The observed *trans*-Golgi and vesicle morphotypes identify different functional classes, mirroring CA and

DN behaviors. The pathophysiological relevance of these findings for embryonic development is supported by in vivo analysis of live *trans*-Golgi morphology in early zebrafish embryos overexpressing ARF3[K127E] and ARF3[D93N]. Nevertheless, the molecular mechanism by which aberrant ARF3 function causes different Golgi fragmentation patterns and the extent to which antero-retrograde transport might be impaired

**Fig. 10 | Aberrant ARF3 function causes axial defects with notochord curvatures and defective axes formation in zebrafish. a** Notochord curvatures of variable severity (purple angles schematics) and bright filed images of WT and mutant ARF3 expressing fish at 30 hpf. The images are representative of embryos from three (WT, K127E, and L12V/D67V), two (P47S), and one batch (other mutations). **b** Rose diagrams showing notochord angles, no. of angles = 17 (WT); 49 (K127E); 32 (L12V/D67V); 31 (P47S); 35 (D93N) and 38 (T32N) pooled from a total n of embryos indicated below (**c**). Mean vector (μ) ± circular SD is shown. Dark and light violet shadings in the rose diagrams represent mild and severe classes of notochord curvatures, respectively. **c** Incidence of embryos with mild or severe notochord curvatures, set 1: no. of embryos = 9 (not injected); 17 (WT); 15 (K127E, ****p < 0.0001) and 11 (L12V/D67V, ***p = 0.0005) of three independent batches; set 2: no. of embryos = 13 (WT); 28 (P47S); 19 (D93N, ***p = 0.0003) and 24 (T32N, ****p < 0.0001) of one batch. Data are expressed as mean ± SEM (set 1) or mean (set 2). **c'** Quantification of the number of notochord curvatures per embryo from one batch (same n of embryos as in c): K127E, ****p < 0.0001; L12V/D67V, **p = 0.0015; D93N, **p = 0.0022 and T32N, ****p < 0.0001. **d** Schematics of *Krox20* and *MyoD* expression at 15 hpf. Black square brackets indicate AP and ML axes. R3 and R5: rhombomeres 3 and 5. **e** Bright-field images showing *Krox20* and *MyoD* in situ mRNA staining (insets show severe cases). The images are representative of embryos from two independent batches for WT, K127E and L12V/D67V and from one batch for the other mutations. **f** Quantification of AP embryo extension, set 1: no. of embryos = 10 (not injected); 10 (WT); 10 (K127E, ****p < 0.0001); 10 (L12V/D67V, **p = 0.0078); set 2: no. of embryos 18 (WT); 22 (P47S, *p = 0.0169); 24 (D93N, *p = 0.0207) and 16 (T32N, ***p = 0.0002) of one batch. In **c'** and **f** data are expressed as box-and-whisker with median (middle line), 25th–75th percentiles (box), and min-max values (whiskers). All the data points and the mean ("+") are also shown. **g** Incidence of fish with different convergence and extension (CE) index values (same no. of embryos as in **f**) ****p < 0.0001 (K127E and L12V;D67V), **p = 0.0015; *p = 0.0251 (D93N), *p = 0.039 (T32N). Different datasets for the same measurement are shown in adjacent plots with the internal WT control for each set. Not injected controls between batches are not significantly different. Non-parametric Kruskal–Wallis followed by Dunn's multiple comparison *post hoc* test (**c', f**), Two-sided Chi-square's test in 2 × 2 contingency table (**c, g**, normal vs. phenotype) are used to assess statistical significance. Source data are provided as a Source Data file.

due to defective COP-I assembly and function remains to be determined. In this context, it should be noted that the specific Golgi phenotype shown by p.K127E, p.L12V/D67V, and, to a less extent, p.T32N are reminiscent of the BFA-induced effect in cells. Upon treatment with BFA, the ARF–GDP–GEF complex titrates the available GEF molecules away from the other ARF proteins, inhibiting their function[42]. Indeed, only co-occurring loss of function of multiple ARF proteins is able to perturb organelle and vesicle integrity[20,112].

Defective Golgi stability and activity is an emerging cause of cortical malformation[12]. The finding of fragmented Golgi in cells and embryos expressing the disease-associated ARF3 mutants assigns this disorder to the recently defined family of "Golgipathies", a group of heterogeneous neurodevelopmental disorders clinically characterized by a wide spectrum of CNS abnormalities[12,13]. Even if not detailed and therefore not easily comparable to our work, the recent functional investigation carried out by Sakamoto et al. on two *ARF3* mutations causing a similar neurodevelopmental condition corroborates the present findings.

Regulated transport through the endolysosomal system assists the differential targeting of cargos containing signaling molecules, polarity, and morphogenic factors to the membrane or to the *trans*-Golgi network for retrieval or to lysosomal degradation. The importance of healthy machinery to support this choice is just starting to gain attention in the context of development and disease[73,113]. Of note, our experiments with fluorescently labeled Tfn in combination with staining for EE, RE, and lysosomes unravel a cargo accumulation and delayed recycling in cells expressing ARF3 mutants specifically leading to early-onset microcephaly and severe skeletal defects. Interestingly, only an increased lysosomal Tfn cargo delivery seems to occur for p.D93N causing late-onset microcephaly and mild skeletal deformities. The data also suggest a general overload of the lysosomal system for all mutants, the relevance of which should be further investigated. Thereby, besides the clear distinction in Golgi morphological patterns, a variable impact on the efficiency of sorting and transport seem to underlie the variable severity observed in our patients. Adding to our evidence, impaired endosomal trafficking of EGF signaling components and Tfn recycling underlie proliferative defects recently identified as a major cause of a neurodevelopmental disorder with microcephaly[114,115].

The attentive in vivo investigation performed here makes a strong case for the importance of correct ARF3 function during a number of processes supporting embryogenesis. First, the extensive phenotypic characterization in zebrafish provides sufficient evidence of the pathogenicity of all the identified *ARF3* variants and their dominant nature. In line with the Golgi phenotype observed in vitro, over-expression of ARF3 mutants and downregulation of endogenous *arf3* in zebrafish embryos corroborate the DN (p.K127E, p.L12V/D67V, and p.T32N) and CA (p.D93N) behaviors in vivo.

Independently of the specific mechanism of dominance, all mutants modeled in fish recapitulated the involvement of brain and axes malformations and the variable strength of the disease manifestation. Strikingly, similar to patients, overexpression of ARF3[K127E] and ARF3[L12V/D67V] produce early-onset severe microcephaly in fish, while fish expressing ARF3[D93N] show late-onset microcephaly. Furthermore, in-depth brain analysis documents underlying brain volume reduction and validates defective forebrain white matter as a common feature of this neurodevelopmental disease. In line with these results, a severe vs. mild clinical phenotype was anticipated in the yeast *arf1* mutants involving Lys[127] and Asp[93], documenting complete or incomplete dominant lethal phenotypes, respectively[56].

Mechanistically, our in vivo work also provides insights into the processes that might hinder neurogenesis and contribute to neurodevelopmental defects. Our cell cycle profiling experiments indicate a possible delay of the mitosis in the early stages and an increased cell death within the developing anterior brain for the majority of the ARF3 fish mutants. A "Golgi checkpoint" sensing Golgi integrity and correct segregation has been suggested as an additional level to control cell cycle progression[116], whose contribution to the observed brain growth and morphogenesis defects is worth further investigation. On the other hand, our live imaging of the developing fish forebrain determined the occurrence of altered spindle morphology in microcephaly-causing ARF3 mutants, which might ultimately underlie mitotic arrest and cell death. Supporting this hypothetic scenario, mitotic arrest resulting in increased apoptosis of cells with aberrant spindle are appreciated as pathological mechanisms underlying conditions characterized by cortical malformations and microcephaly, some of which have already been successfully modeled in zebrafish[80,81,92,93,117]. The involvement of Golgi function in cell division, microtubule organization, and spindle formation is recognized[88,89]. In line with our data, elongated spindles result from the depletion of the constituent Golgi proteins, which keep the integrity of the organelle in mouse oocyte[90]. The role of ARF1 in mediating Golgi morphological changes during mitosis[43], and the function of class I ARF proteins in controlling proliferation have been reported[118,119], including the ARF3 involvement in cell cycle progression and apoptosis in gastric cancer[118]. Our zebrafish data show variably penetrant effects on cell proliferation and death, especially for p.P47S and p.T32N. This resembles the milder effect of these two mutants observed in cells in terms of Golgi integrity, vesicle assembly and recycling and anticipated by the mildest clinical traits associated with them. An investigation including a larger sample size might further clarify the presence of subtler effects. Conversely, additional, and not explored mechanisms might produce the mild phenotypes observed.

Lastly, strengthening the causal association of the described *ARF3* mutations to the observed skeletal traits, variable degrees of axial

malformations have been highlighted in the generated fish models recapitulating the severity of the disease in humans. Likewise in patients, similar defects are associated with DN and CA mutations, with p.K127E and p.L12V/D67V producing the most severe effects. Morphometric and live imaging analysis in early embryonic stages traced back the first assessable effect to defective PCP-mediated cell motility. CE movements, which require a fine-tuning of cell polarity factors within cells and are needed to shape the AP and ML axes[120–122], are evidently affected by ARF3 mutants. Of note, biosynthetic trafficking and correct function of ER and Golgi is essential during animal development for regulating morphogens' distribution[46,47,123], and required for cell polarity establishment and migration, as shown in vitro[124], nematodes[125], and zebrafish[126]. Consistent with our findings, fish expressing the CA microcephaly-associated ARF1 mutant show similar PCP-related axial defects[47]. Moreover, fish mutants for ARF-interacting COP-I/COP-II coat components exhibit skeletal and notochord abnormalities associated with Golgi disassembly[127,128]. On the other hand, post-Golgi trafficking and sorting of polarity components also contribute to brain development, underlying correct asymmetric cell division and migration in vertebrate neurogenesis, axon arborization, and synaptogenesis[46,129]. Along these lines, the impaired function of ARFGEF2 underlying the microcephalic traits observed in patients with ARPHM has been indeed linked to proliferative and migratory defects due to trans-Golgi to membrane trafficking of E-cadherins and beta-catenin[16]. The direct contribution of the impaired recycling found in cells to the observed brain defects in vivo remains to be assessed.

In conclusion, our work identifies ARF3 as a gene implicated, when mutated, in a clinically variable neurodevelopmental disorder belonging to the emerging class of "Golgipathies"[12,13]. Our findings specifically highlight the role of ARF3 in the maintenance of trans-Golgi integrity and document an obligate dependence of early developmental processes and brain morphogenesis on the proper function of this GTPase. The generated in vivo models represent a tool that can be exploited to deepen our understanding of the pathological mechanisms underlying the disease.

## Methods

### Subjects
The study has been approved by the local Institutional Ethical Committee of the Ospedale Pediatrico Bambino Gesù IRCCS (OPBG), Rome (1702_OPBG_2018). Subject 1 was analyzed in the frame of a research project dedicated to undiagnosed disorders (Undiagnosed Patients Program, OPBG), while the other subjects were referred for diagnostic genetic testing. Clinical data and DNA samples were collected, stored, and used following procedures in accordance with the ethical standards of the Declaration of Helsinki protocols, and after signed consent from the participating families. The authors affirm that human research participants provided informed consent for the publication of the images in Supplementary Fig. 1 and of clinical information potentially identifying individuals.

### Exome sequencing analysis
In all families, WES was performed using DNA samples obtained from leukocytes and a trio-based strategy was used. Target enrichment kits, sequencing platforms, data analysis, and WES statistics are reported in Supplementary Tables 3–7 and in the Supplementary Methods. WES data processing, read alignment to the GRCh37/hg19 version of genome assembly, and variant filtering and prioritization by allele frequency, predicted functional impact, and inheritance models were performed as previously reported[130–134]. WES data output is summarized in Supplementary Tables 3–7. Cloning of the genomic portion encompassing the c.34C>G and c.200A>T missense substitutions (p.Leu12Val and p.Asp67Val; Subject 2) was used to confirm that both variants were on the same allele. Variant validation and segregation were assessed by Sanger sequencing in all the subjects included in the study.

### Structural analysis and molecular dynamics simulations
The structural impact of the disease-associated missense changes was assessed using the available three-dimensional structures of human ARF3 complexed with GTP and V. vulnificus multifunctional-autoprocessing repeats-in-toxin (MARTX) (PDB 6ii6, https://www.rcsb.org/structure/6ii6)[52]. The structure was visualized using the VMD visualization software v.1.9.3[135].A model of GTP-bound ARF3 interacting with the cytosolic coat protein complex subunits γ-COP (COPG1) and ζ-COP (COPZ1) was built using the SWISS-MODEL automated protein structure homology modeling server (http://swissmodel.expasy.org)[136] using the 2.90 Å resolution X-ray structure (PDB 3TJZ)[59]. The alignment of template and model amino acid sequences is reported in Supplementary Fig. 21. The p.Asp67Val and p.Pro47Ser mutations were introduced using the UCSF Chimera v.1.15[137]. The side-chain orientations were obtained with the Dunbrack backbone-dependent rotamer library[138], choosing the best rotamer with minimal/no steric clashes with neighboring residues. Following the protonation of titratable amino acids at pH = 7, proteins were added in cubic boxes and solvated in water. Counter-ions were added to neutralize the charges of the system with the genion GROMACS tool[139]. After energy minimizations, the systems were slowly relaxed for 5 ns by applying positional restraints of $1000 \text{ kJ mol}^{-1} \text{ m}^{-2}$ to atoms. Unrestrained simulations were carried out for a length of 500 ns with a time step of 2 fs using GROMACS 2020.2. The CHARMM36 all-atom force field[140] was used for the protein description and water molecules were described by the TIP3P model[141]. V-rescale temperature coupling was employed to keep the temperature constant at 300 K[142]. The Particle-Mesh Ewald method was used for the treatment of the long-range electrostatic interactions[143]. The first 5 ns portion of the trajectory was excluded from the analysis. All analyzes were performed using GROMACS utilities.

### Expression constructs and in vitro mRNA synthesis
The full-length coding sequence of WT human ARF3 (NM_001659.3) was obtained by PCR and cloned into the pcDNA3.1/myc-6His eukaryotic expression vector (Life Technologies). The disease-associated substitutions were introduced in the pcDNA3.1/myc-6His expressing ARF3 WT and into pcDNA3/hArf3(WT)-mCherry (plasmid 79420, Addgene)[144] by site-directed mutagenesis (QuikChange II Site-Directed Mutagenesis Kit, Agilent Technologies, 200522-5). For zebrafish expression experiments, the myc-tagged (C-terminus) WT and mutant ARF3 sequences or mCherry-tagged plasmids were subcloned into the pCS-Dest vector (plasmid 22423, Addgene)[144] via LRII clonase-mediated recombination (ThermoFisher, 12538120). pCS-Dest-mKOFP2-CAAX and pCS-Dest-EGFP-GalT were generated by subcloning the ADDGENE plasmids 75155[145] and 11929[146], respectively. Plasmids were digested and linearized with KpnI-HF (NEB New England Biolabs, R3142S), and mRNA was produced using mMessage mMachine SP6 transcription kit and poly(A) tailing kit (Thermo Fisher, AM1340). All cloned sequences were confirmed by bidirectional DNA sequencing.

### COS-1 cell culture and transient transfection assays
COS-1 cells (CRL-1650-ATCC) were cultured in Dulbecco's modified Eagle's medium supplemented with 10% heat-inactivated fetal bovine serum (FBS, GIBCO, 10270-106), 1x sodium pyruvate (EUROCLONE, 11360-039) and 1x penicillin-streptomycin (EUROCLONE, ECB3001D), at 37 °C with 5% $CO_2$. Subconfluent cells were transfected with plasmids encoding myc- or mCherry-tagged WT and mutant ARF3, EGFP-GalT (Addgene, 11929[146]) and GFP-rab11 (Addgene, Plasmid #12674)[147] using FuGENE 6 (Promega, E2691), according to the manufacturer's instructions.

### Zebrafish husbandry
Zebrafish NHGRI, Tg(Xla.Tubb:DsRed)[148] and Tg(XlEef1a1:dclk2DeltaK-GFP)[149] were cultured following standard protocols. Fish were housed in a water-circulating system (Tecniplast ©) under controlled

conditions (light/dark 14:10, 28 °C, 350–400 μS, pH 6.8–7.2) and fed daily with dry and live food. All animal experiments were conducted under the approval of the Italian Ministry of Health (DGSA -Direzione generale della sanità animale e dei farmaci veterinari, 23/2019-PR).

## ARF3-myc immunoblotting in COS-1 and in zebrafish embryos

Transfected COS-1 cells were lysed in radio-immunoprecipitation assay (RIPA) buffer, pH 8.0, containing phosphatase and protease inhibitors (Sigma-Aldrich, P5726, P0044, P8340). Lysates were kept on ice for 30 min and centrifuged at 16,000×$g$ for 20 min at 4 °C. Samples containing an equal amount of total proteins (15 μg) were resolved by 12% sodium dodecyl sulfate (SDS)−polyacrylamide gel (Biorad, 1610185). Proteins were transferred to nitrocellulose membrane using a dry transfer system (Biorad), and blots were blocked with 5% non-fat milk powder (Biorad, 170-6404) in Phosphate-buffered saline (PBS) containing 0.1% Tween-20 for 1 h at 4 °C and incubated with mouse monoclonal anti-Myc (1:1000, Cell Signaling, 2276 S), mouse monoclonal anti-β-tubulin (1:1000, Thermo Fisher, 32-2600), mouse monoclonal anti-GAPDH (1:1000, Santa Cruz, sc-32233) and anti-mouse HRP-conjugated secondary antibody (1:3000, Thermo Fisher, 31450). For zebrafish immunoblots experiments, total protein lysates from a pool of non-injected control zebrafish embryos and from siblings injected with myc-tagged ARF3$^{WT}$ and mutant ARF3 different stages of development were obtained by syringe homogenization in cold lysis buffer (Tris−HCl 10 mM pH 7.4; EDTA 2 mM; NaCl 150 mM; Triton X-100 1% supplemented with 1X protease inhibitors cocktail (Roche, 11836170001) and equal amounts of protein extracts (40 μg) were separated on a 12% Sodium dodecyl sulfate (SDS)-polyacrylamide gel. The total protein concentration was determined by the Bradford assay (Bio-Rad) using Infinite M Plex (Tecan). After electrophoresis, the proteins were transferred to PVDF membrane using a wet transfer system (Biorad, for myc-tagged ARF3 mutants at 6 and 12 hpf) or nitrocellulose membrane using Trans-Blot Turbo Transfer System (Biorad, myc-tagged ARF3 mutants from 1.75 to 3.7 hpf). Blots were blocked with 5% non-fat milk powder (Biorad) or bovine serum albumin (BSA, Sigma-Aldrich, A8022-100G) in PBS containing 0.1% Tween-20 (Sigma-Aldrich, P2287) overnight at 4 °C constantly shaking and incubated with the primary antibody in blocking solution. The following primary antibodies were used: mouse monoclonal anti-myc (Cell Signaling, dilution 1:1000, 2276), rabbit polyclonal anti-GAPDH (1:1000, Genetex, GTX124503). Following washes in PBST 0.1%, membranes were incubated with anti-mouse- (1:3000, Thermo Fisher, 31450) and anti-rabbit-HRP-conjugated secondary antibodies (1:3000, Thermo Fisher, 31460). Immunoreactive proteins were detected by enhanced chemiluminescence (ECL) detection kit (Thermo Fisher, 34095) according to the manufacturer's instructions, and an Alliance Mini HD9 with Q9 Mini 18.02-SN software (Uvitec) was used for chemiluminescence detection. Uncropped blots are provided in the Source data file and Supplementary information.

## In vitro ARF3 protein stability assays

COS-1 cells were seeded at 3 × 10$^5$ in six-well plates and the following day was transfected with WT or mutant myc-tagged ARF3 expression constructs for 24 hours. A subset of transfected cells was then treated with CHX (10 μg/ml) (Sigma-Aldrich, C7698) for 3 and 6 h and with proteasome inhibitor MG132 (100 μM) (Sigma-Aldrich, C2211) or with the autophagy inhibitor bafilomycin A1 (200 nM) (Sigma-Aldrich, B1793) for 6 h to assess protein stability and degradation pathways. Alliance Mini HD9 with Q9 Mini 18.02-SN software (Uvitec) was used for chemiluminescence detection. Uncropped blots are provided in the Source data file and Supplementary information.

## ARF3 activity (GTP-bound state) assay in COS-1 cells

COS-1 cells (1 × 10$^6$) were seeded in 100 mm Petri dishes and transfected with myc-tagged *ARF3* expression constructs. Twenty-four hours after transfection, cells were washed twice with ice-cold PBS and collected in 50 mM Tris (pH 7.4), 150 mM NaCl, 10 mM MgCl$_2$, 10% Glycerol, 1% NP-40 with proteases and phosphatase inhibitors (Sigma-Aldrich, P5726, P0044, P8340). Cell lysates were further subjected to pull-down using GGA3-conjugated agarose beads (Cell Biolabs, STA-419) and incubated at 4 °C for 60 min. Control samples were pre-incubated with 100 μM GDP (Cell Biolabs, 240104) or the GTP analog guanosine-5′-(γ-thio)-triphosphate (GTPγS) (Cell Biolabs, 240103) for 30 min and then pulled-down according to the manufacturer's instructions (Cell Biolabs, STA-407-1). For immunoblotting analyzes, pulled-down samples including GTPγS/GDP controls and whole cell lysates were combined with a 2× sample buffer and denatured at 95 °C for 5 min. Samples were then separated by SDS-PAGE and incubated with mouse monoclonal anti-Myc and mouse monoclonal anti-GAPDH antibodies. GTP-bound protein level was detected by an ECL detection kit (Thermo Fisher, 34577). Alliance Mini HD9 with Q9 Mini 18.02-SN software (Uvitec) was used for chemiluminescence detection. Uncropped blots are provided in the Source data file.

## Protein expression and purification, nucleotide exchange, and GTP hydrolysis measurements

Proteins were isolated as glutathione S-transferase (GST) fusion proteins in *E. coli* strain CodonPlusRIL, purified after cleavage of the GST tag via gel filtration Superdex 75 or 200 (GE Healthcare, 28989333, 28989335). Nucleotide-free and fluorescent nucleotide-bound ARF3 proteins were using alkaline phosphatase (Sigma-Aldrich, P0762-250UN) and phosphodiesterase (Sigma-Aldrich, P3243-1VL) at 4 °C. Various fluorescence reporter groups, including Mant and Tamra, have been coupled to 2´ (3´)-hydroxyl group of the ribose moiety of GDP and GppNHp (Tamra GTP from Jena Bioscience, #NU-820-TAM, MantdGDP from Jena Bioscience, #NU-205L). All proteins were analyzed by SDS-PAGE and stored at −80 °C. Fluorescence polarization experiments were performed in a Fluoromax 4 fluorimeter in polarization mode. The excitation and emission wavelengths for Mant-deoxy-GDP were 360 nm (Slid width: 8 μm) and 450 nm (slid width: 10 μm), respectively. For nucleotide exchange reactions, 1 μM Mant-deoxy-GDP ARF3, 100 μM GDP, 10 μM ARFGEF BIG2 were used in 200 μl of measurement buffer containing 30 mM Tris/HCl, pH 7.5, 10 mM K$_2$HPO$_4$/KH$_2$PO$_4$, 2 mM MgCl$_2$ and 3 mM dithiothreitol at 25 °C. For GTP hydrolysis activity 1 μM of Tamra-GTP bound proteins were used with the excitation, wavelength of 543 nm (slit width: 8 micron) and emission wavelength of 580 nm (slit width: 10 micron), in a buffer containing 30 mM Tris/HCl (pH 7.5), 150 mM NaCl, 5 mM MgCl$^2$, 3 mM dithiothreitol and a total volume of 200 μl at 25 °C. The observed rate constants ($k_{obs}$) were calculated by fitting the data as single exponential decay using GraFit software v.5.0.13[150].

## RT-PCR prolife of endogenous *arf3a* and *arf3b* mRNAs in zebrafish embryos

To evaluate gene expression of *arf3a* and *arf3b* paralogs throughout zebrafish embryogenesis, total RNA was isolated from whole embryonic tissue samples at different stages of development (1–18 hpf) using TRIzol reagent (Invitrogen, 15596026). The first-strand cDNA was synthesized from total RNA using the SuperScript™ IV First-Strand Synthesis System (Thermo-Fisher, 18091050) according to the manufacturer's protocol and DNAse treatment was performed to avoid genomic DNA contamination. RT-PCR was performed with specific primer sets annealing on conserved regions of *arf3a* and *arf3b* zebrafish paralogs (ENSDART00000103639.5 and ENSDART00000053775.3, respectively) using GoTaq® G2 Green Master Mix (Promega, M7822) (Supplementary Table 11). Amplification of the elongation factor 1α (*eif1α*) was used as a housekeeping control gene. Alliance Mini HD9 with Q9 Mini 18.02-SN software (Uvitec) was used for signal detection. Uncropped gels are provided in the Supplementary information.

### RT-PCR of *myc-tagged ARF3* in COS-1 cells

To verify the expression of myc-tagged $ARF3^{L12V/D67V}$ and $ARF3^{K127E}$ after transient transfection in COS-1 cells, RNA from transfected cells was extracted using RNeasy Mini Kit (Qiagen, 74104) and the cDNA was retrotranscribed using oligo-dT SuperScript IV System kit (Thermo-Fisher, 18091050) according to manufacturer's protocols. The RT-PCR assay was designed to map the boundaries between the C-terminal coding region of ARF3 and the myc-tag sequence (Supplementary Table 11), low number of cycle (i.e. $n = 15$) were used in the PCR reaction to avoid PCR plateau phase and signal saturation. RT-PCR on GAPDH gene was used as internal housekeeping control. KAPA2G Fast ReadyMix (Sigma Aldrich, KK5603) was used to amplify the target sequence, according to protocol instructions. Alliance Mini HD9 with Q9 Mini 18.02-SN software (Uvitec) was used for signal detection. Uncropped gels are provided in the Supplementary information.

### ARF3 overexpression and arf3a and arf3b downregulation in zebrafish embryos

Injection of in vitro synthesized capped mRNAs encoding myc- and mCherry-tagged ARF3 (15 pg, $ARF3^{WT}$, and mutant ARF3), mKOFP-CAAX (15 pg), H2A-mCherry (15 pg), EGFP-GalT (15 pg), EGFP-CAAX (15 pg) and EGFP-GalT (50 pg) was performed in one-cell stage zebrafish embryos using FemtoJet 4x microinjection system (Eppendorf). mRNA was produced using mMESSAGE mMACHINE™ SP6 Transcription Kit Poly A Tailing Kit (ThermoFisher, AM1340, and AM1350). Injected embryos were cultured under standard conditions at 28 °C in fresh E3 medium and for each batch, not-injected fish were used as controls together with fish injected with the WT form of ARF3 mRNA. To perform *arf3a* and *arf3b* knockdown experiments in zebrafish embryos, 0.3 mM of each antisense morpholino oligonucleotides (MO, Gene Tools LLC) targeted specifically to the translational initiation site of *arf3a* and *arf3b* (resulting in 0.6 mM of MO in total) were co-injected into one cell stage embryos (Supplementary Table 11). All the injected fish were monitored every day, and survival rate and phenotype were scored at 24 and 48 hpf.

### ARF3 protein localization and Trans-Golgi morphology analysis in fixed COS-1 cells

COS-1 cells ($20 \times 10^3$) were seeded in 24-well cluster plates onto 12-mm cover glasses and transfected with WT or mutant mCherry-tagged ARF3 expression constructs for 48 h. Cells were then fixed with 4% paraformaldehyde for 10 min at room temperature, followed by permeabilization with 0.025% Triton X-100 for 5 min at room temperature. Cells were stained with mouse monoclonal anti-Golgin-97 antibody (1:50, Invitrogen, A21270) for 1 h at room temperature, rinsed twice with PBS and incubated with Alexa Fluor 488 goat anti-mouse secondary antibody (1:200, Invitrogen, A11017) for 1 h at room temperature. After staining, coverslips were mounted on slides by using Vectashield Antifade mounting medium containing 1.5 μg/ml DAPI (Vector Laboratories, H-1200-10). Images were acquired using a Leica TCS SP8X laser-scanning confocal microscope (Leica Microsystems) using LAS X software v.3.5 equipped with a pulsed and tunable white light laser source, 405 nm diode laser and 2 Internal Spectral Detector Channels (HyD) GaAsP, and an acousto-optical beam splitter (AOBS) allowing separation of multiple fluorescence spectra. Sequential confocal images were acquired using excitation laser lines at 405 nm (for DAPI, emission range 410–480 nm), 488 nm (for Alexa Fluor 488, emission range 500–550 nm) and 594 nm (for mCherry, emission range 600–650 nm), with a HC PL APO CS2 ×63 (numerical aperture, NA, 1.40) oil immersion objective (Leica Microsystems), a $1024 \times 1024$ format image, 0.38 μm pixel size and 400 Hz scan speed. Z-reconstructions were obtained from a z-step size of 0.5 μm, with an electronic zoom of 1.8x. Maximum intensity projection (MIP) of z-series was obtained by LAS X software v.3.5. Deconvolution analysis (HyVolution v.2 software, Leica Microsystems) using 'best resolution' algorithm of Golgin-97⁺ cisternae and vesicles was applied to z-stacks with an electronic zoom of 4x, to improve contrast and resolution of confocal raw images; then surface 3D rendering was generated from the deconvoluted images using LASX 3D Analysis tool (LAS X software v.3.5). Evaluation of different Golgi morphologies was performed by creating different masks with Fiji[151] to determine the Golgi area (Golgin-97⁺) and identify the cell boundaries (Arf3-mCherry⁺) in transfected cells. Total cell area, Golgi area and mean intensity (MI) for Golgin-97 were calculated for each cell. Whole-cell area was determined using mCherry fluorescence as a mask. Golgi mean intensity (MI of Golgin-97⁺ area, arbitrary units) vs. Golgi area [(Golgin-97⁺ area/total cell area)*100] were plotted for $ARF3^{WT}$, $ARF3^{Q71L}$ and $ARF3^{T31N}$ and the parameters giving the best separation between the chosen populations were manually selected, as reported in Fig. 3.

### Time-lapse imaging of Golgi dynamics in COS-1 cells

For live imaging, COS-1 cells ($10 \times 10^4$) were seeded into μ-dishes 35 mm (Ibidi) 24 h before transfection. The day after, cells were co-transfected with WT or mutant mCherry-tagged *ARF3* and EGFP-GalT constructs. Four hours post-transfection, time-lapse acquisitions were performed with a Leica TCS-SP8X confocal microscope (Leica Microsystems) with a PlApo CS2 ×20/0.75 objective, using excitation lines at 488 nm (for EGFP, emission range 500–550 nm) and 594 nm (for mCherry, emission range 600–650 nm). Parallel live imaging of control and mutant samples was performed simultaneously using the Mark & Find mode of the LAS X software v.3.5. Cells were monitored every 15 min and imaging was carried out with a $1024 \times 1024$ format, 0.38 μm pixel size, scan speed of 600 Hz, a zoom magnification up to 1.5 and z-step size of 0.7 μm, time-lapse microscopy was performed with a stage incubator (OkoLab) allowing to maintain stable conditions of temperature at 37 °C, with 5% of $CO_2$ and humidity during live cell imaging.

### Trans-Golgi analysis in live zebrafish embryos

*Trans*-Golgi morphology was assessed from single confocal images of the animal pole of late gastrulae injected with $ARF3^{WT}$ and ARF3 mutants as well as EGFP-GalT and mKOFP-CAAX mRNA. Images were acquired using a Leica TCS SP8X or Stellaris 5 confocal microscope (Leica Microsystems) equipped with LAS X software v.4.5, a pulsed and tunable white light laser source, 405 nm diode laser, hybrid detectors, and an acousto-optical beam splitter (AOBS) allowing separation of multiple fluorescence spectra, HC FLUOTAR L VISIR ×25/0.95 water-immersion objective, a $1024 \times 1024$ format at 400 Hz and z-step of 0.75 μm using laser line and emission filters as above. *Trans*-Golgi morphology (EGFP-GalT⁺) in each cell was scored independently by two researchers and two different categories of the EGFP-GalT⁺ signal were identified: cells showing a "ribbon-like" displayed a recognizable EGFP-GalT⁺ ribbon-like or a circular or semi-circular compact structure, while if no circular nor ribbon-like structure could be recognized and instead the pattern was more dotted, either with a single large and small dots recognizable or with spread dots distributed randomly inside the cytoplasm, cells were classified as harboring a "non-ribbon/puncta-like" EGFP-GalT⁺ staining, as indicated in the main text and in Supplementary Fig. 10.

### Transmission electron microscopy analysis of Golgi morphology

COS-1 cells were seeded at $2 \times 10^5$ in six-well plates and transfected with ARF3 WT, K127E and D93N for 48 h. Cells were then fixed with 2.5% glutaraldehyde in 0.1 M sodium cacodylate buffer (pH 7.2) 1 h at RT, washed in buffer and post-fixed in 1% $OsO_4$ in 0.1 M sodium cacodylate buffer for 1 h at RT. Post-fixed specimens were washed in buffer, dehydrated through a graded series of ethanol solutions (30–100% ethanol) and embedded in Agar Resin Kit (Agar Scientific, R1031). Ultrathin sections, obtained by a UC6 ultramicrotome (Leica), were stained with uranyl acetate and Reynolds' lead citrate and examined at 100 kV by a Philips EM 208S transmission electron microscope (FEI-Thermo Fisher) equipped with acquisition system Megaview III SIS camera (Olympus-SIS Milan, Italy) and iTEM3 software.

## Assessment of COP-I vesicles assembly, Tfn accumulation, and recycling in COS-1 cells expressing WT and mutant ARF3

COS-1 cells ($20 \times 10^3$) were seeded in 24-well cluster plates onto 12-mm cover glasses and transfected with WT or mutant mCherry-tagged ARF3 expression constructs. After 48 h of transfection, COS-1 cells were washed and serum starved for 45 minutes at 37 °C and was then incubated with 50 µg/ml Alexa Fluor 488 or Alexa Fluor 647 conjugated transferrin (Invitrogen, T13342 and T23366, respectively) in serum-free medium for the indicated time (5 or 30 min). At the end of the 37 °C incubation period used to follow intracellular trafficking, the cells were cooled to 4 °C, washed twice with ice-cold PBS to remove unbound transferrin, and then incubated twice for 2 min at 4 °C with ice-cold stripping buffer (150 mM NaCl, 20 mM HEPES, 5 mM KCl, 1 mM CaCl2, 1 mM MgCl2, pH 5.5) to remove the excess of transferrin bound to the cell surface. To follow the formation of COP-I vesicles and the levels of Tfn present in early endosomes (EE), and lysosomes after 30 min of incubation, cells were fixed with 4% paraformaldehyde for 10 min at room temperature, followed by permeabilization with 0.025% Triton X-100 for 5 min at room temperature and were then stained with antibodies recognizing COP-I, EE and lysosomal markers overnight for 1 h at 4 °C. These antibodies were used: rabbit polyclonal anti-ß COP (1:2000, Abcam, ab2899), rabbit monoclonal anti-Rab5 (1:200, Cell Signaling, 3547S) and mouse monoclonal anti-LAMP-2 (1:200, Santa Cruz, sc-18822), respectively. Cells were then rinsed twice with PBS and incubated with Alexa Fluor 633 goat anti-mouse secondary antibody (1:200, Invitrogen, A21050) or Alexa Fluor 488 goat anti-rabbit secondary antibody (1:200, Invitrogen, A11070) for 1 h at room temperature. To assess the levels of Tfn present in recycling endosomes (RE) after 30 min of incubation, cells were co-transfected with the plasmid encoding the GFP-tagged RE marker Rab11. All the slides were mounted using Vectashield Antifade mounting medium containing 1.5 µg/ml DAPI (Vector Laboratories, H-1200-10). Confocal z-stacks of the cells incubated 5 min with Tfn and of those stained for βCOP or the cells expressing GFP-Rab11 and incubated 30 min with Tfn were obtained using TCS-SP8X confocal microscope (Leica Microsystems) with an HC PL APO CS2 ×63 (NA 1.40) oil immersion, using excitation lines at 488 nm (for GFP, emission range 500–550 nm), 594 nm (for mCherry, emission range 600–650 nm), 640 (for Alexa Fluor 647, emission range 650–700 nm) and 630 nm (for Alexa Fluor 633, emission range 640–690 nm), a $1024 \times 1024$ format at 400 Hz, 0.14 µm pixel size, a zoom magnification of ×2.5 and z-step size of 0.7 µm. Confocal z-stacks of the cells stained for Rab5 and Lamp2 at 30 min upon Tfn treatment were obtained using Stellaris 5 (Leica Microsystems), A HC PL APO CS2 ×63/1.40 oil objective was used. Confocal acquisitions were performed using 405, 488, 594, 633 nm laser excitation lines for the different fluorophores used and emission spectra as above. Sequential acquisitions were performed and with a $1024 \times 1024$ format at 400 (Rab 5) or 200 Hz (Lamp 2), and a z-step of 0.7 µm. For all the experiments, laser intensity and detector parameters and offsets were unchanged per each condition of a single experiment. Phenotype categories for cells expressing WT and mutant ARF3 were independently inspected by two researchers and classified as reported in the main text. Briefly, for COP-I vesicles assembly, the number of cells (%) showing "assembled", "disassembled" or "extended" COP-I compartment at the PN was estimated by the occurrence of clustered, absent /highly scattered or increased β-COP+ area, respectively. The β-COP+ area was measured using a manual ROI selection, ROI manager, and measurement tool in Fiji. For each cell, the β-COP+ area was normalized by the total area of the nucleus and cells with an extended COP-I compartment were estimated by a ratio β-COP+ area/nucleus area. The distribution patterns of the Tfn inside the cytoplasm were scored in replicate for the different incubation times. The Tfn signal distribution inside the cell and its relative clustering at the perinuclear compartment (PN) were inspected and the number of cells (%) showing different classes of Tfn phenotype was calculated. Three

different categories of Tfn distributions were identified (accumulation at the PN = "clustered", presence of a Tfn+ clusters and sparse dotted signal throughout the cell = "semidispersed", no Tfn+ clusters visible = "dispersed"). Co-occurrence of Tfn with Rab5, Rab11 or Lamp2 at the PN was quantified using the colocalization algorithm of IMARIS softare v.9.5 (Bitplane) and the built-in thresholding method, which was equally applied to all conditions within each experiment. Manders overlap coefficients were calculated for the fraction of Tfn overlapping the respective fluorophores in a manually selected ROI corresponding to the visible Tfn+ PN cluster and viceversa[152].

## Zebrafish body axis, notochord, and head phenotyping

Embryos were screened for gross phenotype penetrance and classified as class I ("normal"), class II ("developmental delay"), class III ("mild" microcephaly, with/without microphthalmia and/or mild shortening and lateral bending of body axis) and class IV ("severe" microcephaly/anencephaly with/without microphthalmia and marked trunk deformity, defective body elongation, and severe lateral bending) at 24 and 48 hpf, as reported in the main text. For detailed analysis, not injected controls and injected fish at 12 and 15 hpf (for body axis), 30 hpf (for notochord), and 48 hpf (for head size) were embedded in 2% low melting agarose (Sigma-Aldrich) dissolved in E3 medium. Bright-field images were acquired at Leica M205FA microscope (Leica Microsystems) equipped with LAS X software v.3.0, ×0.63 magnification (for body axis and head size) and Leica Thunder Imager microscope (Leica Microsystems) with N PLAN 5X/0.12 PHO objective (for notochord) equipped with LAS X software v.3.7. These parameters were assessed: (i) head size measured by the area surface between the rostral most part of the head and the optic vesicle; (ii) degree of the notochord angles (plotted in a Rose diagram using Oriana software v.4[153]) and incidence of embryos showing mild and severe notochord curvatures (calculated on the average degree of all the curvatures per embryo); (iii) number of notochord curvatures and (iv) antero-posterior (AP) embryo extension, measured as length from the most anterior region of *Krox20* domain (R3) to the posterior region of *MyoD* domain (paraxial mesoderm) and (v) convergence-extension index, calculated as the ratio of AP and medio-lateral axes (ML, calculated as the length of the visible first somite); (vi) angle between the antero-posterior ends.

## Zebrafish live imaging of brain volume and 3D rendering from *Tg(NBT:dsRed)* fish

Live *Tg(NBT:dsRed)* fish injected with WT and mutant ARF3-encoding mRNAs at 48 hpf were embedded in 2% low melting agarose/E3 medium and imaged using Leica Stellaris 5 confocal microscope (Leica Microsystems) equipped with LAS X software v. 4.5 using a sensitive hybrid detector and keeping minimal laser power (579 nm wavelength). Live z-stacks were performed using HC FLUOTAR L VISIR water immersion ×25/0.95 objective, and with an excitation lines at 594 nm (emission range 600–650 nm), a $512 \times 512$ format at 400 and 600 Hz and a z-step size of 2.5 µm. Volumetric brain reconstructions and quantifications were obtained using 3D Volume (Blend model) and Surfaces rendering reconstruction algorithm of IMARIS v.9.5 (Bitplane), employing the same parameters for the different individuals.

## Whole-mount immunofluorescence in 48 hpf zebrafish embryos

Zebrafish embryos were fixed in 4% PFA/PBS (Thermo Fisher, 28908) for 3 h at room temperature (RT), dehydrated through methanol washes from 25% (in PBS) to 100% and stored at −20 °C. Fixed samples were rehydrated back to PBS through serial washes and incubated with 150 mM Tris–HCl pH 9.0 for 5 min at RT and 15 min at 70 °C for antigen retrieval. The samples were then permeabilized with 0.8% PBST (Triton, Sigma-Aldrich, X-100) and 1 µg/ml proteinase K (Sigma-Aldrich, P2308) for 40 min at RT (for acetylated α-tubulin and HuC/Elav) or with cold Acetone for 20 min (for PCNA and Phospho-Histone H3, pH3). The

samples were post-fixed in 4% PFA/PBS for 20 min at RT and incubated 4 h in a solution containing 10% of normal goat serum (NGS) and 2% of BSA in 0.8% PBST for 4 h at 4 °C. These primary antibodies were used: mouse anti-acetylated tubulin (1:250, Sigma Aldrich, T7451), rabbit anti-ElavI3 + 4 (1:100, GeneTex, GTX128365), rabbit anti-PCNA at (1:250, GeneTex, GTX12449), mouse anti-pH3 (1:250, Abcam, ab14955) overnight at 4 °C with gentle shaking. These secondary antibodies were used: goat anti-mouse Alexa Fluor 488 (1:1000, Thermo Fisher, A11001), goat anti-rabbit Alexa Fluor 633 (1:1000, Thermo Fisher, A21070), goat anti-rabbit DyLight 594 (1:1000, GeneTex, GTX213110-05). The Stellaris 5 confocal microscope (Leica Microsystems) equipped with LAS X software v. 4.5 and an HC FLUOTAR L VISIR water immersion 25x/0.95 objective were used for z-stack image acquisition. Ventral z-stacks of embryos stained for acetylated α-tubulin, mounted in 2% low melting agarose/PBS (Sigma-Aldrich, A9414), were acquired with: 499 nm laser line (emission range 510–600 nm), 1024 × 1024 format at 400 Hz, and 1.5 μm z-step size. Z-stacks of embryos stained for PCNA and pH3, mounted in 90% glycerol/PBS, were acquired sequentially with 591 nm and 631 nm laser lines and emission range 596–636 and 644–700 nm, respectively. Volumetric brain acquisitions were obtained by scanning with a 1024 × 1024 format at 200 Hz and 0.5 or 0.7 μm z-step size and with a digital zoom of 2.5 (ventral brain) or by scanning with a 512 × 512 format at 400 Hz and 2 μm z-step size (dorsal brain). Embryos stained for HuC/Elav were imaged using the confocal microscopes Leica TCS-SP8 or Olympus FV1000 equipped with FV10-ASW version 4.1, 20×/0.75 objective, using laser line 543 (for HuC/Elav, emission range 560–620 nm) and 635 nm (for acetylated tubulin, emission range 655-755 nm) scanning with a 1024 × 1024 format, a speed of 400 Hz and z-step size of 2 μm.

### Analysis of the forebrain anterior commissure

For morphological analyzes of the forebrain anterior commissure and its lateral bundles, the medial and lateral width was calculated using the "line selection" tool in Fiji[151] as indicated in Supplementary Fig. 17.

### Analysis of the number of mitotic cells and the pH3 distribution in precursor cells of the developing anterior brain

The number of pH3[+] precursors cells (mitotic) was quantified in both ventral and dorsal anterior brain z-stacks of the stained 48 hpf embryos. The number of cells was quantified manually or using the "spot analysis" algorithm of IMARIS software 9.5 (Bitplane) throughout the 3D volume of the confocal acquisitions, applying the same parameters to all the scans. The automatic analysis was manually inspected and corrected. Superficial staining corresponding to large epidermal cells was excluded from the analysis. A 3D rendering in the whole dorsal anterior brain was also performed using the same software and showing the pH3+ cells. For the ventral scans, the pH3 signal positivity and distribution pattern was assessed in precursor cells found within the defined ROI corresponding to the proliferative zone of a ventral portion of the forebrain, clearly discernible in the confocal acquisitions. The total number of pH3+ cells (mitotic) were counted along the full z-stack of the ROI. Following Tang et al.[154], cells at different mitotic phases corresponding to early (prophase + prometaphase), metaphase, and late (anaphase + telophase) were scored based on the pH3 signal distribution, indicative of the cell's chromatin topological status. For the dorsal scans, to assess possible alterations of the normal spatial distribution of pH3[+] cells within the optic tectum, the number of mitotic cells in different regions (left and right neuropil vs. medial proliferative zone and cerebellum), was manually quantified.

### Staining and analysis of apoptotic cells in the anterior brain

To assess apoptotic cells in the anterior brain of developing fish expressing WT and mutant ARF3, live staining with acridine orange (AO) was performed. Briefly, microinjected larvae at 48 hpf were incubated with 100 μg acridine orange (Sigma-Aldrich, A6014) in E3 medium for 1 h in the dark at 28 °C then washed extensively with E3 and mounted for microscopy in multi-well dishes using 1.5% low melting agarose/E3 (Sigma-Aldrich, A9414). Live z-stacks of the anterior forebrain in x, y, z were acquired at the Thunder Imager microscope (Leica Microsystems), using HC PL Fluotar ×10/0.32 DRY objective, 2048 × 2048 format, 475-nm excitation line and with a z-step of 5 μm. AO[+] spots were counted using the "multi-points" tool of Fiji[151] after adjusting brightness and contrast equally across all the conditions of a single experiment.

### Live imaging and analysis of mitotic spindles in precursor cells of the anterior brain

To examine spindle morphology of dividing progenitor cells in zebrafish forebrain, transgenic Tg(XlEef1a1:dclk2DeltaK-GFP) fish marking microtubules were injected with mRNA encoding my-tagged ARF3[WT], ARF3[K127E] and ARF3[D93N] together with the mRNA encoding H2AmCherry marking the chromatin. Fish at 1 dpf (~30 hpf) were embedded in 1.5% low melting agarose in E3 medium and live confocal x, y, z, t acquisitions of the forebrain region were obtained at the Stellaris 5 confocal microscope (Leica Microsystems), with an HC Fluotar VISIR L ×25/0.95 water immersion objective, 488 nm laser line (for GFP, emission range 504–600 nm), and 594 nm (for mCherry, emission range 625–700 nm) in sequential model and with a format of 512 × 512 at 400 Hz and with a z-step of 1.5 μm. A time interval (TI) of 4 minutes was set between consecutive scans for ~1 h (ARF3[WT] and ARF3[D93N], acquired with a digital zoom of 1.7) and >5 h (ARF3[WT] and ARF3[K127E], acquired with a digital zoom of 1). The mitotic spindle's length and width were analyzed using the selection and measurement tools in Fiji[151].

### Whole-mount in situ hybridization of *Krox20* and *MyoD* mRNA

The fragments of Krox-20 and MyoD cDNA used for riboprobe synthesis were amplified from a zebrafish cDNA preparation by PCR using One Taq DNA polymerase (NEB New England Biolabs, BM0509S) and the primers listed in Supplementary Table 11. The PCR fragments were cloned into pGEM-T Easy vector (Promega, A1380) and sequences were confirmed by DNA sequencing. The digoxigenin-labeled antisense riboprobes were synthesized by in vitro transcription with DIG RNA labeling kit SP6/T7 (Roche, 11277073910). In situ hybridization analysis in whole-mount zebrafish embryos at 15 hpf was performed as described in Thisse et al.[155]. Briefly, samples were permeabilized with proteinase K treatment (1 μg/ml, Sigma-Aldrich, P2308-25MG) for 2 min, preincubated in 2% blocking reagent (Roche, 11096176001) and incubated with the transcribed riboprobes (2 ng/μl) in hybridization mix: 50% formamide (Sigma-Aldrich, F9037-100ML), 1.3× SSC 20X (175.3 g NaCl, 88.2 g Na Citrate 300 mM, PH 7), 100 g/ml heparin, (Sigma-Aldrich, H3149-25KU), 50 μg/ml yeast RNA (Sigma-Aldrich, 10109223001), 0.2% Tween-20.0 (Sigma-Aldrich, P2287), 5% CHAPS (Merck, 850500P-1G), 5 mM EDTA pH 8 (Sigma-Aldrich, E7889-100ML) at 65 °C for at least 15 h. Afterward, samples were rinsed with scalar dilutions of SSC solutions (25%, 50%, 75%) and incubated with anti-alkaline phosphatase (AP)-conjugated antibody (1:5000, Roche, 11093274-910) for 2 h at room temperature. Chromogenic staining was developed via BM Purple substrate (Sigma-Aldrich, 11442074001) according to the manufacturer's instructions. Specimens were mounted in 90% glycerol and dorsal images were acquired at the Olympus TH4-200 microscope (Olympus) equipped with Olympus cellSens Standard software v.1.14, with C Plan 10X/0.25 RC1 objective or Thunder Imager (Leica Microsystems) with N PLAN ×5/0.12 PHO objective.

### Embryo extension analysis at 13 hpf and confocal live imaging of zebrafish embryos during late gastrulation

Overall extension of embryos expressing myc-tagged ARF3 WT and mutants at 13 hpf was estimated by calculating the angle amplitude

between A-P ends (head and tail bud, respectively) from bright-field images acquired at Leica M205FA microscope (Leica Microsystems), using the "angle measurement" tool of Fiji[151]. For in vivo gastrulae imaging, embryos at the mid-gastrula stage were embedded in 2% low melting agarose (Sigma-Aldrich, A9414) in E3 medium. 4D ($x$, $y$, $z$, $t$) fluorescent data were acquired using Leica TCS-SP8 confocal microscope (Leica Microsystems) with hybrid detectors and keeping minimal laser power. Time-lapses were acquired with a PlApo CS2 ×20/0.75 objective using a 488 nm laser line (for GFP, emission range 495–550 nm) and 594 nm (for mCherry, emission range 605–680 nm). Z-stacks were obtained with a TI of 30 min, a 1024 × 1024 format at 400 Hz and a z-step size of 3 µm. Fluorochromes unmixing was performed by the acquisition of an automated-sequential collection of multi-channel images, to reduce spectral crosstalk between channels, and the same setting parameters were used for all examined samples. Embryo live imaging was performed simultaneously using the Mark & Find the mode of the LAS X software.

### Statistics and reproducibility

Data were analyzed independently by at least two researchers and statistical assessments were performed using GraphPad software v.9 (Prism). Log-rank (Mantel-Cox) test was used to assess survival in zebrafish mutants and morphants. For phenotypes penetrance assessment Chi-squared test in a 2 × 2 contingency table was used, performed as pairwise statistical comparisons across experimental conditions as indicated in the text and figures. Normality tests (Anderson–Darling, D'Agostino and Pearson, Shapiro–Wilk and Kolmogorov–Smirnov tests) were run to assess normal distribution of the data. Parametric data with more experimental groups were analyzed with ANOVA test, non-parametric data with Kruskal–Wallis test and specific *post hoc* tests were always used as indicated in the figure legends and Statistical Table 12. All the analyses were two-tailed.

### Image processing

Raw images were analyzed with Fiji[151], LAS X Life Science imaging software (Leica Microsystems, v.3, 3.5, 3.7, 4.5), FV10-ASW software v.4.1, Olympus cellSens Standard imaging software v.1.14 (Olympus Life Science) and IMARIS v.9.5 (Bitplane) and processed using Photoshop or Illustrator (Adobe Systems Incorporated) for figure assembly. Image acquisition parameters (laser lines and power, detector settings, objective used) were maintained equally within each experiment. Brightness and contrast were adjusted equally across whole images and between images belonging to the same experiment. Exceptions are explained here. In Fig. 10, due to possible background differences within each imaged well, each brightfield image showing the notochord was individually adjusted for brightness and contrast to allow the best notochord visualization. Given the high signal intensity that would shadow the Golgi staining, in Supplementary Fig. 10 the brightness of the mKOFP signal (blue) used only for membrane rendering was reduced in post-processing for ARF3[K127E] as compared to ARF3[WT]. All the schematic illustrations were generated by researchers using Illustrator 2021–2022 (Adobe) except for the human brain illustration item in Fig. 8 which was created with BioRender.com and modified in Illustrator 2021–2022 (Adobe).

### Reporting summary

Further information on research design is available in the Nature Research Reporting Summary linked to this article.

## Data availability

The clinical data were collected after signed consent forms. The entire dataset is included within the manuscript. Given the progressive nature of the disease, additional information eventually collected after this publication will be made available upon request to the corresponding authors (antonella.lauri@opbg.net, marco.tartaglia@opbg.net) and referring clinicians. The sequencing data are available under restricted access for privacy/ethical reasons, access can be obtained by contacting the corresponding authors. The ARF3 variants identified in this study have been deposited in the ClinVar database under the following accession codes: SCV002549683 (c.34C>G, p.Leu12Val) [(ARF3):c.34C%3EG%20(p.Leu12Val)], SCV002549684 (c.95C>A, p.Thr32Asn) [(ARF3):c.95C%3EA%20(p.Thr32Asn)], SCV002549685 (c.139C>T, p.Pro47Ser) [(ARF3):c.139C%3ET%20(p.Pro47Ser)], SCV002549686 (c.200A>T, p.Asp67Val) [(ARF3):c.200A%3ET%20(p.Asp67Val)], SCV002549687 (c.277G>A, p.Asp93Asn) [(ARF3):c.277G%3EA%20(p.Asp93Asn)], and SCV002549688 (c.379A>G, p.Lys127Glu) [(ARF3):c.379A%3EG%20(p.Lys127Glu)]. The UCSC GRCh37/hg19 human genome assembly used as a reference for reads alignment is available at https://www.ensembl.org/info/website/tutorials/grch37.html. The dbSNP150, gnomAD V.2.1.1, ClinVar, and COSMIC v.96 databases used in this paper are available at https://gnomad.broadinstitute.org/, https://genome.ucsc.edu/cgi-bin/hgTrackUi?db=hg38&g=snp150Common and (https://www.ncbi.nlm.nih.gov/clinvar/), https://cancer.sanger.ac.uk/cosmic, respectively. The raw blots and raw data for the different measures of this study are provided in the Supplementary Information/Source data file. Due to the large size of each dataset, all the raw imaging data, supporting the findings of the work are available from the corresponding authors upon request. All the constructs generated in this study will be shared upon request to the corresponding authors. Source data are provided with this paper.

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

## Acknowledgements

H2A-mCherry:pDest and GFP-CAAX:pDest plasmids were kindly provided by Dr. Mette Handberg-Thorsager (Max Planck Institute of Molecular Cell Biology and Genetics, Dresden, MPI-CBG, Germany), pcDNA3/hArf3(WT)-mCherry was a gift from Kazuhisa Nakayama (Addgene plasmid # 79420; http://n2t.net/addgene:79420; RRID:Addgene_79420), EGFP-GalT was a gift from Jennifer Lippincott-Schwartz (Addgene plasmid # 11929; http://n2t.net/addgene:11929; RRID:Addgene_11929), GFP-rab11 WT was a gift from Richard Pagano (Addgene plasmid # 12674; http://n2t.net/addgene:12674;RRID: Addgene_12674). *Tg(XlEef1a1:dclk2DeltaK-GFP)* was a kind gift from Prof. M. Mione (Laboratory of Experimental Cancer Biology—CIBIO, Trieste, Italy). We acknowledge Cineca ELIXIR-IIB for computing resources and thank Dr. Giovanna Zambruno for providing support and technical advice on TEM. This work was supported, in part, by Fondazione Bambino Gesù (Vite Coraggiose to M.T.), Italian Ministry of Health (CCR-2017-23669081, RCR-2020-23670068_001 and RCR-2021-23671215 to M.T.; RF-2018-12366931 to F.C.R., G.C., and B.D.; Ricerca Corrente 2021, Ricerca Corrente 2022 to A.L., S.C., and M.T., and RF-2013-02355240 to R.G.), Ministero della Salute (Ricerca 5x1000) to A.L. and M.T, Italian Ministry of Research (FOE 2019 to M.T.), the Tuscany Region Call for Health 2018 (DECODE EE, to R.G.), and the European Union Seventh Framework Program (DESIRE [602531] to R.G.), Istituto Superiore di Sanità (Bando Ricerca Indipendente ISS 2020-2022-ISS20-39c812dd2b3c to S.C,).

## Author contributions

G.F. designed and performed the in vivo experiments and contributed to analyzing in vitro cell experiments and writing the manuscript. V.M. designed and performed the in vitro experiments and contributed to writing the manuscript. F.C.R. coordinated the clinical data collection and phenotyping, analyzed the clinical data, and contributed to writing the manuscript; M.V. performed the in situ hybridization assay, contributed to constructs preparation and to the analysis of cells experiments; N.M. performed the cell-free biochemical experiments with F.B.; S.C. designed and performed the Golgi morphological experiments in cells and contributed to the recycling and pull-down experiments with E.Z.; G.P. performed the cell proliferation and death assays in zebrafish; L.A.C. and S.Pe. contributed to the confocal scanning experiments. A.Z., A.V., F.P., S.Pi., A.B., and M.I. generated and analyzed the genomic data; G.C., performed the structural analyses and molecular dynamics simulations with I.G.P. and B.C.; L.B. e A.T. performed the TEM experiments; C.M. performed the ARF3 RT-PCR experiment in COS-1 cells; R.M. contributed to TEM sections preparations; M.B., C.B., D.M., A.Se., M.M., M.V.G., A.B., C.J.C., R.G., A.Sl. and B.D. identified the patients, and collected and analyzed the clinical data; M.R.A. supervised the cell-free biochemical experiments and analyzed the data; A.L. and M.T. conceived, designed, and supervised the project, analyzed the data, wrote and revised the manuscript. G.F., V.M., and F.C.R. equally contributed to the work, M.V. and N.M. equally contributed to the work. A.L. and M.T. jointly coordinated the study.

## Competing interests

The authors declare no competing interests.

## Additional information

[1]Genetics and Rare Diseases Research Division, Ospedale Pediatrico Bambino Gesù, IRCCS, 00146 Rome, Italy. [2]Institute of Biochemistry and Molecular Biology II, Medical Faculty and University Hospital Düsseldorf, Heinrich Heine University Düsseldorf, Düsseldorf, Germany. [3]National Center for Rare Diseases, Istituto Superiore di Sanità, 00161 Rome, Italy. [4]Department for Innovation in Biological Agro-food and Forest systems (DIBAF), University of Tuscia, 01100 Viterbo, Italy. [5]Department of Biology and Biotechnology "Charles Darwin", Università "Sapienza", Rome 00185, Italy. [6]UFR Santé de l'Université d'Angers, INSERM U1083, CNRS UMR6015, MITOVASC, SFR ICAT, F-49000 Angers, France. [7]Département de Génétique, CHU d'Angers, 49000 Angers, France. [8]Institute of Biomembranes, Bioenergetics and Molecular Biotechnologies, Centro Nazionale delle Ricerche, 70126 Bari, Italy. [9]Servizio grandi strumentazioni e core facilities, Istituto Superiore di Sanità, 00161 Rome, Italy. [10]Centro di riferimento per la medicina di genere, Istituto Superiore di Sanità, 00161 Rome, Italy. [11]Pediatric Neurology, Neurogenetics and Neurobiology Unit and Laboratories, Meyer Children's Hospital, University of Florence, 50139 Florence, Italy. [12]Confocal Microscopy Core Facility, Ospedale Pediatrico Bambino Gesù, IRCCS, 00146 Rome, Italy. [13]Department of Oncology and Molecular Medicine, Istituto Superiore di Sanità, 00161 Rome, Italy. [14]Super Computing Applications and Innovation, CINECA, 40033 Casalecchio di Reno, Italy. [15]Fondazione IRCCS Ca' Granda Ospedale Maggiore Policlinico, 20122 Milan, Italy. [16]Mariani Center for Fragile Children Pediatric Unit, Azienda Socio Sanitaria Territoriale Lariana, 22100 Como, Italy. [17]Department of Laboratories Ospedale Pediatrico Bambino Gesù, IRCCS, 00146 Rome, Italy. [18]Genetic Medicine, Dept of Pediatrics, University of California San Francisco, Ca, Fresno, Ca, San Francisco, CA 94143, USA. [19]Medical Genetics, ASST Papa Giovanni XXIII, 24127 Bergamo, Italy. [20]These authors contributed equally: Giulia Fasano, Valentina Muto, Francesca Clementina Radio. ✉e-mail: antonella.lauri@opbg.net; marco.tartaglia@opbg.net

