## [Peer Review File · Nature Communications]

Dominant ARF3 variants disrupt Golgi integrity and cause a neurodevelopmental disorder recapitulated in zebrafishREVIEWER COMMENTS

Reviewer #1 (Remarks to the Author):

In this impressive manuscript from Fasano, Muto, Radio et al., the authors identify 5 unrelated patients with neurodevelopmental disorders (NDD) with novel, de novo, dominant-acting, ARF3 missense variants.

All of these missense variants correspond to amino acids in or near the guanine nucleotide binding pocket, and similar missense variants in other small GTPases have also been reported to be associated with other diseases.

They postulate that these missense variants would favor the activated (GTP-bound) state of ARF3. They perform MD simulations, investigate some of the variants biochemically, and also investigate the effects of some variants when introduced into zebrafish. They interpret the results of these experiments to mean that the variants induce ARF3 to be constitutively GTP-bound (active).

They perform extensive characterization of the effects on zebrafish development, and find striking parallels to the human patients, providing further validation that the ARF3 variants are the cause of the human NDD disorders.

I think this is an interesting and important study as it provides significant new information about the role of ARF proteins in human disorders.

However, I strongly disagree with the mechanistic interpretations of the molecular effects of the missense variants. As detailed below, I think the authors need to consider that ARFGEF poisoning (stable mutant ARF3-ARFGEF complexes that prevent activation of ARFs 1-5) is a more likely explanation for the disease mechanism than is constitutively active ARF3. The fact that the patient phenotypes appear to strongly resemble those of the previously reported patients with ARFGEF2 variants provides additional support for this idea. The authors are advised to re-write the interpretations and conclusions of their study, otherwise they risk publishing a report postulating an unlikely disease mechanism.

Major points:

1. The *total* amount of *active* ARF3 does not appear to be robustly increased by the three variants tested in Figure 2C. The authors have quantified active ARF3 as a ratio to total ARF3 for each variant, but physiologically one could imagine that the total amount of ARF3, not relative amount, is more important in cells). See also next point.
2. Two of the three tested variants have poor expression and/or are more readily degraded. Taken together, this data does not fit well with the author's interpretation that these variants are causing phenotypes by being hyperactive. Therefore an alternative (and also interesting) dominant-acting mechanism would be that at least some of the five patient variants may be poisoning the ARF-GEFs. If the variants cannot bind nucleotide, this potentially would enable them to bind to their GEFs more strongly, by mimicking the transition state of the GEF reaction (there are many papers showing that nucleotide-free mutants bind to GEFs strongly, and therefore poison them). This mechanism would be physiologically potent, as poisoning the ARFGEF1,2, and GBF1 proteins would affect activation of all the Golgi ARFs (ARF1,3,4, and 5), which share some redundancy of function.
3. The authors find that overexpression of the ARF3 variant K127E in zebrafish induces a significant change in the localization of GalT. To my eye, it appears very similar to the effect of the small molecule BFA, which inhibits the Golgi ARF-GEFs and results in redistribution of GalT and other Golgi enzymes to the ER (for cis/medial enzymes) or other punctate compartments (for TGN enzymes),

based on the work reported in many papers over the years. Notably, such poisoned ARF3-ARFGEF complexes would be expected to localize to the TGN, similar to what happens with BFA treatment, just as the authors observe in Figure 3e. These are additional reasons for thinking that some or all of these patient variants are poisoning the ARF-GEFs rather than acting as constitutively-active mutants.

4. The authors could potentially distinguish between the two possibilities of GTP-locked versus ARFGEF poisoning through additional experiments in which they directly compare the patient mutants to both nucleotide-free ARF mutants and GTP-locked ARF mutants. But I think it would be sufficient to address my concerns through (extensive) re-writing of the interpretations and conclusions of the existing experimental data.

Minor point: The authors should explicitly state near the beginning of the results section that these patient missense variants are found as heterozygous genotypes, to enable the reader to understand early on that these mutations are expected to be dominant. Even though the word "dominant" is in the title, it would be good to include it near the beginning of the results.

Reviewer #2 (Remarks to the Author):

Manuscript NCOMMS-21-25633

Dominantly acting variants in ARF3 have disruptive consequences on Golgi integrity and cause microcephaly recapitulated in zebrafish

Fasano G, et al

Comments to the authors

In this manuscript, Fasano et al. identify 6 de novo missense variants in 5 unrelated patients with neurological abnormalities in ARF3, a gene encoding a member of the Adenosine diphosphate-ribosylation factor from the RAS superfamily of small GTPases. The study aims at validating the pathogenicity of the identified variants. To this end, the authors analyze (i) in silico predictions using 3-dimensional structures of GTP-bound ARF3, (ii) protein stability of the overexpressed variants using western blotting, including GTP-bound state quantification (iii) Golgi apparatus morphology in cells overexpressing the variants. Next, the zebrafish model is used to reproduce the developmental defects, including microcephaly and alterations of the AP axis (as previously reported for ARF1), following microinjection of a subset of mutated ARF3 mRNA. Finally, the author conclude that the disease-associated variants stabilize the ARF3 protein in its active GTP-bound form, causing Golgi disorganization associated with microcephaly and cortical atrophy as well as microsomia and rib abnormalities.

Given the importance of small GTPases in regulating intracellular membrane trafficking, the recent evidence that genetic defects in Golgi-associated factors are associated with neurodevelopmental conditions often including microcephaly (Golgiopathies) and the fact that only ARF1 has been associated with a human disease so far (MIM103180), this manuscript is of potential great significance. However, the study is limited to the functional validation of a subset of variants and does not address the molecular mechanisms through which ARF3 activation cause Golgi dysfunction. In addition, the functional data provided are in my view not sufficiently convincing to fully support the consequences of the variants as presented in the current manuscript. There are also a number of concerns in the way the clinical findings presented are interpreted:

Major comments

Clinical issues

The main concern here is that the features (microcephaly, cortical atrophy, microsomy, rib abnormalities) used to define this new NDD are present in only a subset of patients; MRIs for some patients are missing to illustrate some assertions, and some provided MRIs also do not fully support the features described in Table 1:

1. Neurodegeneration: It is stated page 7, line 166, that the identified ARF3 variants cause neurodegeneration in all five patients. Although cortical atrophy and neurodegeneration are obvious from brain MRI in subject 1 and 2, there is no clear sign of degeneration nor atrophy in MRI from subject 3. Further, Brain MRIs are not provided for subjects 4 and 5 (although the details given in the suppl. Clinical report indicate they do exist) and the clinical data in the main text or in the supplementary information do not mention neurodegeneration. Could the missing MRIs be provided and could the authors explain what evidence they found for neurodegeneration in subjects 3, 4, and 5?
2. Microcephaly: By definition, microcephaly corresponds to brain growth deficit evidenced by a head circumference (OFC) 2 to 3 standard deviations below the mean. According to OFC data provided in table 1, only 3 patients out of five display microcephaly. Subject S3 has a slightly reduced head circumference within the normal range (-1SD) while subject S5 instead has a slightly augmented head circumference, also within normal range (+1.5SD). In such circumstances, microcephaly cannot be considered as a common trait to characterize the syndrome unless the variant identified in S3 and S5 are irrelevant to the disease which would be consistent with the observation that the Pro47Ser mutation does not affect protein stability or GTPase activity (Figure 2). There is something confusing here that the author need to clarify.
3. Ponto-cerebellar hypoplasia is mentioned in Table 1 for subjects S1 and S2 but the size of the cerebellum as shown in Suppl. figure 1 seems normal. Can the authors explain why they talk about cerebellar hypoplasia? Regarding the pons, it is not visible in the sections provided for these 2 patients. Sagittal-medial sections (for example in the same section plan as shown for subject 3, where the pons appears normal) should be provided to appreciate a potential pons hypoplasia.
4. Microsomy: Again, microsomy seems to be obvious only in subjects 1 & 2, but not in the remaining 3 patients. Same comment regarding the skeletal anomalies which are documented only for the first 2 patients.
5. White matter defects and thinning of the corpus callosum seem to be a feature common to all 5 patients. I suggest the authors rather emphasize this point in the description of this NDD.

Functional validation issues

6. Page 10. Line 235. It is claimed that "all residues cluster within or close to the GTP/GDP binding pocket" but it does not appear to be true for Leu12Val. This statement must be corrected and the difference discussed.
7. Using in silico predictions for pathogenicity validation of the genetic variants is an interesting and relevant approach, if further supported by experimental evidence. This sounds of particular importance in the present study where a strong clinical heterogeneity is found in association with various missense variants distributed throughout the protein sequence. Unfortunately, most of the functional experiments (in cell lines or in zebrafish) have been carried out with the variants associated with Subjects S1 and S2 and therefore do not fully allow appreciating whether or not other variants

could trigger milder effects likely to explain the huge clinical differences observed. The biochemical behaviour (GTP-bound status) should be studied for Asp93Asn and Thr32Asn, especially as in the case of the Pro47Ser variant, the mutant protein seems as stable as the control protein (Figure 2a) and does not appear to have a greater affinity to GTP (Figure 2c). Actually, this would be the behaviour expected from a non-pathogenic variant. Alternatively it could be that the Pro47Ser variant result in an activation of ARF3 while preventing its association with GGA3. Therefore, a GTP binding assay should be performed in a system independent of any other protein/interaction to quantify the ability of each variants in binding GTP.

8. The data provided for the pull-down experiments using the GGA3 protein-binding domain are also quite confusing: From figure 2c, we would expect that the amount of myc-tagged ARF3 co-precipitated with GGA3 PBD beads corresponds to a fraction of the myc-tagged protein initially expressed in the total cell lysates. This seems to be the case only for the WT protein and the Pro47Ser variant but not for the other 2 variants. Can the author provide an explanation? How many times has this experiment been repeated? In my opinion, key controls are missing in this assay: a positive control (GTP- γ S) and a negative control (GDP) should be added (for example as shown for ARF1 in doi:

10.1038/npgjgenmed.2016.36 – see figure 5). Constitutively active and inactive controls would also greatly help to improving this assay.

9. Figure 2: Can the authors comment on why the size of ARF3 appears lower in the case of the double variant Leu12Val/Asp67Val? Are there phosphorylated forms of ARF3 explaining this observation?

10. Figure 2: The two “severe variants” Lys127Glu and Leu12Val/Asp67Val seem to be unstable according to western blot data (Figure 2a). How is this compatible with a dominant effect? More generally, there is some concern in validating variants effect only with an over-expression approach. Could the author check the amount of GTP bound ARF3 vs total ARF3 present in patient’s primary cells to demonstrate there is no haploinsufficiency? Alternatively, it could also be easy and relevant, in case no patient cells are available, to create the variants by the CRISPR/Cas technique and evaluate the subsequent effect on GTP affinity.

11. Figure 3 and suppl Fig 5. Golgi fragmentation. Image quality can be improved. Can author provide higher magnifications and single channels in black & white to appreciate the GA fragmentation? It is hard to appreciate the Golgi structure from the images provided in Suppl. figure 5. Some GA appear fragmented even though they do not express ARF3-P47S-mCherry. Again, why only a subset of variants have been tested? Testing a subset of variants would make sense if the clinical consequences were homogeneous, but given the strong heterogeneity, analyzing all the variants appears important. The analysis would be more convincing if constitutively active and inactive ARF3 mutants were used as positive and negative controls.

12. ARF3 has been shown to have diverse effect on the cell cycle (see 10.18388/abp.2020_5519, 10.1016/j.yexcr.2019.111624), a timely regulated mechanism often affected in a large number of congenital microcephaly. Alternatively (or additionally?), ARF3 likely functions in concert with ARF1 to regulate post-Golgi trafficking steps, Golgi integrity and recycling endosomes (as the authors aptly mentioned in the discussion), which have also been associated with postnatal microcephaly. Did the authors measure the cell cycle and/or endosome recycling activity following activation of ARF3? At least can they discuss the potential mechanism of ARF3 activation with regards to the apparition and kind of microcephaly (congenital/postnatal)?

Minor comments

1. Table I: S5 – Speech/language: severely delayed instead of severe delayed
2. Figure 5f: legend is truncated

3. Page 24 line 538: p.Leu12Val/p.Asp67Val correspond to Subject 2, not Subject 3)
4. Page 12 line 284: p.Leu12Val not p.Lys12Val

Reviewer #3 (Remarks to the Author):

This paper reports several human subjects with microcephaly and cortical atrophy who carry de novo ARF3 missense mutations. Based on the previously solved three-dimensional structure of the GTPase, all of mutant residues in ARF3 mutant variants are found to reside within or close to the GTP/GDP binding pocket. By testing the binding affinity of the ARF3 mutant variants ARF3(K127E), ARF3(L12V;D67V) and ARF3(P47S) with GGA3 and effect of their overexpression on location of the trans-Golgi marker GalT or Golgin-97 in COS7 cells as well as by their overexpression effect on zebrafish embryonic development, the authors propose that ARF3(K127E) and ARF3(L12V;D67V) act as dominant hyperactive forms to affect Golgi apparatus integrity and embryonic development. Overall, the findings are valuable for clinical diagnosis. However, the relevant mechanistical insights need more solid evidence.

The authors believed that reduction of ARF3(K127E) and ARF3(L12V;D67V) mutant protein levels when expressed in COS7 cells is caused by reduced stability and accelerated degradation associating with proteasome and autophagy pathway (Fig. 2a and 2b). However, this idea lacks compelling experimental evidence. For example, recovery of ARF3(L12V;D67V) protein levels by the inhibitor treatment was not statistically significant (Fig. 2b). More importantly, current experimental results can't exclude the possibility of contributions from reduced mRNA stability and translation efficiency of the mutants. Thus, mRNA levels should be checked and degradation dynamics of existing proteins in the presence of CHX should be monitored.

In Fig. 2c, the fraction of the active (GTP-bound) form of ARF3 variants was estimated as the GGA3-bound ARF3 fraction. Based on the higher relative percentage of GGA3-bound ARF3 mutants (Fig. 2c, right panel), the authors suggest that these mutants are hyperactive. However, even if the variants bind to GTP more stably, the absolute amount of GFP-bound ARF3 in cells is more important in respect to biological/physiological consequence. Then, in the top panel of Fig. 2c, the amount of GGA3-bound ARF3 proteins should be estimated and shown. As shown in the figure, the amount of GGA3-bound ARF3(L12V;D67V) is obviously lower than that of GGA3-bound ARF3-WT. In this regard, the phenotype observed in the patient carrying the ARF3(L12V;D67V) mutation (Subject 2) may not be caused by hyperactivity of the mutant protein; instead, it may be ascribed to insufficiency of ARF3 activity.

As indicated by Western blots, protein level of ARF3(K127E) or ARF3(L12V;D67V) was drastically decreased compared to ARF3-WT. Why didn't ARF3(K127E)-mCherry (Fig. 3b, 3c, and Fig. S5b) and ARF3(L12V;D67V)-mCherry (Fig. S5c) overexpressed in COS7 show weaker signals than ARF3-WT-mCherry?

In Fig. 3 and Fig. S5, co-localization of ectopically expressed ARF3-WT and mutants with EGFP-GalT was analyzed. Based on some changes in the localization of EGFP-EGFP, the authors conclude that Golgi fragmentation occurred upon overexpression of ARF3 mutants. This conclusion definitely needs more evidence, e.g., from electronic microscopic observation. The mutant proteins may also affect vesicle formation and trafficking, which could be revealed, upon their overexpression, by immunofluorescence of other markers such as the trans-Golgi marker TGN46, the cis-Golgi marker GM130, the COPI complex marker beta-COPI, the early endosomal marker EEA1 (or others), and the endosomal marker RAB7.

The authors assessed effect of Arf3 mutant overexpression in zebrafish embryos and observed pleiotropic effects on embryonic development. The effects are believed to be caused by dominant

function of hyperactivity of the Arf3 mutants. This idea is mostly a speculation only. To test whether the identified Arf3 mutant alleles have a hyperactive or dominant negative activity, their overexpression effect should be compared with that of Arf3 knockout or Arf3 knockdown. Importantly, arf3 knockdown and Arf3 mutant overexpression can be co-performed in zebrafish embryos to test rescue effect. If the Arf3 mutants are truly hyperactive and dominant, their effect on embryonic development should be alleviated by knocking down endogenous arf3a/3b mRNAs.

From the point of view of developmental biologists, it is concerned whether ARF3 mutations affect function of endogenous maternal or zygotic ARF3. It is well known that a single signaling pathway can regulate neural induction and anteroposterior neuroectodermal patterning distinctly. It is possible that maternal and zygotic ARF3 play different roles. This should be tested experimentally.

POINT TO POINT RESPONSE TO REVIEWERS' COMMENTS

General reply to all reviewers: We are glad the reviewers recognize the potential value of our study, which we have extensively revised, following all the recommendations and requests. This was achieved by both generating new datasets for each level of investigation (molecular, cellular, and developmental) and critically re-assessing all the results, the interpretation and the available literature. As a result of this, we now generated 18 new figures and have extensively revised the majority of the originally submitted ones as well as all the sections of the manuscript. We report here a table visually summarizing all the revision work, which we hope can guide the reviewers through the revised manuscript. Below the table, the point-to-point response to each reviewer follows.

Figure (revised version)	Status (reference to the submitted Fig. number)	Title	Description (modifications or new results)
Fig. 1	Revised (submitted Fig. 1)	Structural organization of ARF3, location of mutated residues and molecular dynamics analyses	The hydrogen bond between Asp ⁹³ and Lys ¹²⁷ is now shown d. A new panel (d') is added, which shows the disruption of the bonding caused by p.Asp93Asn.
Fig.2	Revised (submitted Fig. 2)	Expression, stability and GTPase activity of WT and mutant ARF3 proteins in COS-1 cells	New datasets and analyses reporting the stability (including degradation dynamics) and activity for all mutants. GTP-bound fraction is now reported as absolute value (not normalized by ARF3 expression levels, dominant negative (DN) and constitutively active (CA) mutants were used in the same activity assay as reference, cells expressing ARF3 ^{WT} and treated with GDP or GTP were used as internal control.
Fig.3	New figure	ARF3 mutants induce variable Golgi morphological alterations in COS-1 cells	New dataset and analysis showing different classes of trans -Golgi morphology in COS1 cells transfected with all identified ARF3 mutants. DN and CA mutants were used as reference.
Fig.4	Slightly modified (submitted Fig. 3)	Trans -Golgi fragmentation visualized by EGFP-GalT in cells and zebrafish embryos expressing the mutant mCherry-tagged ARF3 ^{K127E}	The schematics was reduced in size, an error in the figure legend was corrected.
Fig. 5	New figure	Variable impact of ARF3 mutations on COP-I vesicles assembly	New dataset and analysis reporting beta-COP staining in COS-1 cells transfected with all identified ARF3 variants and showing a variable impact of the mutations on COP-I assembly.
Fig. 6	New figure	Internalized Tfn accumulates in the perinuclear region and in Rab5 ⁺ endosomes of COS-1 cells expressing ARF3 ^{K127E} and ARF3 ^{L12D/V/D67V}	New dataset and analysis reporting on Tfn cargo recycling (accumulation at perinuclear region and co-localization with Rab5 ⁺ endosomes) in COS-1 cells transfected with all identified ARF3 mutants.

Fig.7	Extensively revised (submitted Suppl. Fig.7)	Expression of mutant ARF3 and downregulation of endogenous arf3 induce distinct phenotypes in zebrafish.	Addition of new in vivo dataset and analyses performed in zebrafish embryos to assess the impact of all ARF3 mutants on embryo development and to compare it with the effect from downregulation of arf3 in zebrafish. The original description of the phenotype is re-assessed and expanded.
Fig. 8	Extensively revised (partial submitted Fig.4)	Occurrence of microcephaly and reduced brain volume in zebrafish expressing a subset of ARF3 mutants	Addition of new in vivo dataset and analyses performed in zebrafish embryos to assess the occurrence of early- and late- onset head phenotyping (as head area and anterior brain volume) in the identified ARF3-mutations. A prospect of human patients' clinical features is now also reported.
Fig. 9	New figure	Increased number of cells in early mitosis and apoptotic within the developing forebrain of zebrafish expressing a subset of ARF3 mutants	New dataset and analyses performed on zebrafish brains for all the identified ARF3 mutants and detailing the cell proliferation and mitosis stages defects within the forebrain as well as the occurrence of apoptotic cells.
Fig.10	Extensively revised (partial submitted Fig.4)	Aberrant ARF3 function causes axial defects with notochord curvatures and defective axes formation in zebrafish	Addition of new datasets and analyses performed in zebrafish for all the identified ARF3 mutants and reporting notochord and AP-ML phenotyping. The original analysis performed to assess the convergence extension (CE) defects was refined. As a result of this we now report the CE index (which more precisely documents the overall defects at the level of gastrulation), instead of ML embryos and somites' quantification.
Suppl. Table 1	Revised (submitted Table 1)	Clinical features of the five subjects with pathogenic ARF3 missense mutations	Re-evaluation of the clinical reports and source data (including MRI) and update of the features reported for each patient.
Suppl. Table 2	Unmodified (submitted Suppl. Table 1)	List of the ARF3 missense mutations identified in the study	-
Suppl. Table 3-7	Unmodified (submitted Suppl. Table 2-6)	WES statistics and data output (subject 1-5)	-
Suppl. Table 8	Unmodified (submitted Suppl. Table 7)	Pathogenic variants in ARF3 paralogs, and HRAS, NRAS and KRAS GTPases affecting codons corresponding and/or adjacent to the residues mutated in ARF3	-
Suppl. Table 9	Unmodified (submitted Suppl. Table 8)	ARF3-GTP hydrogen bonds present for more than 50% of simulation time in the ARF3:COPG1-COPZ1 complex considering ARF3 ^{WT} , ARF3 ^{D67V} and ARF3 ^{P47S}	-

Suppl. Table 10	Unmodified (submitted Suppl. Table 9)	ARF3:COPG1 hydrogen bonds present for more than 40% of simulation time in the ARF3:COPG1-COPZ1 complex considering ARF3 ^{WT} , ARF3 ^{D67V} and ARF3 ^{P47S}	-
Suppl. Table 11	Revised (submitted Suppl. Table 10)	List of primers used in this study	Updated with all the primers used in the study.
Suppl. Table 12	Extensively revised (submitted Suppl. Table 11)	Overview of the statistical analyses	Updated with all the statistical analyses performed in this study (contains re-evaluation of old dataset and analysis of the new dataset produced in revision).
Suppl. Fig.1	Extensively revised (submitted Suppl. Fig. 1)	Clinical features of individuals with de novo ARF3 mutations	Updated with new clinical documentation: the figure now contains MRIs and detailed description from all patients harboring identified ARF3 mutations involved in this study. In addition, available x-rays and images documenting the skeletal abnormalities were added (Subject 1, 2 and 4).
Suppl. Fig. 2	Unmodified (submitted Suppl. Fig 2)	ARF3 multiple sequence alignment and mutations tolerance landscape	-
Suppl. Fig. 3	Unmodified (submitted Suppl. Fig. 3)	2D projection of ARF3 ^{WT} , ARF3 ^{D67V} and ARF3 ^{P47S} on the respective essential subspace along eigenvectors 1 and 2 resulting from MD simulations	-
Suppl. Fig. 4	Revised (submitted Suppl. Fig. 4)	Myc-tagged ARF3 protein levels in COS-1 cells and zebrafish embryos during gastrulation and segmentation stages	Addition of the densitometric quantification of ARF3 protein levels in zebrafish embryos.
Suppl. Fig. 5	New figure	RT-PCR of myc-tagged ARF3 ^{WT} , ARF3 ^{K127E} and ARF3 ^{L12V/D67V} and quantification of total fluorescent intensity of mCherry tagged-ARF3 ^{K127E} , ARF3 ^{L12V/D67V} and ARF3 ^{D93N} upon overexpression in COS-1 cells.	New datasets reporting the mRNA levels of ARF3 ^{L12V/D67V} Compared to WT in transfected COS-1 cells and the fluorescence quantification of mCherry-levels in COS-1 cells transfected with vectors expressing mCherry-tagged ARF3 ^{K127E} , ARF3 ^{L12V/D67V} and ARF3 ^{D93N} .
Suppl. Fig.6	New figure	Nucleotide exchange and GTP hydrolysis measurements of ARF3 ^{WT} , ARF3 ^{K127E} , ARF3 ^{L12V/D67V} , ARF3 ^{D93N} , ARF3 ^{T32N}	New dataset obtained from a cell-free assay on purified ARF3 mutants reporting nucleotide exchange rate and GTP hydrolysis assessment.

Suppl. Fig. 7	New figure	3D rendering of Golgi morphotypes observed in cells expressing ARF3 ^{WT} , ARF3 ^{Q71L} and ARF3 ^{T31N} , relative to Figure 3	Black and white zoom on the confocal 3D rendering (Fig.3) showing the trans-Golgi morphology (Golgin 97 signal) exhibited by DN and CA ARF3 and used as reference for the classification.
Suppl. Fig. 8	New figure	Ultrastructural morphology of Golgi apparatus in COS-1 cells transfected with ARF3 ^{WT} , ARF3 ^{K127E} and ARF3 ^{D93N} revealed by TEM imaging	New dataset obtained by TEM, providing support for difference types of Golgi fragmentation patterns obtained by overexpressing ARF3 ^{K127E} or ARF3 ^{D93N} in COS-1 cells.
Suppl. Fig. 9	New figure	Endogenous arf3a and arf3b paralogs expression during early zebrafish embryogenesis	New in vivo dataset generated from zebrafish embryos' RNA extracts at different stages and reporting the relative levels of arf3a and arf3b mRNA during embryogenesis (before and after maternal to zygotic transition, MZT).
Suppl. Fig.10	New figure	Quantification of trans -Golgi morphology in precursor cells of the animal pole of 6 hpf zebrafish embryos expressing ARF3 ^{WT} , ARF3 ^{K127E} and ARF3 ^{D93N}	New in vivo dataset and analyses performed in early zebrafish gastrulae to evaluate trans -Golgi morphology in a subset of ARF3-overexpressing zebrafish mutants (ARF3 ^{K127E} and ARF3 ^{D93N}). The trans -Golgi morphology in cells was assessed as compared to known CA and DN ARF3 controls (ARF3 ^{Q71L} and ARF3 ^{T31N} , respectively).
Suppl. Fig. 11	New figure	Intracellular Tfn distribution after 5 and 30 minutes of incubation	New dataset and analysis reporting on Tfn intracellular distribution in COS-1 cells transfected with all identified ARF3 mutants and in not-transfected cells and incubated with 5 or 30 min with Tfn.
Suppl. Fig. 12	New figure	Co-localization analysis of Tfn and Rab11 at PN upon 30 min Tfn incubation	New dataset and analysis reporting on Tfn cargo recycling (co-localization with Rab11+ endosomes) in COS-1 cells transfected with all identified ARF3 mutants.
Suppl. Fig.13	New figure	Altered lysosomal trafficking of Tfn upon 30 min incubation in cells expressing ARF3 mutations	New dataset and analysis reporting co-localization of Tfn with Lamp2 marker (lysosome) in COS-1 cells transfected with all identified ARF3 mutants.
Suppl. Fig.14	New figure	Myc-tagged ARF3 protein levels during early zebrafish embryogenesis	New dataset and analysis of performed on zebrafish embryos' protein extracts from fish overexpressing ARF3 ^{WT} and ARF3 ^{K127E} to evaluate the expression profile of the synthetic injected mRNA before and after MZT.
Suppl. Fig. 15	New figure	Incidence of gross phenotypes in fish overexpressing WT and mutant ARF3 at 48 hpf and survival curve of arf3a and arf3b MO-injected fish at 24 and 48 hpf	Quantification plots and survival rate of a subset of the new dataset shown in Figure 7. Assessment of gross phenotype in ARF3-overexpressing zebrafish embryos at 48 hpf is reported and quantification of the survival rate in arf3 morphants for establishing dose-response relationship.

Suppl. Fig. 16	Unmodified (submitted Suppl. Fig. 9)	Anterior brain volume at 48hpf is reduced in zebrafish expressing ARF3 ^{K127E}	-
Suppl. Fig. 17	New figure	Zebrafish embryos expressing ARF3 mutants exhibit thinning of the major tract of the forebrain white matter	New in vivo dataset and analyses performed on zebrafish anterior ventral brain scans reporting the anterior commissure morphological changes in fish expressing ARF3 mutants.
Suppl. Fig. 18	New figure	Ectopic proliferative cells are found within the developing diencephalon of zebrafish expressing ARF3 ^{K127E} and ARF3 ^{D93N} mutants	New in vivo dataset and analyses performed on zebrafish anterior dorsal brain scans reporting proliferative cells number and distribution for all identified ARF3 mutants, with a significant effect for ARF3 ^{K127E} and ARF3 ^{D93N} mutants.
Suppl. Fig. 19	New figure	Impaired spindle morphology of mitotic cells within the developing forebrain of zebrafish embryos expressing ARF3 ^{K127E} and ARF3 ^{D93N}	New in vivo dataset and analyses performed from live time lapse movies of developing forebrain in zebrafish embryos expressing ARF3 ^{WT} , ARF3 ^{K127E} and ARF3 ^{D93N} . Spindle morphology analysis is now reported.
Suppl. Fig. 20	Revised (submitted Suppl. Fig. 10)	Assessment of the number of somites in ARF3 mutants at 15 hpf.	Quantification of the n. of somites on the left and right side of embryos at 15hpf referring to Fig. 10 (data for a batch with all ARF3 mutants are now reported).
Suppl. Fig. 21	Revised (submitted Fig. 6)	Overexpression of ARF3 ^{K127E} and ARF3 ^{D93N} in zebrafish embryos impairs convergence and extension movements during gastrulation	Panels b and c have been updated with images and quantification from additional datasets examining the angle between AP ends as outcome of CE movements in all identified ARF3 mutants.
Suppl. Fig. 22	Unmodified (submitted Suppl. Fig. 11)	Comparison of the FASTA sequences available in the UniProt database and the PDB entry 3TJZ.	-

Reviewer #1 (Remarks to the Author):

In this impressive manuscript from Fasano, Muto, Radio et al., the authors identify 5 unrelated patients with neurodevelopmental disorders (NDD) with novel, de novo, dominant-acting, ARF3 missense variants.

All of these missense variants correspond to amino acids in or near the guanine nucleotide binding pocket, and similar missense variants in other small GTPases have also been reported to be associated with other diseases. They postulate that these missense variants would favor the activated (GTP-bound) state of ARF3. They perform MD simulations, investigate some of the variants biochemically, and also investigate the effects of some variants when introduced into zebrafish. They interpret the results of these experiments to mean that the variants induce ARF3 to be constitutively GTP-bound (active).

They perform extensive characterization of the effects on zebrafish development, and find striking parallels to the human patients, providing further validation that the ARF3 variants are the cause of the human NDD disorders.

I think this is an interesting and important study as it provides significant new information about the role of ARF proteins in human disorders.

However, I strongly disagree with the mechanistic interpretations of the molecular effects of the missense variants. As detailed below, I think the authors need to consider that ARFGEF poisoning (stable mutant ARF3-ARFGEF complexes that prevent activation of ARFs 1-5) is a more likely explanation for the disease mechanism than is constitutively active ARF3. The fact that the patient phenotypes appear to strongly resemble those of the previously reported patients with ARFGEF2 variants provides additional support for this idea. The authors are advised to re-write the interpretations and conclusions of their study, otherwise they risk publishing a report postulating an unlikely disease mechanism.

Reply: we are thankful to the reviewer for recognizing the significance and potential relevance of our work that identifies and functionally validates de novo mutations affecting ARF3 as the cause of a newly recognized neurodevelopmental disease and provides insights on the role of this small GTPase in development and disease. We are glad that the reviewer appreciated the extensive in vivo characterization performed using zebrafish and a model system, which we have further expanded in this revised version of our work. The constructive reviewer's comments and remarks gave us the opportunity to critically re-evaluate the generated data and expand the discussion towards the pathological mechanisms possibly in place. To do so, and to address all the concerns, we have: 1. tested all the identified mutants in vitro and in vivo, 2. performed additional experiments addressing the molecular cellular and developmental impact of the variants using known dominant negative (DN) and constitutively active (CA) controls which guided our interpretation, 3. carried out additional experiments in zebrafish to shed light into the mechanism of disease. Finally, in the revised version of the manuscript, we discuss the resemblance of some of our mutants with the BFA-induced effect, associated with ARFGEF poisoning, as proposed by the reviewer.

Major points:

1. The ***total*** amount of ***active*** ARF3 does not appear to be robustly increased by the three variants tested in Figure 2C. The authors have quantified active ARF3 as a ratio to total ARF3 for each variant, but physiologically one could imagine that the total amount of ARF3, not relative amount, is more important in cells). See also next point.
2. Two of the three tested variants have poor expression and/or are more readily degraded. Taken together, this data does not fit well with the author's interpretation that these variants are causing phenotypes by being hyperactive. Therefore an alternative (and also interesting) dominant-acting mechanism would be that at least some of the five patient variants may be poisoning the ARF-GEFs. If the variants cannot bind nucleotide, this potentially would enable them to bind to their GEFs more strongly, by mimicking the transition state of the GEF reaction (there are many papers showing that nucleotide-free mutants bind to GEFs strongly, and therefore poison them). This mechanism would be physiologically potent, as poisoning the ARFGEF1,2, and GBF1 proteins would affect activation of all the Golgi ARFs (ARF1,3,4, and 5), which share some redundancy of function.

*Reply: we thank the reviewer for the constructive remarks. As the reviewer correctly pointed out, some of the mutations (p.K127E, p.L12V/D67V) causing a severe phenotype are characterized by reduced stability (which we have now documented in more details, reported in **new Fig.2**). Following the reviewer's comment, we now show the absolute GTP-bound fraction of all the mutants, without normalizing for the total amount of the protein (**new Fig. 2**). We apology for the word "hyperactive", which we have mistakenly used in the original version of our manuscript. As correctly stated by the reviewer, a more complex mechanism might underlie the phenotype of ARF3^{K127} and ARF^{L12V/D67V}. We also now clarify that based on the in vitro data a loss-of-function effect for p.K127E, p.L12V/D67V could not be ruled out. To this aim, we performed a number of new experiments aimed to assess the mechanism of action. Among these, we have: tested the nucleotide exchange rate of the successfully purified mutants, showing a clear increase for ARF3^{K127} and ARF3^{D93N} (**new Supplementary Fig. 6**), used detailed Golgi endo-phenotyping in cells and known DN*

and CA ARF1/3 mutants (**new Fig.3**), and compared the effect obtained by overexpressing the mutants to that observed by downregulation endogenous arf3 in zebrafish (**new Fig.7**). Overall, the newly collected data clarified the existence of dominantly acting mutations having a different impact on ARF3 function, and reject the presence of a loss-of function behavior. In addition, in line with the interesting scenario proposed by the reviewer, we now reference and discuss the “GEF poisoning” effect, resembling some of the phenotype we observed.

3. The authors find that overexpression of the ARF3 variant K127E in zebrafish induces a significant change in the localization of GalT. To my eye, it appears very similar to the effect of the small molecule BFA, which inhibits the Golgi ARF-GEFs and results in redistribution of GalT and other Golgi enzymes to the ER (for cis/medial enzymes) or other punctate compartments (for TGN enzymes), based on the work reported in many papers over the years. Notably, such poisoned ARF3-ARFGEF complexes would be expected to localize to the TGN, similar to what happens with BFA treatment, just as the authors observe in Figure 3e. These are additional reasons for thinking that some or all of these patient variants are poisoning the ARF-GEFs rather than acting as constitutively-active mutants.

*Reply: we thank the reviewer for pointing to this similarity. We agree that for some mutants the trans-Golgi appearance resembles the effect induced by BFA. Besides discussing this aspect, we now provide a comprehensive qualitative and quantitative analysis of trans-Golgi morphotypes and COP-1 vesicles formation caused by the mutants (**new Fig.3, revised Fig.4, new Fig.5, new Supp. Fig.7 and new Supp. Fig.8**). The new data on Golgi fragmentation and morphological patterns, together with the GTP binding results, discriminate the different behaviors of the mutants. This point is now discussed in the text.*

4. The authors could potentially distinguish between the two possibilities of GTP-locked versus ARFGEF poisoning through additional experiments in which they directly compare the patient mutants to both nucleotide-free ARF mutants and GTP-locked ARF mutants. But I think it would be sufficient to address my concerns through (extensive) re-writing of the interpretations and conclusions of the existing experimental data.

Reply: following the precious remark, we expanded our discussion on the possible mechanism. Besides re-assessing and re-writing extensively the interpretation and conclusions of our original datasets, the new datasets that we generated with internal controls (DN and CA ARF3) clarify different dominant mechanisms involving the mutants. The data also make it unlikely that these mutations are loss-of-function. We are happy to receive and include more discussion points if needed.

Minor point: The authors should explicitly state near the beginning of the results section that these patient missense variants are found as heterozygous genotypes, to enable the reader to understand early on that these mutations are expected to be dominant. Even though the word “dominant” is in the title, it would be good to include it near the beginning of the results.

Reply: we thank the reviewer for pointing this out. In the original manuscript (and in the revised one) we have indicated that the mutations are “de novo” (mono-allelic) in all the sections, starting from the Abstract. To better guide the reader towards our hypothesized mechanism, we have now paid attention to mention early on in the paper that the mutations are “de novo” and therefore found in heterozygous state, as the reviewer correctly states.

Reviewer #2 (Remarks to the Author):

Manuscript NCOMMS-21-25633

Dominantly acting variants in ARF3 have disruptive consequences on Golgi integrity and cause microcephaly recapitulated in zebrafish

Fasano G, et al

Comments to the authors

In this manuscript, Fasano et al. identify 6 de novo missense variants in 5 unrelated patients with neurological abnormalities in ARF3, a gene encoding a member of the Adenosine diphosphate-ribosylation factor from the RAS superfamily of small GTPases. The study aims at validating the pathogenicity of the identified variants. To this end, the authors analyze (i) in silico predictions using 3-dimensional structures of GTP-bound ARF3, (ii) protein stability of the overexpressed variants using western blotting, including GTP-bound state quantification (iii) Golgi apparatus morphology in cells overexpressing the variants. Next, the zebrafish model is used to reproduce the developmental defects, including microcephaly and alterations of the AP axis (as previously reported for ARF1), following microinjection of a subset of mutated ARF3 mRNA. Finally, the author conclude that the disease-associated variants stabilize the ARF3 protein in its active GTP-bound form, causing Golgi disorganization associated with microcephaly and cortical atrophy as well as microsomy and rib abnormalities.

Given the importance of small GTPases in regulating intracellular membrane trafficking, the recent evidence that genetic defects in Golgi-associated factors are associated with neurodevelopmental conditions often including microcephaly (Golgiopathies) and the fact that only ARF1 has been associated with a human disease so far (MIM103180), this manuscript is of potential great significance. However, the study is limited to the functional validation of a subset of variants and does not address the molecular mechanisms through which ARF3 activation cause Golgi dysfunction. In addition, the functional data provided are in my view not sufficiently convincing to fully support the consequences of the variants as presented in the current manuscript. There are also a number of concerns in the way the clinical findings presented are interpreted:

Reply: we thank the reviewer for recognizing the potential significance of our study. Our original aim was to share the finding of ARF3 as a new disease-causing gene implicated in a new neurodevelopmental disorder and link this human disease to disruption of Golgi integrity, supported by in vitro and in vivo validation data. Based on the reviewer feedback, we realize that only partially fulfill our goal, and agree that a complete characterization of all the discovered variants provides a better validation of their functional relevance and insights on the underlying mechanism and allows a better understanding of the clinical heterogeneity observed in the patients. Thereby, to address the reviewer's criticism on the limited coverage of our initial study, we now expand our in vitro and in vivo experimental validation to all the identified ARF3 variants. Specifically, to ensure a comprehensive picture of the molecular, cellular, and developmental alterations exhibited by all mutations and test different mechanisms of action, we performed new experiments addressing the following aspects: 1. mutants' molecular behavior in terms of nucleotide binding in cells by performing a direct comparison with known DN and CA ARF1/3 variants; 2. nucleotide exchange and GTP hydrolysis of purified proteins; 3. effect on Golgi integrity in cells and zebrafish; 4. cargo recycling ability; 5. phenotype obtained by overexpression ARF3 mutants or downregulating arf3 in fish. Moreover, new experiments on all mutants and including new levels of investigation expand the brain developmental alterations observed with respect to growth and proliferative potential and provide insights into the occurrence and underlying cause of body axes malformations. Finally, to address the reviewer's concerns on the insufficient data supporting the heterogeneity of the clinical aspects, we attempted our best to gather the available clinical information for all the patients. As a result, in the revised version of the manuscript, we now expand the patients' clinical reports and brain imaging data, detailing the observed heterogeneity, and provide a newly revised Supp. Fig.1, revised Table 1 (now supplementary) and revised Supp.Clinical reports. Nevertheless, we would like to point out to the reviewer the genuine limitation in expanding further the display of clinical data beyond what presented now. As the reviewer certainly understands, such limitation is due to the severity of most of the cases shown and to the burden associated with the diagnostic ordeal. This rightly represents a limitation to the possibility pursue additional patients' consultations, instrumental examination and/or retrieval of biological material (e.g. biopsy).

Major comments

Clinical issues

The main concern here is that the features (microcephaly, cortical atrophy, microsomy, rib abnormalities) used to define this new NDD are present in only a subset of patients; MRIs for some patients are missing to illustrate some assertions, and some provided MRIs also do not fully support the features described in Table 1:

1. Neurodegeneration: It is stated page 7, line 166, that the identified ARF3 variants cause neurodegeneration in all five patients. Although cortical atrophy and neurodegeneration are obvious from brain MRI in subject 1 and 2, there is no clear sign of degeneration nor atrophy in MRI from subject 3. Further, Brain MRIs are not provided for subjects 4 and 5 (although the details given in the suppl. Clinical report indicate they do exist) and the clinical data in the main text or in the supplementary information do not mention neurodegeneration. Could the missing MRIs be provided and could the authors explain what evidence they found for neurodegeneration in subjects 3, 4, and 5?

Reply: following the reviewer's advice, we have now collected and provide the missing MRI scans for all the subjects (**revised Supplementary Fig. 1**). We agree with the reviewer that S1 and S2 show overlapping severe features. In the revised version of the manuscript, we now explain in more in details the clinical traits and the differences observed. As indicated in the **revised Supplementary Figure 1 and Table 1**, both patients are characterized by severe and generalized cortical atrophy and ventricular dilatation, severe corpus callosum and brainstem hypoplasia, particularly affecting the pons. The cerebellar hemispheres are spared by the atrophy, the inferior vermis however is partially involved, and the resulting surface of the 4th ventricle is considerably enlarged. We would point out that atrophy progression is observed in S2 between 3 and 14 months indicative of a possible neurodegenerative mechanism at the basis of the evolutive neurological findings. However, we agree with the reviewer that there is no sufficient information showing neurodegenerative nature of the disease in all the patients and we have amended the text accordingly, removing the feature "neurodegeneration" from the description of the clinical traits of the disease.

2. Microcephaly: By definition, microcephaly corresponds to brain growth deficit evidenced by a head circumference (OFC) 2 to 3 standard deviations below the mean. According to OFC data provided in table 1, only 3 patients out of five display microcephaly. Subject S3 has a slightly reduced head circumference within the normal range (-1SD) while subject S5 instead has a slightly augmented head circumference, also within normal range (+1.5SD). In such circumstances, microcephaly cannot be considered as a common trait to characterize the syndrome unless the variant identified in S3 and S5 are irrelevant to the disease which would be consistent with the observation that the Pro47Ser mutation does not affect protein stability or GTPase activity (Figure 2). There is something confusing here that the author need to clarify.

Reply: as correctly stated by the reviewer, only a subset of the mutations we described causes microcephaly (early onset for S1 and S2 and late onset for S4). As described, S3 shows a slightly reduced head size. We have now carefully re-examined the available clinical documentation and we rectify the information provided in the **revised Supplementary clinical reports and Supplementary Table 1** (the examination at 30 m is now reported, which shows an OFC of -1.44 SD for S3). In addition, as discussed, MR1 scans now available for S3 show severe hypoplasia of the anterior part of the temporal lobes (**revised Supplementary Fig.1**). To clarify the impact of the p.P47S substitution on development and address the reviewer's request, besides assessing its impact on cells, we have now generated fish overexpressing the mutant. Judging from all the functional dataset, we consider the p.P47S to have a mild pathogenic impact, in agreement with the clinical reports for S3. In particular, we would like to point out that the *in vivo* data collected for this variant supports its mild impact on development and brain formation. Indeed, recapitulating the scenario of S3, despite no clear sign of microcephaly (gross phenotype) was observed in fish expressing ARF3^{P47S}, a more detailed 3D imaging analysis on brain volume showed a significant reduction in the anterior brain mass. More generally, we believe that the newly provided clinical records together with the newly generated dataset in zebrafish, which fully recapitulate the patients' features, provide solid evidence for the differential impact of the mutations in terms of severity of brain and body axis phenotype, thereby answering in full the reviewer's questions and concerns. Of note, substantiating the involvement of mutant ARF3 with brain development, we recently identified another subject carrying a *de novo* ARF3 missense change (c.376A>G, p.Asn126Asp) affecting a residue adjacent to Lys127 (S1). Consistently, this patient shows clinical features overlapping with those observed in S1 and S2, including microcephaly. After careful consideration, given the late notice of this novel case (at the end of this revision process) and from here the impracticality of performing functional validation of sufficient solidity within the timeframe of the revision, we decided not to include this new patient in the revised manuscript. However, we are open to include the clinical characterization of the patient in case the reviewers and the Editor consider these new data relevant for the overall message of our work. In addition, we would like to point out to the reviewer that during this revision an additional work reporting two mutations in ARF3 was published, substantiating the role of this small GTPase in neurodevelopment (Sakamoto et al., 2022, doi: 10.1093/hmg/ddab224.), which we now discuss in more details in the revised manuscript. Unfortunately, the relatively limited analysis performed in Sakamoto et al., 2022 (especially with respect to the *in vivo* model) does not allow for an extensive functional comparison with our findings. Nevertheless, the severe case reported in Sakamoto et al., 2022 shows the same p.Asp57Val (D67V) in ARF3 observed in S2 (reported here in cis with p.Leu12Val), with overlapping traits (DD, severe microcephaly, progressive cerebral and brain stem atrophy and epilepsy) and this mutation shows similar behavior with respect to GTP binding and Golgi pattern.

3. Ponto-cerebellar hypoplasia is mentioned in Table 1 for subjects S1 and S2 but the size of the cerebellum as shown in Suppl. figure 1 seems normal. Can the authors explain why they talk about cerebellar hypoplasia? Regarding the pons, it is not visible in the sections provided for these 2 patients. Sagittal-medial sections (for example in the same section plan as shown for subject 3, where the pons appears normal) should be provided to appreciate a potential pons hypoplasia.

Reply: following the reviewer's suggestion, we provide more documentation and description related to the clinical manifestation of the patients in the revised version of the manuscript. In particular, MRI scans, including sagittal sections (as requested by the reviewer) of all the patients are now provided (**revised Supplementary Fig 1**). A visibly reduced pons can be observed in S1 and S2 (yellow circle in **revised Supplementary Fig 1**). As far as the cerebellum is concerned, the vermis is involved in S1 and the resulting surface of the 4th ventricle is considerably enlarged (black asterisk in the axial images for S1, S2 in **revised Supplementary Fig.1**).

4. Microsomy: Again, microsomy seems to be obvious only in subjects 1 & 2, but not in the remaining 3 patients. Same comment regarding the skeletal anomalies which are documented only for the first 2 patients.

Reply: we thank the reviewer for noticing these inconsistencies in the original report. We agree with the reviewer that severe microsomy is only evident in S1 and S2, while S3-5 show a stature within the normal range, as indicated in the revised **Supplementary table 1**. We have amended the text accordingly. As far as the skeletal abnormalities are concerned, the reviewer is correct that major defects are again observed only in S1 and S2. Addressing the reviewer's concern about the lack of documentation in the submitted work, we now also provide the available x-ray records for S1 and S2 that we gathered during the revision, documenting beyond doubt the striking skeletal abnormalities (**revised Supplementary Fig.1c**). In addition, we would point out that a milder involvement of the skeleton (i.e., kyphotic spine) was also reported in S4 and S5. More, precisely, as already indicated in the originally submitted Table 1 and also in the **revised Supplementary Table 1**, S4 shows pectus excavatum (for which a picture was available and is now included in the **revised Supplementary Fig.1c**). S5 displays a gibbus deformity involving the thoracic vertebrae, reported in the Table.

5. White matter defects and thinning of the corpus callosum seem to be a feature common to all 5 patients. I suggest the authors rather emphasize this point in the description of this NDD.

Reply: as the reviewer correctly states, white matter defects and thinning of the corpus callosum are common feature of the NDD described here, as reported in the original Table 1. Following the reviewer's suggestion, we now stress out this aspect in the revised text. Importantly, supporting the correct evaluation of the reviewer, we validated the occurrence of analogous defects also in our zebrafish models, common for all mutants and more severe for p.K127E and p.L12V/D67V (S1 and S2, respectively). These data are now presented in the **new Supplementary Fig 17** and corroborate the thinning of commissural nerve tracks in the anterior brain (forebrain) as a common feature of this NDD.

Functional validation issues

6. Page 10. Line 235. It is claimed that "all residues cluster within or close to the GTP/GDP binding pocket" but it does not appear to be true for Leu12Val. This statement must be corrected and the difference discussed.

Reply: we apologize with the reviewer for our inaccuracy, that we have failed to spot in our final proofreading before submission and are grateful to the reviewer for noticing it. Indeed, as the reviewer correctly states our considerations were limited to the amino acid substitutions considered as functionally relevant. We have now amended the text to correct for this mistake and we now state: "Microcephaly-associated ARF3 variants affect residues within the guanine nucleotide binding pocket" (**Abstract**), "All residues except for Leu¹² cluster within or close to the GTP/GDP binding pocket" (**Results, II paragraph**). We now also clarify that Leu12 is not present in the crystal used to predict the possible effect of the mutants on the protein's activity and we discuss in more details the location of Leu12 within the N-terminal, which is myristoylated, a modification that can impact ARF3 membrane binding and thereby activity.

7. Using in silico predictions for pathogenicity validation of the genetic variants is an interesting and relevant approach, if further supported by experimental evidence. This sounds of particular importance in the present study where a strong clinical heterogeneity is found in association with various missense variants distributed throughout the protein sequence. Unfortunately, most of the functional experiments (in cell lines or in zebrafish) have been carried out with the variants associated with Subjects S1 and S2 and therefore do not fully allow appreciating whether or not other variants could trigger milder effects likely to explain the huge clinical differences observed.

Reply: following the constructive reviewer's remark, we performed in vitro and in vivo functional studies at different levels (molecular, cellular and organismal) for all the variants identified in the reported patients. As the reviewer correctly anticipated, altogether our results (for a total of 18 new figures) resolve the differential impact of the mutations with respect to organelle stability and vesicle assembly, cargo recycling. In addition, presence of microcephaly, occurrence of early vs. late microcephaly, brain and axis altered formation can be also fully appreciated in the fish model and support the presence clinical variability.

The biochemical behaviour (GTP-bound status) should be studied for Asp93Asn and Thr32Asn, especially as in the case of the Pro47Ser variant, the mutant protein seems as stable as the control protein (Figure 2a) and does not appear to have a greater affinity to GTP (Figure 2c). Actually, this would be the behaviour expected from a non-pathogenic variant. Alternatively it could be that the Pro47Ser variant result in an activation of ARF3 while preventing its association with GGA3. Therefore, a GTP binding assay should be performed in a system independent of any other protein/interaction to quantify the ability of each variants in binding GTP.

Reply: as requested by the reviewer, we have tested all the mutants, including p.D93N and p.T32N at the molecular, cellular and developmental level. Additional experiments in cells were performed, which confirm a trend of increased GTP binding for p.P47S (revised Figure 2). We would like to point out to the reviewer that in the submitted version of this manuscript we have assessed the possible structural perturbations of all the mutants, including p.P47S. We agree with the reviewer that a different mechanism of action could be associated to the pathogenicity of this variant. Indeed, already in the submitted version of the paper, we reported that Pro⁴⁷ is located within the surface portion of switch 1 region, which by GTP binding is known to change conformation and mediate binding to effectors. We have also performed molecular dynamics simulations starting from the known interaction of ARF1 with COP complex, which, as reported in the submitted version of this manuscript, showed a dramatic perturbation of the intermolecular binding network with COPG1 due to a substantial rearrangement of the Switch 1 (revised Fig.1e h, supplementary Fig. 3). Instead, no major effect was observed in terms of GTP binding (Supplementary Table 9), which supported the existence of a different pathogenic mechanism for this variant. Hence, as stated in the Results, while our "...structural analyses predicted that all variants but p.Leu12Val affect ARF3 GTP/GDP binding and/or the GTPase activity, a more articulated impact on conformational rearrangements mediating binding to effectors was suggested for p.Pro47Ser...". Adding to this, the newly generated dataset on the endophenotype analyzed at the level of the Golgi and COP-I vesicle assembly demonstrate a clear effect of this variant, which was significantly different from the WT controls. p.P47S showed a behavior similar to the known CA (p.Q71L, used as control for Golgi morphology, revised Figure 3, 4) and to p.D93N, also classified as CA. In addition, we also observe mild effects of this variants in fish embryos. Specifically, we observed significant changes compared to ARF3^{WT} in terms of: gross embryos phenotype (even if mild, revised Fig.7c''), brain size, white matter defects and cell death (revised Fig.8,c', Figure 9g, Supplementary Fig.17), as well as reduced antero-posterior length (Supplementary Figure 10f). In the revised version of the manuscript, we discuss these findings and the mild impact of this variant. In addition, following the suggestion from the reviewer, we have teamed up with expert colleagues to perform also a biochemical assessment of purified WT and mutant ARF3 proteins in a cell-free system (revised Supplementary Fig.6). Please note that we could only test the proteins for which we obtained substantial protein purification within the time-frame of this revision (K127E, L12V/D67V, D93N, T32N). Thereby the behavior of p.P47S could not be assessed and will be investigated in the near future. Our results so far show an increased nucleotide exchange for K127E and D93N, which was GEF independent while L12V/D67V showed a reduced exchange rate and for all the variants analyzed no major effect was observed in terms of GTP hydrolysis. We now also discuss that similar effects (in terms of nucleotide exchange and GTP hydrolysis) were reported for pathogenic RAS gene mutations implicated in RASopathies. We would like to point out to the reviewer that, as we now discuss in the revised manuscript, while the data provide support for differential behavior of the tested proteins in terms of nucleotide exchange, they are not conclusive per se because of the limitation of the cell-free system used. Myristoylation could not be achieved in our expression system, which also lacked the membrane and lipids environment, which is of functional relevance for the GTP/GDP exchange and activation of these small GTPases. Thereby, we reasoned that we might overcome the limitations of these assays for assessing the protein behavior and pathogenicity by analyzing the above-mentioned cellular and developmental phenotypes linked to the protein's function obtained by each mutant; possibly in direct comparison with known DN and CA ARF mutants. This alternative functional approach matched the requests by all the reviewers, and we believed provided a solid ground for mechanistic interpretation of the alterations caused by ARF3 mutants. Indeed, altogether, while supporting the presence of DN and CA acting ARF3 variants, the functional analysis performed during this revision, also rejects the hypothesis of non-pathogenicity for the variant p.P47S.

8. The data provided for the pull-down experiments using the GGA3 protein-binding domain are also quite confusing: From figure 2c, we would expect that the amount of myc-tagged ARF3 co-precipitated with GGA3 PBD beads corresponds to a fraction of the myc-tagged protein initially expressed in the total cell lysates. This seems to be the case only for the WT protein and the Pro47Ser variant but not for the other 2 variants. Can the author provide an explanation? How many times has this experiment been repeated? In my opinion, key controls are missing in this assay: a positive control (GTP- γ S) and a negative control (GDP) should be added (for example as shown for ARF1 in doi: 10.1038/npgenmed.2016.36 – see figure 5). Constitutively active and

inactive controls would also greatly help to improving this assay.

Reply: we apologize for the lack of clarity. While correct in principle, the reviewer's remark does not take into consideration that different amount of input total proteins was used for the experiments. As we reported in the Methods section, 15 µg of total proteins from cell lysates were loaded for WB analysis, while 1 mg of total proteins was used for the co-IP experiments and the whole amount of immunoprecipitate is used for the subsequent WB assay. In the experiment, the WB analysis using whole lysates is performed to compare the relative amount of ARF3 proteins. The experiment was repeated at least 3 times in the course of the revision. To reply to the reviewer request and conform with Nature standards, we now show the replica (column graph with experimental replicates shown (dots), mean and SEM) in **revised Fig.2**. In addition, we followed the suggestion of the reviewer and, similarly to the paper cited, we now carried out the same experiment by using internal controls (GTPy-S and GDP) on the WT protein, which confirms the validity of the assay. Following the additional recommendation of the reviewer we improved the interpretation of the assay by also adding known DN and CA mutants (p.T31N and p.Q71L, respectively). We are thankful to the reviewer for these precious comments, which helped us to start to resolve different functional classes among the identified mutants.

9. Figure 2: Can the authors comment on why the size of ARF3 appears lower in the case of the double variant Leu12Val/Asp67Val? Are there phosphorylated forms of ARF3 explaining this observation?

Reply: we have explained the reason for the shift of the band in our submitted manuscript (legend of the **Supplementary Figure 4**). We demonstrated that the shift is caused by D67V and not L12V as we have tested their gel run separately. We believe that the different run depends upon to the reported phenomenon of "gel shifting" (Shi et al., 2012) caused by an increased binding to SDS molecules and a change of the electrophoretic mobility. Of note, this running behavior of ARF3^{D67V} was confirmed also in a different laboratory who used different expression systems and tags (see Sakamoto et al., 2021). We are not aware of phosphorylation of ARF3. We are however open if the reviewer considers additional speculations on the cause of the shift, but we believe that demonstrating the actual hypothesis is beyond the scope of this paper. We are also open to change the position of the explanation in other parts of the main text if the reviewer considers it useful.

10. Figure 2: The two "severe variants" Lys127Glu and Leu12Val/Asp67Val seem to be unstable according to western blot data (Figure 2a). How is this compatible with a dominant effect? More generally, there is some concern in validating variants effect only with an over-expression approach. Could the author check the amount of GTP bound ARF3 vs total ARF3 present in patient's primary cells to demonstrate there is no haploinsufficiency? Alternatively, it could also be easy and relevant, in case no patient cells are available, to create the variants by the CRISPR/Cas technique and evaluate the subsequent effect on GTP affinity.

Reply: we thank the reviewer for this valid comment. The reviewer is correct that the levels of p.K127E and p.L12V/D67V are unstable. In the revised version of the manuscript we provide confirmation by additional experiment of the accelerated degradation of these proteins. Considering our data in vitro and in zebrafish, also using MO approach to silence endogenous arf3 expression (as suggested by the reviewer 3) we reject the possibility of a loss-of-function effect of the single ARF3 proteins. A complex dominant behavior might be in place here, which might affect the activity of different ARF proteins altogether (see the comment of the reviewer 1 on "GEF poisoning" effect). We now discuss the resemblance with the cellular phenotype obtained by BFA treatment and DN mutants of our two variants. We agree that a study on primary cells from patients would add valuable information for the pathogenicity of these variants, however, these were not available in the frame of this work and revision. However, as the reviewer and the editor would agree, given the severe burden associated with the disease and in absence of a true need for patientcare, we regretted to invite patients and families involved to try to collect bioptic material. This would be against the signed ethical obligations and standards. While the suggestion to generate the variants by CRISPR/Cas technique is a valuable suggestion and we certainly are setting up tools and protocols to this goal, it is currently beyond the scope of this paper. We would like to point out also that pitfalls (for instance due to off-targets effects) exist also for this methodology, which necessitates careful validation of the data obtained.

11. Figure 3 and suppl Fig 5. Golgi fragmentation. Image quality can be improved. Can author provide higher magnifications and single channels in black & white to appreciate the GA fragmentation? It is hard to appreciate the Golgi structure from the images provided in Suppl. figure 5. Some GA appear fragmented even though they do not express ARF3-P47S-mCherry. Again, why only a subset of variants have been tested? Testing a subset of variants would make sense if the clinical consequences were homogeneous, but given the strong heterogeneity, analyzing all the variants appears important. The analysis would be more convincing if constitutively active and inactive ARF3 mutants were used as positive and negative controls.

Reply: we thank the reviewer for these valuable suggestions. Following all the recommendations, we have expanded and repeated the experiment addressing Golgi morphotypes for all mutants and we have included known DP and DN ARF mutants. We qualitatively and quantitatively characterized the Golgi integrity alterations, and we now report on the differential impact of the mutations on Golgi morphology (**new Fig.3**). As the reviewer correctly states, p.P47S has a mild CA phenotype also in the new dataset. As requested, we also provide black and white images of the different Golgi morphotypes that we have observed in our mutants (compact, expanded, fully or partially dispersed) in revised **new Fig. 7**). The overall loss of Golgi WT morphology and the presence of cells with dispersed or expanded phenotype was also confirmed in fish expressing ARF3^{K127E} and ARF3^{D93N} (**new Supplementary Fig.10**). In addition, to provide more details on the observed Golgi phenotype we also performed a TEM analysis of the mutant ARF3^{K127E} and ARF3^{D93N}, which showed clearly distinguishable patterns of Golgi morphological alterations in our cell experiments. Indeed, TEM results confirm the presence of two types of Golgi loss of integrity (**new Supplementary Fig.8**). The classical ribbon-like organization was not present in the mutants, and while ARF3^{K127E} exhibited loss of the mini-stacks with numerous diffused vesicles and small cisternae scattered in a wide area (**Supplementary Figure 8 b,b'**), cells expressing ARF3^{D93N} displayed a different pattern with marked increase in swollen cisternae and a diffuse vesiculation, within the defined area normally occupied by Golgi (**Supplementary Figure 8 c,c'**). A similar fragmentation pattern had previously been described for CA ARF mutants doi: 10.1083/jcb.124.3.289., doi: 10.1074/jbc.M611716200.).

12. ARF3 has been shown to have diverse effect on the cell cycle (see 10.18388/abp.2020_5519, 10.1016/j.yexcr.2019.111624), a timely regulated mechanism often affected in a large number of congenital microcephaly. Alternatively (or additionally?), ARF3 likely functions in concert with ARF1 to regulate post-Golgi trafficking steps, Golgi integrity and recycling endosomes (as the authors aptly mentioned in the discussion), which have also been associated with postnatal microcephaly. Did the authors measure the cell cycle and/or endosome recycling activity following activation of ARF3? At least can they discuss the potential mechanism of ARF3 activation with regards to the apparition and kind of microcephaly (congenital/postnatal)?

Reply: We are grateful to the reviewer for the suggestion to investigate or discuss the possible impairment of cell cycle and recycling caused by our ARF3 mutants. We chose the developing brain in our zebrafish models – which recapitulate the microcephaly and white matter defects observed in a subset of patients- to directly assess alterations in the dividing cells within the anterior forebrain, possibly explaining the phenotype observed. We employ specific markers of proliferation and mitosis, and we demonstrate impairment of the proliferative behavior and an increased presence of progenitors in the pro/prometaphase in a subset of the mutants. An increased number of apoptotic cells was also quantified (**new Fig. 9, new supplementary Fig.18**). Aberrant spindle formation within the forebrain was also documented in ARF3^{K127E} and ARF3^{D93N} in live experiments with a dedicated fish line labeling microtubule (**new Supplementary Fig 19**). As the reviewer correctly states, these mechanisms are known causes of microcephaly and we now expand on this in the discussion. On the other hand, we chose to employ the established overexpression in vitro system to study recycling, which is more suitable to trafficking analysis and which also address Reviewer 3's request to test different markers along the endocytic-lysosomal pathway (**new Fig.6, new supplementary Fig. 11-13**). Our new dataset indicates an overall delayed cargo recycling for the strong mutants (p.K127E and p. L12V/D67V), with accumulation of Tnf within the perinuclear endosome compartments and an increased targeting to lysosomes for p.D93N. In all mutants, Tfn lysosomal loading was increased. As stated by the reviewer, the involvement of post-Golgi trafficking in NDD with microcephaly is emerging. Similarly, Carpentieri *et al.*, recently demonstrated that Tfn cargo delay in Rab5+ endosomes contribute to microcephaly (Carpentieri, J.A., Di Cicco, A., Lampic, M. *et al.* Endosomal trafficking defects alter neural progenitor proliferation and cause microcephaly. **Nat Commun 13, 16 (2022)**. <https://doi.org/10.1038/s41467-021-27705-7>).

Minor comments

1. Table I: S5 – Speech/language: severely delayed instead of severe delayed
2. Figure 5f: legend is truncated
3. Page 24 line 538: p.Leu12Val/p.Asp67Val correspond to Subject 2, not Subject 3)
4. Page 12 line 284: p.Leu12Val not p.Lys12Val

Reply: we thank the reviewers for noticing these typos, which we have corrected.

Reviewer #3 (Remarks to the Author):

This paper reports several human subjects with microcephaly and cortical atrophy who carry de novo ARF3 missense mutations. Based on the previously solved three-dimensional structure of the GTPase, all of mutant residues in ARF3 mutant variants are found to reside within or close to the GTP/GDP binding pocket. By testing the binding affinity of the ARF3 mutant variants ARF3(K127E), ARF3(L12V;D67V) and ARF3(P47S) with GGA3 and effect of their overexpression on location of the trans-Golgi marker GalT or Golgin-97 in COS7 cells as well as by their overexpression effect on zebrafish embryonic development, the authors propose that ARF3(K127E) and ARF3(L12V;D67V) act as dominant hyperactive forms to affect Golgi apparatus integrity and embryonic development. Overall, the findings are valuable for clinical diagnosis. However, the relevant mechanistical insights need more solid evidence.

Reply: we are glad that the reviewer acknowledges the value of our findings with regards to improving diagnostic resolution of unknown neurodevelopmental diseases. We are also thankful for the comments and suggestions that, together with the valuable inputs from the other reviewers, guided our new experiments and a re-evaluation of all the data obtained with the goal to refine the description and discussion of the possible pathological mechanisms in place. We performed additional experiments to provide more solid evidence of the mechanisms and we could now distinguish differential impacts of the all the variants identified on nucleotide binding, Golgi and vesicles integrity, recycling and zebrafish embryogenesis, focusing on brain and axes formation.

The authors believed that reduction of ARF3(K127E) and ARF3(L12V;D67V) mutant protein levels when expressed in COS7 cells is caused by reduced stability and accelerated degradation associating with proteasome and autophagy pathway (Fig. 2a and 2b). However, this idea lacks compelling experimental evidence. For example, recovery of ARF3(L12V;D67V) protein levels by the inhibitor treatment was not statistically significant (Fig. 2b). More importantly, current experimental results can't exclude the possibility of contributions from reduced mRNA stability and translation efficiency of the mutants. Thus, mRNA levels should be checked and degradation dynamics of existing proteins in the presence of CHX should be monitored.

*Reply: to address the reviewer's concern and investigate the reduced protein levels originally observed in the WB analysis, we performed additional experiments, following the suggestions provided. As requested, we first conducted experiments in presence of CHX (revised Fig.2a), which corroborated our initial interpretation that the ARF3^{K127E} and ARF3^{L12V/D67V} mutants have a reduced stability within cells and are rapidly degraded upon transduction. Next, to exclude possible effect at the level of mRNA, we performed RT-PCR against ARF3 in COS-1 cells transfected with the construct expressing ARF3^{L12V/D67V}, as a representative of poorly stable mutant. We also included ARF3^{D93N}, which did not show relevant changes in the WB analysis. These data are now presented in **new Supplementary Fig 5** and confirm that the low protein levels are due to protein degradation for p.L12V/D67V as no change in mRNA levels are observed for this variant, similarly to p.D93N and ARF3^{WT}. In addition, with respect to the mechanism responsible for protein degradation, and addressing the reviewer's concern, we have re-evaluated the original dataset and repeated the experiment with the inhibitors MG132 and Bafilomycin. The new dataset confirms the trend observed in the originally submitted work and is now included in the final graph in the **revised Fig. 2a'**. As the correctly stated by the reviewer, it is true that, comparing the protein levels among the conditions, the observed rescue of ARF3^{L12V/D67V} mutant did not reach statistical significance upon addition of the proteasome inhibitor. Nevertheless, it is reasonable to think that the lack of a rescue comparable to the levels of the WT protein, may be due to the especially fast and substantial degradation of the mutant p.L12V/D67V already at time 0 (just 24hpf upon DNA transfection) before the start of the treatment with the inhibitors. Given this circumstance, in order to rule out if the mutant ARF^{L12V/D67V} is degraded by protostomes or autophagy, we directly calculated the fold change (FC) of the ARF^{L12V/D67V} protein levels after treatment with Mg132 as compared to its basal levels before treatment (**revised Fig.2 a'**). The FC analysis bypasses the fundamental differences in protein levels between the conditions at time 0 and confirms the involvement of both protein degradation pathways for p.K127E and p.L12V/D67V mutant.*

In Fig. 2c, the fraction of the active (GTP-bound) form of ARF3 variants was estimated as the GGA3-bound ARF3 fraction. Based on the higher relative percentage of GGA3-bound ARF3 mutants (Fig. 2c, right panel), the authors suggest that these mutants are hyperactive. However, even if the variants bind to GTP more stably, the absolute amount of GFP-bound ARF3 in cells is more important in respect to biological/physiological consequence. Then, in the top panel of Fig. 2c, the amount of GGA3-bound ARF3 proteins should be estimated and shown. As shown in the figure, the amount of GGA3-bound ARF3(L12V;D67V) is obviously lower than that of GGA3-bound ARF3-WT. In this regard, the phenotype observed in the patient carrying the ARF3(L12V;D67V) mutation (Subject 2) may not be caused by hyperactivity of the mutant protein; instead, it may be ascribed to insufficiency of ARF3 activity.

Reply: given the complex behavior of the p.K127E and p. L12V/D67V we understand the reviewer's concern and we apology for the mis-used of "hyperactive", as already replied to reviewer 1. As requested, we now consider solely the absolute GTP-bound state of the variants (**revised Fig.2b,c**) and we have amended the revised manuscript accordingly. Following the reviewer's suggestion, and integrating also the requests from reviewers 1 and 2, we have performed additional experiments to further clarify the genetic mechanisms of all the ARF3 mutations described. First, we performed GTP binding assays from cell lysates for all the mutants in direct comparisons with known CA and DN controls. We now quantify the absolute amount of the GTP-bound ARF3, irrespectively of the total protein levels, as suggested by the reviewer. In addition, and to reply also to the similar concerns raised by reviewer 1 and 2, within the time frame of the revision we have conducted also biochemical investigation of the purified WT and mutant ARF3 proteins with respect to their ability to perform nucleotide exchange and GTP hydrolysis (**new Supplementary Fig. 6**). We now also provide an in-depth characterization of the impact of all the mutants on Golgi stability in cells, using known CA and DN (**new Fig.3, 4, Supplementary Fig.7,8,10**), which helped us to classify the mutations in different categories with respect to their cellular effect. Lastly, as far as the concern of possible protein insufficiency for ARF3^{K127E} and ARF3^{L12V/D67V} as cause of the disease in the patients, we would like to point out that the occurrence per se of Golgi impairment and brain/axis phenotype collected respectively in cells and fish upon overexpression (and not downmodulation) of ARF3^{K127E} and ARF3^{L12V/D67V} and not of ARF3^{WT}, rejects the hypothesis of a loss-of-function effect of these mutants. Moreover, following the reviewer's suggestion (see below), we have also expanded our in vivo analysis for all the mutants in direct comparison to arf3 downregulation to distinguish the effects of downregulation vs overexpression and thereby better clarify the underlying genetic mechanism (**new Fig.7**). This new set of in vivo data further corroborates the presence of DN and CA mutations, showing severe and mild effects, respectively and resembling the patients' variable expression of the disease (**revised Supplementary Fig.1, Supplementary Table1, Supplementary clinical reports**). Overall, we think that the new in vitro and in vivo datasets provide a solid evidence of the dominant nature of the mutations, their variable impact of on cells and embryogenesis, supporting the heterogenous in the severity of the clinical traits, correctly pointed out by the reviewers. Of note, we now discuss also possible alternative dominant mechanism of action (i.e. "ARF GEF poisoning effect") as suggested by reviewer 1.

As indicated by Western blots, protein level of ARF3(K127E) or ARF3(L12V;D67V) was drastically decreased compared to ARF3-WT. Why didn't ARF3(K127E)-mCherry (Fig. 3b, 3c, and Fig. S5b) and ARF3(L12V;D67V)-mCherry (Fig. S5c) overexpressed in COS7 show weaker signals than ARF3-WT-mCherry?

Reply: we thank the reviewer for pointing this out and we understand the concern. Protein levels might vary in a heterogenous population of cells transiently overexpressing proteins of interest. In the submitted Fig.3b,c (**new Fig.4b,c**), differences in the captured z-layers between the samples in the initial frames (on the left of the figure) caused by the live nature of the experiment (time lapse) likely explains this apparent discrepancy. Moreover, it should be considered that WB analysis captures quantitatively the total protein levels from the whole transfected cell population from which the total cell lysate is collected, conversely to single cell/field microscopy which can single out variability within the population and whose sensibility depends on the laser and detector settings. It is therefore possible that by using the same confocal settings for all the mutants, as we did, cells whose fluorescence is extremely dim (likely in ARF3^{K127E} and ARF3^{L12V/D67V}) were not collected because hardly visible with minimal laser power. However, to fully address the reviewer's question, and to not rely only on the impression given by single images, we have quantitatively investigated the fluorescent intensity of ARF3 mCherry signal from a large population of cells in two technical and biological replicates. We assessed specifically the mutants showing accelerated protein degradation, i.e. ARF3^{K127E}, ARF3^{L12V/D67V} compared to WT ARF3. We also analyzed the fluorescence intensity for a population of cells ARF3^{D93N}, which shows instead stable protein expression in WB analysis and appears to be hyperactive. The data confirm the reduced protein levels for the p.K127E and p.L12V/D67V, that would be likely captured in the WB, while no change was measured for p.D93N. These data are now provided in the **new Supplementary Fig. 5**. Additionally, only very recently we started to establish viral-mediated DNA delivery as additional technique to obtain a more stable expression in cells upon selection. These efforts go beyond the scope and time-frame of this work; however, we would like to share that in our initial test line expressing mCherry-tagged ARF3^{K127E} using this alternative DNA transduction system we recapitulate the reduced levels of mCherry compared to those expressing ARF3^{WT} that we observed via transient transfections of the same construct. Of note, we document the same Golgi endophenotype in these cells, which support the validity of the functional data obtained in our transfection-mediated overexpression system, which generate a more heterogenous population. We do not think that we need to include this additional evidence in the revised manuscript (given also that it would still be available via the publication of this reply); however, we are open to include it if the reviewer and editor think it might be relevant.

In Fig. 3 and Fig. S5, co-localization of ectopically expressed ARF3-WT and mutants with EGFP-GaIT was analyzed. Based on some changes in the localization of EGFP-EGFP, the authors conclude that Golgi fragmentation occurred upon overexpression of ARF3 mutants. This conclusion definitely needs more evidence, e.g., from electronic microscopic observation. The mutant proteins may also affect vesicle formation and trafficking, which could be revealed, upon their overexpression, by immunofluorescence of other markers such as the trans-Golgi marker TGN46, the cis-Golgi marker GM130, the COPI complex marker beta-COPI, the early endosomal marker EEA1 (or others), and the endosomal marker RAB7.

Reply: following all the recommendations of reviewer 3, which are in line with the comments from reviewer 1, we have expanded and repeated the experiment addressing Golgi morphotypes for all mutants and we have included known AC and DN ARF mutants. We qualitatively and quantitatively characterized the Golgi integrity alterations, and we now report on the differential impact of the mutations on Golgi morphology (**new Fig.3, new Supplementary Fig. 7**). The overall loss of Golgi WT morphology and the presence of cells with dispersed or expanded phenotype was also confirmed in fish expressing ARF3^{K127E} and ARF3^{D93N} (**new Supplementary Fig.10**). In addition, to provide more details on the observed Golgi phenotype, as requested by the reviewer, we have now performed also a TEM analysis of the mutant ARF3^{K127E} and ARF3^{D93N}, which showed clearly distinguishable patterns of Golgi morphological alterations in our cell experiments. Indeed, our TEM results confirm the presence of two types of Golgi loss of integrity (**new Supplementary Fig.8**). The classical ribbon-like organization was not present in the mutants, and while ARF3^{K127E} exhibited loss of the mini-stacks with numerous diffused vesicles and small cisternae scattered in a wide area (**Supplementary Figure 8 b,b'**), cells expressing ARF3^{D93N} displayed a different pattern with marked increase in swollen cisternae and a diffuse vesiculation, within the defined area normally occupied by Golgi (**Supplementary Figure 8 c,c'**). A similar fragmentation pattern had previously been described for CA ARF mutants (doi: 10.1083/jcb.124.3.289., doi: 10.1074/jbc.M611716200.). Furthermore, we thank the reviewer for pointing to the possible effect of our mutants on vesicle formation and cargo recycling. Indeed, the involvement of ARF proteins in these functions is known and we have now expanded the Introduction and discussion, relatively to this important aspect. Moreover, following the reviewer's suggestion and to provide a deeper understanding of the possible mechanism of disease we have investigated COP-1 vesicle appearance in cells expressing all the identified ARF3 mutants (**new Fig. 5**) and have further assessed possible cargo accumulation and recycling defects by expression and immunostaining of a number of different markers of the endocytic-lysosomal pathway (as suggested by the reviewer) (**new Fig. 6, new Supplementary Fig.11-13**). Confirming the observations obtained by assessing Golgi morphology, disassembly of the COP-1 vesicles was observed for the severe mutants (p.K127E and p.L12V/D67V), while p.D93N and p.P47S showed an increased number of cells with an expansion of the compartment. p.T32N showed a variable and mild impact overall. The results resemble known CA and DN effect on vesicle formation for ARF (doi: 10.1091/mbc.e04-12-1042.). We assessed intracellular Tfn distribution upon 5 and 30 min of incubation, to follow its destiny upon internalization and assessed Tfn signal co-occurrence with early and recycling endosomal (Rab5 and Rab11) and lysosomal (Lamp2) markers, which worked well in our assay. These newly generated in vitro data provide an extra level of mechanistic understanding and are discussed in detail in the revised version of the manuscript. An accumulation of Tfn+ cargos was observed for the severe mutants (p.K127E and p.L12V/D67V causing early-onset microcephaly and severe spine deformity), which also showed an increased co-localization with EE and RE. On the other hand, an increased targeting of the Tfn to lysosomes was observed for p.D93N, causing late onset microcephaly and mild skeletal deformity in S4. For all mutations, lysosomes increased their loading with Tfn+ cargos. These experiments shed light on a differential impact of the mutants on vesicle formation and cargo recycling and provide additional functional evidence of their pleiotropic impact on cell physiology.

The authors assessed effect of Arf3 mutant overexpression in zebrafish embryos and observed pleiotropic effects on embryonic development. The effects are believed to be caused by dominant function of hyperactivity of the Arf3 mutants. This idea is mostly a speculation only. To test whether the identified Arf3 mutant alleles have a hyperactive or dominant negative activity, their overexpression effect should be compared with that of Arf3 knockout or Arf3 knockdown. Importantly, arf3 knockdown and Arf3 mutant overexpression can be co-performed in zebrafish embryos to test rescue effect. If the Arf3 mutants are truly hyperactive and dominant, their effect on embryonic development should be alleviated by knocking down endogenous arf3a/3b mRNAs.

*Reply: we apology for having misused “hyperactive” for all the mutants analyzed. We acknowledge that the term does not fully represent the heterogenous molecular, biochemical, and phenotypical condition of the disease. To assess the compatibility of the observed phenotype with a dominant mechanism associated with loss of function of ARF3 we have designed specific start codon Morpholino-based (MO) oligonucleotides against the endogenous zebrafish paralogs of arf3 to reduce arf3 expression in embryos, starting at early stages. In agreement with what had previously been reported for several dominant negative vs loss of function mutations of human developmental diseases (<https://doi.org/10.1073/pnas.1000219107>), our new datasets employing different concentrations of arf3a/b MO cause a subtle phenotypic effect compared to the mild/severe head reduction and body curvatures observed in fish injected with the dominant mutants and originally described. In our experiments, the MO phenotype was dose-dependent and was rescued by a small amount of WT ARF3, indicating the specificity of the effect. Of note, the use of MO was not able to rescue the effect for K127 and L12D67 and T32N, and these conditions did not improve the overall phenotype observed in mutants (“-MO” condition), as shown by the “worsening index” in revised **new Figure 7**, which corroborate the hypothesis of them acting as classical dominant negative. While such worsening degree is not observed for P47S, a significant improvement of the phenotype (alleviation) was documented for p. D93N, consistent with the CA behavior already anticipated by the other in vitro and in vivo data.*

From the point of view of developmental biologists, it is concerned whether ARF3 mutations affect function of endogenous maternal or zygotic ARF3. It is well known that a single signaling pathway can regulate neural induction and anteroposterior neuroectodermal patterning distinctly. It is possible that maternal and zygotic ARF3 play different roles. This should be tested experimentally.

*Reply: we thank the reviewer for this valuable note. We agree that now that this new disease-gene has been validated in vivo and that we establish new insight into its overlooked role in development it would be of great interest to investigate in more details the possible differential contribution of maternal vs zygotic arf3 on brain and axes formation. Indeed, deciphering the differential roles of these small GTPases (such as ARF3 and the other members of the ARF family) in comparative settings across vertebrate species is of fundamental interest in the field of developmental biology. Nevertheless, we would like to point out that this kind of question is outside the primary scope of this work. The aim of our study is instead focused on identifying the genetic cause of a previously unknown rare disease and functionally validate the pathogenicity of the variants identified as possible causative genetic lesion to the best of our possibilities (thereby including also in vivo models). We took advantage on our in vivo system to contextually gain insights into the function of ARF3 in development, which is also poorly investigated. It should be noted that a specific evaluation of the zebrafish maternal or zygotic function differences, that -as said- we were not set to determine, would require the establishment of a transgenic line to control the zygotic expression of ARF3 mutants. As the reviewer knows this requires lengthy government approval and line generation (at least in F3) to then investigate a potentially pathogenic effect of the mutant strain. Also, one is allowed to initiate the genetic modification for the line only once the permission is formally obtained and the time required for the completion of the whole process would most likely extend the time of this revision. In addition, the generation of this line is hardly justifiable from an ethical perspective, when carefully evaluating the benefit vs burden relation to generate this line (potentially pathogenic and therefore in pain) for the scope of the study. Indeed, respecting the 3R principle to Reduce, Replace and Refine (Russell W.M.S., Burch R.L. *The Principles of Humane Experimental Technique*. Methuen & Co Ltd.; London, UK: 1959.) and according to the Directive 2010/63/EU regulating animal experiment, the amount of evidence generated in this work by solely the transient models in embryos (and that do not involve adult animal procedures, generating F3 transgenic lines for which we have no permit) fulfilled the need to test the pathogenicity of ARF3 mutations for embryonic development and proved their role beyond doubt. Therefore, despite we consider important to define the maternal vs zygotic contribution of arf3 on brain/axis development, at this stage we feel that for the primary scope of proving the pathogenicity of the identified ARF3 variants, the additional generation of a stable mutant line might represent an unnecessary burden on animals. Such a mutant would be a valuable tool and we will surely take up the suggestion from the reviewer and consider it for a follow up development biology study, which would be more ethically acceptable per se once this original functional validation is publicly available. Nevertheless, to better define the relevance of the relative maternal vs zygotic contribution of the ARF3-mediated effect with the available model, and answer to the reviewer, we have first investigated the endogenous expression of arf3a/b during zebrafish early stages and in the time frame of the*

MZT (2.5/3 hpf) when transcripts which are maternally transmitted decrease and zygotic transcripts starts to accumulate (**new Supplementary Fig.9**). We observe transcription of *arf3* already at the maternal phase, and a stronger presence of *arf3* mRNA (*arf3b* especially) after the MZ transition, indicating that the major function of this protein starting from the zygotic time period onwards. Parallely, we have investigated the timing of the protein expression of our injected polyadenilated and capped mRNAs (**new Supplementary Fig.14**). The data show that the overexpressed protein starts to accumulate only around the MZT, and very little amount of WT and mutant protein is produced before this time point, especially for the highly degraded mutants (*p.K127E* shown as example) which actually generate microcephaly and axial defects. Therefore, although we cannot exclude at this point a maternal contribution to the phenotype observed, we believe the data strongly points towards a specific *arf3* zygotic function in the disease. In the revised manuscript we present the dataset and we discuss on the need to further clarify possible different contribution of maternal vs zygotic in the near future.

REVIEWERS' COMMENTS

Reviewer #1 (Remarks to the Author):

The authors have addressed my major concerns by providing extensive new data and by adjusting their interpretations and conclusions. I am happy to support publication of this revised manuscript.

One very minor suggestion, the term "endo-phenotype" used by the authors is rather obscure - I suggest changing this to something more conventional, like "phenotype" or "Golgi morphology".

Reviewer #2 (Remarks to the Author):

Manuscript NCOMMS-21-25633A

Dominant ARF3 variants disrupt Golgi integrity and cause a neurodevelopmental disorder recapitulated in zebrafish

Fasano G, et al

Comments to the authors

In this revised version, Fasano et al. have considerably improved their manuscript. The work done is impressive and the major issues I had raised have been properly and convincingly addressed. In particular, missing brain imaging data have been added, further supporting the NDD features described, many additional in vitro and in vivo validation experiments have been performed with appropriate controls and previously missing mutants, and importantly, all mutants have been modeled in fish showing the involvement of brain and axes malformations. Title, abstract and result section have been extensively re-worked to fit the presented data and the discussion is substantially and adequately expanded.

Minor comments

1. In the abstract, please mention that the variants identified involve the human ARF3 gene: "We report that de novo missense variants in human ARF3, encoding a small GTPase regulating Golgi dynamics, cause a developmental disease impairing CNS and skeletal formation"
2. Page 20, the reference 79 is not adequate. Please replace with more relevant references such as 10.3389/fcell.2021.784700 and 10.1101/gad.348866.121
3. Page 24 line 733: "in most" is missing: Microcephaly and progressive cerebral atrophy occurred in most affected individuals

Reviewer #3 (Remarks to the Author):

The author made great efforts to reveal mechanisms of heterogeneous functions of ARF3 mutations identified from human patients and improved the manuscript very much. Concerns raised by this reviewer have been mostly addressed.

Here are some minor points for modifying the manuscript further:

Lines 426-427, "in precursor cells (envelope layer cells, ELC) of early gastrula" is inappropriate. In the zebrafish community, the envelope layer is commonly called EVL. Expression as "in envelope layer (EVL) cells of ---" is better.

Fig. S9, the legend needs to be modified. First, what expression product, mRNA or protein, should be made clear. Second, the statement "Maternal arf3a transcripts slightly decrease just prior the onset of zygotic transcription (MZT)" may not stand because it may arise from experimental variations/errors (this happens often for RT-PCR analysis). Finally, the statement "the expression of arf3b paralogue is more consistent that (than) arf3a paralogue during development" may confuse readers without specification of developmental period.

Point to point response to the reviewers

General reply: we are glad that our revision work satisfies the reviewers' requests and concerns. We would like to thank all the reviewers and the Editor for their comments which guided us and contributed to improve our manuscript greatly. The mature work presented here now provides robust molecular, cellular and organismal-based evidence of this form of rare NDD.

Reviewer #1 (Remarks to the Author):

The authors have addressed my major concerns by providing extensive new data and by adjusting their interpretations and conclusions. I am happy to support publication of this revised manuscript.

Reply: we are glad that the reviewer understands the extent of the newly collected data and the revised discussion, which -we believe- was needed to address the important mechanistic questions from the reviewers. We thank the reviewer for the questions and comments, which contributed to the improved manuscript that we present now.

One very minor suggestion, the term “endo-phenotype” used by the authors is rather obscure - I suggest changing this to something more conventional, like “phenotype” or “Golgi morphology”.

Reply: In the newly revised manuscript, we followed the stylistic request and have replaced the term “endo-phenotype” with other synonyms such as “phenotype” and “morphology”.

Reviewer #2 (Remarks to the Author):

Manuscript NCOMMS-21-25633A

Dominant ARF3 variants disrupt Golgi integrity and cause a neurodevelopmental disorder recapitulated in zebrafish

Fasano G, et al

Comments to the authors

In this revised version, Fasano et al. have considerably improved their manuscript. The work done is impressive and the major issues I had raised have been properly and convincingly addressed. In particular, missing brain imaging data have been added, further supporting the NDD features described, many additional in vitro and in vivo validation experiments have been performed with appropriate controls and previously missing mutants, and importantly, all mutants have been modeled in fish showing the involvement of brain and axes malformations. Title, abstract and result section have been extensively re-worked to fit the presented data and the discussion is substantially and adequately expanded.

Reply: we are glad of the reviewer's acknowledgment of the extensive work performed during the revision in terms of i. new data collection and analysis and ii. re-writing, which addressed all

her/his/their concerns. We thank the reviewer for the questions and comments, which contributed to the improved manuscript that we present now.

Minor comments

- 1. In the abstract, please mention that the variants identified involve the human ARF3 gene: “We report that de novo missense variants in human ARF3, encoding a small GTPase regulating Golgi dynamics, cause a developmental disease impairing CNS and skeletal formation**
- 2. Page 20, the reference 79 is not adequate. Please replace with more relevant references such as 10.3389/fcell.2021.784700 and 10.1101/gad.348866.121**
- 3. Page 24 line 733: “in most” is missing: Microcephaly and progressive cerebral atrophy occurred in most affected individuals**

Reply: we thank the reviewer for this additional feedback. As requested, in the newly revised version of the manuscript, we have:

- 1- specified that the mutation affect a human protein in the abstract*
- 2- change the previous reference with the ones suggested by the reviewer*
- 3- corrected the typo indicated.*

Reviewer #3 (Remarks to the Author):

The author made great efforts to reveal mechanisms of heterogeneous functions of ARF3 mutations identified from human patients and improved the manuscript very much. Concerns raised by this reviewer have been mostly addressed.

Reply: we are happy that, likewise the other reviewers (see replies above), also the reviewer 3 agrees that the work we performed during the revision contributed to shed light on the mechanism underlying the NDD caused by different ARF3 mutations. Again, all the comments and questions raised by the reviewers were a major contribution to the improvement of the originally submitted manuscript and we thank the reviewer for this.

Here are some minor points for modifying the manuscript further:

Lines 426-427, “in precursor cells (envelope layer cells, ELC) of early gastrula” is inappropriate. In the zebrafish community, the envelope layer is commonly called EVL. Expression as “in envelope layer (EVL) cells of ---” is better.

Fig. S9, the legend needs to be modified. First, what expression product, mRNA or protein, should be made clear. Second, the statement “Maternal arf3a transcripts slightly decrease just prior the onset of zygotic transcription (MZT)” may not stand because it may arise from experimental variations/errors (this happens often for RT-PCR analysis). Finally, the statement “the expression of arf3b paralogue is more consistent that (than) arf3a paralogue during development” may confuse readers without specification of developmental period.

Reply: in the newly revised version of the manuscript we have addressed these points as follows:

- 1- we have replaced “ELC” with “EVL” cells;*
- 2- we have modified the legend of Supplementary Fig.9 by clarifying that the image relates to mRNA levels (first), we have rephrased the second statement and eliminated the third statement as they were unclear.*